# Mitochondrial dysfunction abrogates dietary lipid processing in enterocytes

Chrysanthi Moschandrea[1,2], Vangelis Kondylis[3,4], Ioannis Evangelakos[5], Marija Herholz[2,6], Farina Schneider[2,7], Christina Schmidt[2,7,8], Ming Yang[2,7,8], Sandra Ehret[5], Markus Heine[5], Michelle Y. Jaeckstein[5], Karolina Szczepanowska[2,6], Robin Schwarzer[1,2], Linda Baumann[2,6], Theresa Bock[1,2], Efterpi Nikitopoulou[8], Susanne Brodesser[2], Marcus Krüger[1,2], Christian Frezza[2,7,8], Joerg Heeren[5], Aleksandra Trifunovic[2,4,6 ✉] & Manolis Pasparakis[1,2,4 ✉]

Digested dietary fats are taken up by enterocytes where they are assembled into pre-chylomicrons in the endoplasmic reticulum followed by transport to the Golgi for maturation and subsequent secretion to the circulation[1]. The role of mitochondria in dietary lipid processing is unclear. Here we show that mitochondrial dysfunction in enterocytes inhibits chylomicron production and the transport of dietary lipids to peripheral organs. Mice with specific ablation of the mitochondrial aspartyl-tRNA synthetase DARS2 (ref. 2), the respiratory chain subunit SDHA[3] or the assembly factor COX10 (ref. 4) in intestinal epithelial cells showed accumulation of large lipid droplets (LDs) in enterocytes of the proximal small intestine and failed to thrive. Feeding a fat-free diet suppressed the build-up of LDs in DARS2-deficient enterocytes, which shows that the accumulating lipids derive mostly from digested fat. Furthermore, metabolic tracing studies revealed an impaired transport of dietary lipids to peripheral organs in mice lacking DARS2 in intestinal epithelial cells. DARS2 deficiency caused a distinct lack of mature chylomicrons concomitant with a progressive dispersal of the Golgi apparatus in proximal enterocytes. This finding suggests that mitochondrial dysfunction results in impaired trafficking of chylomicrons from the endoplasmic reticulum to the Golgi, which in turn leads to storage of dietary lipids in large cytoplasmic LDs. Taken together, these results reveal a role for mitochondria in dietary lipid transport in enterocytes, which might be relevant for understanding the intestinal defects observed in patients with mitochondrial disorders[5].

Mitochondrial dysfunction leads to deficiency in oxidative phosphorylation (OXPHOS) and metabolic defects that can affect almost any cell type and cause devastating diseases. Although mitochondrial diseases are usually described as encephalomyopathies, they often involve multiple organs, including the gastrointestinal (GI) tract[5,6]. The GI manifestations of mitochondrial diseases are frequently overlooked as they are considered either not life-threatening or nonspecific (for example, anorexia, abdominal pain, chronic constipation, diarrhoea or persistent vomiting). Defects in neuroendocrine and smooth muscle cells have been implicated in causing the GI manifestations, whereas the possible role of mitochondria in enterocytes remains largely unexplored[5,6]. Here we investigate the role of mitochondria in enterocytes, in particular in the processing and transport of dietary lipids.

## DARS2 deficiency causes lipid accumulation in IECs

To study the role of mitochondria in intestinal epithelial cells (IECs), we generated mice lacking DARS2 specifically in IECs by crossing *Dars2fl/fl*

mice[2] with *Vil1-cre* mice[7] (*Dars2fl/flVil1-cretg/wt*, hereafter referred to as *Dars2IEC-KO*). DARS2 deficiency inhibits the production of mitochondrial DNA (mtDNA)-encoded respiratory chain subunits and causes severe mitochondrial dysfunction[2]. *Dars2IEC-KO* mice were born at the expected Mendelian ratio but showed severely reduced body weight, failed to thrive and could not survive beyond the age of 4 weeks (Fig. 1a,b). Immunoblot analyses of total protein and mitochondrial protein extracts from primary IECs from 7-day-old *Dars2IEC-KO* pups revealed efficient ablation of DARS2 and strongly reduced levels of mtDNA-encoded respiratory chain subunits (CI, CIII, CIV and CV) (Fig. 1c and Extended Data Fig. 1a). Consequently, reduced formation of OXPHOS supercomplexes was detected in mitochondria from the small intestine (SI) of *Dars2IEC-KO* mice (Extended Data Fig. 1b). Enzyme histochemical staining showed strong cytochrome *c* oxidase (COX) deficiency, and electron microscopy (EM) analyses revealed swollen mitochondria with less densely packed and fragmented cristae in SI enterocytes of *Dars2IEC-KO* mice (Fig. 1d,e).

The SI of 7-day-old *Dars2IEC-KO* pups were considerably shorter than those of *Dars2fl/fl* littermates, and showed perturbed tissue architecture

[1]Institute for Genetics, University of Cologne, Cologne, Germany. [2]Cologne Excellence Cluster on Cellular Stress Responses in Aging-Associated Diseases (CECAD), University of Cologne, Cologne, Germany. [3]Institute for Pathology, Medical Faculty and University Hospital of Cologne, University of Cologne, Cologne, Germany. [4]Center for Molecular Medicine (CMMC), University of Cologne, Cologne, Germany. [5]Department of Biochemistry and Molecular Cell Biology, University Medical Center Hamburg-Eppendorf, Hamburg, Germany. [6]Institute for Mitochondrial Diseases and Aging, Medical Faculty, University of Cologne, Cologne, Germany. [7]Medical Faculty and University Hospital of Cologne, University of Cologne, Cologne, Germany. [8]MRC Cancer Unit, University of Cambridge, Hutchison/MRC Research Centre, Cambridge Biomedical Campus, Cambridge, UK. ✉e-mail: atrifuno@uni-koeln.de; pasparakis@uni-koeln.de

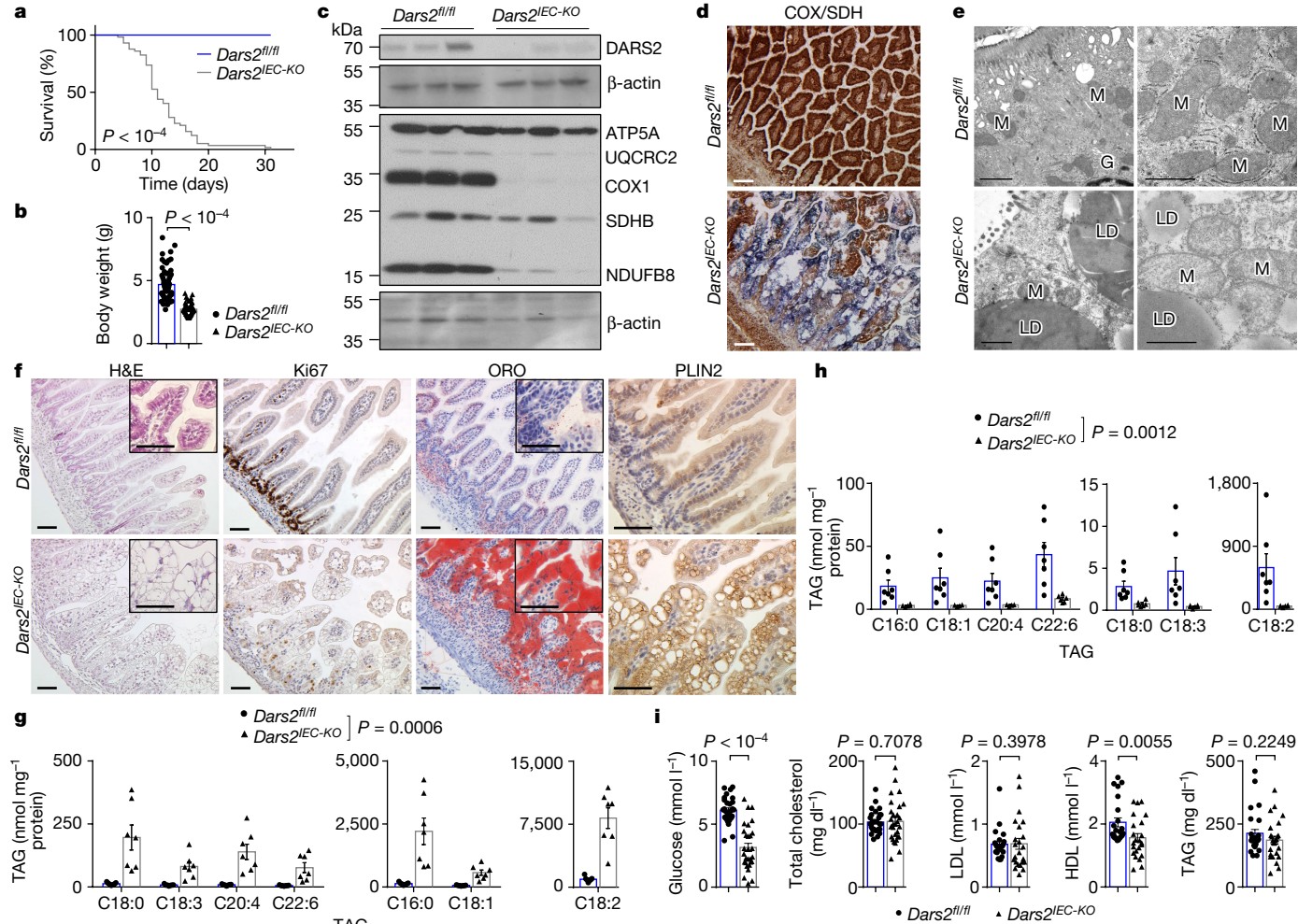

**Fig. 1 | _Dars2_[IEC-KO] mice develop severe intestinal pathology with massive lipid accumulation within large LDs in enterocytes. a,b**, Kaplan–Meier survival curves (**a**) and body weight at the age of 7 days (**b**) of _Dars2_[fl/fl] (_n_ = 56 (**a**), _n_ = 68 (**b**)) and _Dars2_[IEC-KO] (_n_ = 57 (**a**), _n_ = 66 (**b**)) mice. **c**, Immunoblot of IEC protein extracts from 7-day-old _Dars2_[fl/fl] (_n_ = 3) and _Dars2_[IEC-KO] (_n_ = 3) pups with the indicated antibodies. β-actin was used as the loading control. **d**, Representative images of SI sections from _Dars2_[fl/fl] and _Dars2_[IEC-KO] mice stained with enzyme histochemical staining for COX and SDH. **e**, Representative transmission electron microscopy (TEM) micrographs of SI sections from 7-day-old _Dars2_[fl/fl] and _Dars2_[IEC-KO] mice (_n_ = 3 per genotype). G, Golgi; M, mitochondria. **f**, Representative images of SI sections from _Dars2_[fl/fl] and _Dars2_[IEC-KO] mice stained with haematoxylin & eosin (H&E), ORO or immunostained

for PLIN2 and Ki67. **g,h**, TAG species content in SI (**g**) and liver (**h**) of _Dars2_[fl/fl] (_n_ = 7) and _Dars2_[IEC-KO] (_n_ = 7 SI, _n_ = 6 liver) mice. **i**, Concentration of glucose, total cholesterol, TAGs, HDL-cholesterol and LDL-cholesterol in sera from 7-day-old _Dars2_[fl/fl] and _Dars2_[IEC-KO] mice (_n_ = 29 (glucose, total cholesterol) per genotype; _n_ = 23 (HDL, LDL) per genotype; _n_ = 28, _n_ = 25 (TAG) for _Dars2_[fl/fl] and _Dars2_[IEC-KO], respectively). In **b,g–i**, dots represent individual mice, bar graphs show the mean ± s.e.m. and _P_ values were calculated using two-sided nonparametric Mann–Whitney _U_-test. In **a**, _P_ values were calculated using two-sided Gehan–Breslow–Wilcoxon test. In **d,f**, histological images are representative of the number of mice analysed as indicated in Supplementary Table 4. In **c**, each lane represents one mouse. Scale bars, 1 μm (**e**) or 50 μm (**d,f**). For gel source data, see Supplementary Fig. 1.

with blunted villi and lower numbers of Goblet cells and absorptive enterocytes (Fig. 1f and Extended Data Fig. 1c,d). Immunostaining for Ki67 revealed strongly decreased epithelial cell proliferation in intestinal crypts of _Dars2_[IEC-KO] mice (Fig. 1f), and reduced expression of _Olfm4_ and _Lgr5_ indicated a depleted stem cell compartment (Extended Data Fig. 1d). Immunostaining for cleaved caspase-3 and caspase-8 did not reveal increased numbers of dying cells in the intestines of _Dars2_[IEC-KO] mice. Similarly, immunostaining for CD45 and F4/80 did not reveal increased numbers of infiltrating immune cells (Extended Data Fig. 1e). A prominent microscopic feature of SI sections from _Dars2_[IEC-KO] mice was the presence of large cytoplasmic vacuoles in enterocytes (Fig. 1f). These vacuoles did not stain with periodic acid–Schiff, which detects glycoproteins and mucins, but stained positive with oil red O (ORO), which detects neutral lipids (Fig. 1f and Extended Data Fig. 1d), which suggested that the vacuoles correspond to large LDs. Indeed, immunostaining for perilipin 2 (PLIN2), a protein that coats LDs[1,8],

confirmed that IECs in _Dars2_[IEC-KO] mice contain large cytoplasmic LDs, which was in contrast to the few tiny LDs found in IECs from _Dars2_[fl/fl] mice (Fig. 1f). Accordingly, mass spectrometry (MS)-mediated lipidomics analysis revealed strongly increased levels of lipids, particularly of triacylglycerol (TAG) species in the intestine of _Dars2_[IEC-KO] pups (Fig. 1g and Extended Data Fig. 2a–f). By contrast, the livers of _Dars2_[IEC-KO] pups displayed a strong reduction in TAG levels compared with their _Dars2_[fl/fl] littermates (Fig. 1h). Moreover, reduced amounts of glucose and high-density lipoprotein (HDL) but normal levels of total cholesterol, low-density lipoprotein (LDL) and TAG were detected in the serum of _Dars2_[IEC-KO] mice (Fig. 1i). Additionally, IECs from _Dars2_[IEC-KO] mice showed reduced expression of several enzymes important for lipid biosynthesis (Extended Data Fig. 2g), which indicated that increased lipid synthesis is not the cause of fat accumulation in LDs. Collectively, these results suggest that DARS2 deficiency in enterocytes causes impaired transport of dietary lipids, which results in their accumulation within large LDs.

## LD accumulation in enterocytes lacking SDHA or COX10

We then asked whether the intestinal pathology caused by DARS2 deficiency could be reproduced through the ablation of specific OXPHOS subunits. We therefore generated mice lacking succinate dehydrogenase A (SDHA), an enzyme involved in the tricarboxylic acid cycle (TCA) and OXPHOS complex II, or lacking protohaem IX farnesyltransferase (COX10), an assembly factor of complex IV[3,4,9], specifically in IECs by crossing mice carrying respective *loxP*-flanked alleles with *Vil1-cre* mice. Both *Sdha^IEC-KO* and *Cox10^IEC-KO* mice were born at Mendelian ratios but developed a postnatal phenotype similar to that of *Dars2^IEC-KO* animals; that is, reduced body weight, failure to thrive and severe hypoglycaemia (Extended Data Fig. 3a–c,f–h). No considerable differences in total cholesterol, HDL, LDL and TAGs were detected in the serum of either *Sdha^IEC-KO* mice or *Cox10^IEC-KO* mice compared with their control littermates (Extended Data Fig. 3c,h). Immunoblot analyses confirmed efficient ablation of complex II and complex IV in *Sdha^IEC-KO* and *Cox10^IEC-KO* IECs, respectively, without affecting other OXPHOS subunits (Extended Data Fig. 3d,e,i,j). Histological analyses revealed impaired IEC proliferation and lipid accumulation within large LDs in enterocytes from both *Sdha^IEC-KO* mice and *Cox10^IEC-KO* mice (Extended Data Fig. 3d,i), as observed in *Dars2^IEC-KO* mice. Therefore, loss of specific subunits of respiratory chain complexes II or IV phenocopied the intestinal pathology induced by DARS2 deficiency in IECs. This result shows that mitochondrial dysfunction causes impaired transport and accumulation of lipids in enterocytes.

## DARS2 loss in adult IECs causes LD accumulation

The increased lipid accumulation in enterocytes of *Dars2^IEC-KO* pups could be related to the high fat content of milk or to developmental defects caused by DARS2 ablation during embryogenesis[7]. We therefore assessed the consequences of tamoxifen-inducible DARS2 deletion in IECs of adult *Dars2^fl/fl*Villin-creER^T2* mice (hereafter referred to as *Dars2^tamIEC-KO*) fed a normal chow diet (NCD). Tamoxifen administration on five consecutive days caused rapid weight loss that necessitated the euthanasia of *Dars2^tamIEC-KO* mice 7–8 days after the last injection (Fig. 2a and Extended Data Fig. 4a). Immunoblot and proteomics analyses of proximal IECs from *Dars2^tamIEC-KO* mice euthanized 7 days after tamoxifen injection confirmed efficient DARS2 ablation and severe depletion of OXPHOS subunits (Fig. 2b and Extended Data Fig. 4b,c). Gene set enrichment analysis (GSEA) of the proteomics data confirmed the depletion of OXPHOS subunits in IECs of *Dars2^tamIEC-KO* mice compared with *Dars2^fl/fl* littermates (Extended Data Fig. 4d and Supplementary Table 1). The proteomics analysis revealed that the ATF4-regulated pathway was among the most enriched signatures in DARS2-deficient enterocytes (Extended Data Fig. 4d,e). This result indicated that the mitochondrial integrated stress response was activated, a result previously reported in other models of mitochondrial dysfunction[10–12]. RNA sequencing (RNA-seq) confirmed that the ATF4 signature was strongly upregulated in the proximal SI of *Dars2^tamIEC-KO* mice 7 days after the last tamoxifen injection, which was also observed at 3 days after tamoxifen albeit to a lesser extent (Extended Data Fig. 4f,g and Supplementary Table 2).

GSEA of both the transcriptomics and proteomics data revealed downregulation of lipid metabolism pathways in DARS2-deficient enterocytes 7 days after tamoxifen induction (Extended Data Fig. 4d,f). These changes were not observed in RNA-seq data from *Dars2^tamIEC-KO* mice 3 days after tamoxifen administration (Supplementary Table 2), which suggested that they are not a primary consequence of DARS2 ablation but instead reflect a secondary response of the cells to the substantial metabolic alterations caused by mitochondrial dysfunction. Our data also showed reduced levels of proteins important for lipid biosynthesis at day 7 after tamoxifen injection, including fatty acid

synthase (FASN) and fatty acid binding protein 2 (FABP2) (Extended Data Fig. 5a,b). Over-representation analysis using the gene ontology terms (Supplementary Table 1) of the significantly changed proteins revealed that many lipid metabolism pathways were downregulated in DARS2-deficient enterocytes, and 'lipid droplet formation' was one of the most upregulated terms (Extended Data Fig. 5c,d). PLIN2 was the most highly induced protein in DARS2-deficient IECs in the proteomics dataset, which was verified by immunoblotting (Extended Data Fig. 5a,b). To further investigate the consequences of DARS2 ablation, we performed metabolomics analysis, which showed a broad metabolic deregulation in DARS2-deficient proximal IECs at 8 days after tamoxifen induction. The suppression of mitochondrial metabolism in *Dars2^tamIEC-KO* mice was corroborated by a reduction in aspartate[13,14] and an accumulation of succinate, which are hallmarks of OXPHOS dysfunction[15] (Extended Data Fig. 5e,f and Supplementary Table 3). Moreover, several glycolytic intermediates were accumulated in DARS2-deficient IECs, which suggested that the cells switched to glycolysis (Extended Data Fig. 5e,f). Consistent with a compensatory activation of glycolysis, the ratio of ATP to ADP, an indicator of the energy charge of the cell, was not reduced in DARS2-deficient enterocytes (Extended Data Fig. 5f). We also observed significant changes in purine and pyrimidine metabolism (Extended Data Fig. 5e), which was in line with the reported activation of mitochondrial integrated stress response in patients with mitochondrial disorders and in models of mitochondrial dysfunction[12,16–18]. Furthermore, IECs from *Dars2^tamIEC-KO* mice showed a marked accumulation of acylcarnitines, a result indicative of impaired fatty acid oxidation (Extended Data Fig. 5e).

Following necropsy of *Dars2^tamIEC-KO* mice, we observed a dilated, fluid-filled GI tract, with the proximal SI appearing white, which indicated massive lipid accumulation (Extended Data Fig. 6a). EM analyses showed that most mitochondria in *Dars2^tamIEC-KO* enterocytes appeared swollen, with less densely packed and fragmented cristae (Fig. 2c). Immunohistological evaluation revealed prominent respiratory chain deficiency, diminished numbers of proliferating cells, Goblet cells and absorptive enterocytes, and considerably reduced expression of stem cell markers in both the proximal and distal SI of *Dars2^tamIEC-KO* mice 7–8 days after tamoxifen induction (Fig. 2d and Extended Data Fig. 6b,c,e). Notably, immunostaining for cleaved caspase-3 and CD45 did not reveal increased numbers of dying cells or infiltrating immune cells, respectively, in the intestine of *Dars2^tamIEC-KO* mice (Extended Data Fig. 6b,d). This result shows that DARS2 deficiency does not induce enterocyte death or inflammation. As confirmation, an inflammatory gene expression signature was not observed in the RNA-seq data (Extended Data Fig. 4f and Supplementary Table 2). Similar to *Dars2^IEC-KO* mice, enterocytes in the proximal SI of *Dars2^tamIEC-KO* mice were filled with large LDs stained with ORO and PLIN2 (Fig. 2e). Lipidomics analyses also revealed increased TAG amounts in enterocytes from *Dars2^tamIEC-KO* mice (Fig. 2f). Serum glucose, TAGs and total cholesterol levels were not notably changed in *Dars2^tamIEC-KO* mice compared with control mice (Fig. 2g). In contrast to the proximal SI, enterocytes in the distal SI of *Dars2^tamIEC-KO* mice did not contain large LDs (Fig. 2e), which indicated that lipid accumulation occurred exclusively in proximal enterocytes, cells that are primarily responsible for the absorption, processing and transport of dietary fats[1]. To obtain insight into the kinetics of lipid accumulation, we examined intestinal tissue from *Dars2^tamIEC-KO* mice 3 and 5 days after the last tamoxifen injection (Extended Data Fig. 7). Efficient DARS2 ablation, decreased expression of OXPHOS subunits and strong suppression of IEC proliferation were detected in enterocytes from *Dars2^tamIEC-KO* mice 3 and 5 days after tamoxifen administration (Extended Data Fig. 7a–h). Although we did not detect signs of lipid accumulation at day 3 after tamoxifen injection, proximal enterocytes from *Dars2^tamIEC-KO* mice 5 days after tamoxifen induction contained small LDs, which indicated that lipid accumulation occurs already at this stage (Extended Data Fig. 7d,h). Collectively, mitochondrial dysfunction caused by inducible DARS2 ablation in IECs causes

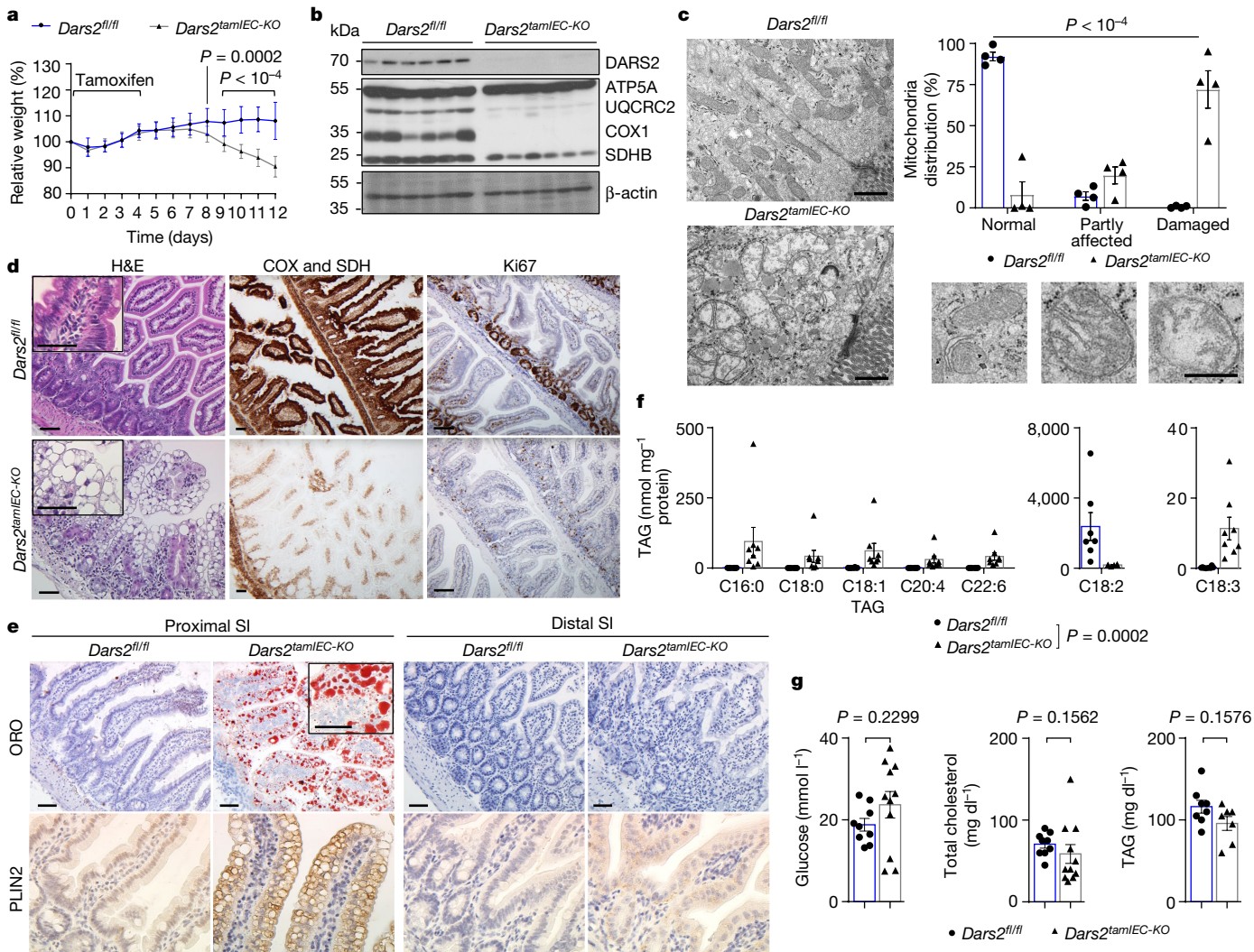

**Fig. 2 | Inducible DARS2 ablation in IECs of adult mice causes lipid accumulation in proximal enterocytes. a**, Relative body weight change of 8–12-week-old *Dars2^fl/fl* and *Dars2^tamIEC-KO* mice after tamoxifen administration (*n* = 21 per genotype). **b**, Immunoblot analysis with the indicated antibodies of protein extracts from SI IECs of *Dars2^fl/fl* and *Dars2^tamIEC-KO* mice 7 days after the last tamoxifen injection (*n* = 6 per genotype). β-actin was used as the loading control. **c**, Representative TEM micrographs of proximal SI sections and quantification of the mitochondria integrity distribution as a percentage of normal, partly affected and damaged mitochondria based on the electron density and cristae morphology in *Dars2^fl/fl* mice (*n* = 4 mice, *n* = 663 mitochondria in *n* = 69 IECs) and *Dars2^tamIEC-KO* mice (*n* = 4 mice, *n* = 707 mitochondria in *n* = 80 IECs) 7 days after tamoxifen. **d**, Representative images of sections from the proximal SI of *Dars2^tamIEC-KO* and *Dars2^fl/fl* mice stained with H&E, COX and SDH or immunostained with Ki67. **e**, Representative images of proximal and distal SI sections of *Dars2^fl/fl* and *Dars2^tamIEC-KO* mice stained with ORO or immunostained with PLIN2. **f**, TAG content in proximal SI of *Dars2^fl/fl* and *Dars2^tamIEC-KO* mice (*n* = 8 per genotype). **g**, Concentration of glucose, total cholesterol and TAGs in sera from *Dars2^fl/fl* mice (*n* = 9 (glucose, total cholesterol), *n* = 8 (TAG)) and *Dars2^tamIEC-KO* mice (*n* = 11 (glucose, total cholesterol), *n* = 7 (TAG)) 7 days after the last tamoxifen injection. In **c**, **f** and **g**, dots represent individual mice, bar graphs show the mean ± s.e.m. and *P* values were calculated using two-way analysis of variance (ANOVA) with Bonferroni's correction for multiple comparison (**a**), two-sided chi-square test (**c**) or two-sided nonparametric Mann–Whitney *U*-test (**f**,**g**). In **d**,**e**, histological images are representative of the number of mice analysed as indicated in Supplementary Table 4. In **b**, each lane represents one mouse. Scale bars, 1 μm (**c**) or 50 μm (**d**,**e**). For gel source data, see Supplementary Fig. 1.

substantial metabolic reprogramming and prominent accumulation of lipids in proximal enterocytes.

## Dietary lipids accumulate in *Dars2^tamIEC-KO* IECs

The accumulation of large LDs in proximal but not distal enterocytes of *Dars2^tamIEC-KO* mice, together with the overall downregulation of lipid biosynthesis pathways, indicated that the stored lipids probably originate from dietary fat. Dietary lipids emulsified by bile acids are digested by pancreatic lipase within the intestinal lumen to produce fatty acids, monoacylglycerols, cholesterol and lysophospholipids. These lipids are then taken up by enterocytes in the proximal SI where they are re-esterified into TAGs, cholesteryl esters (CEs) and phospholipids[1]. The majority of these lipids are then packaged into chylomicrons (CMs) that are released at the basolateral side and transported by the lymphatic system into the circulation and eventually to peripheral tissues[1]. Enterocytes also temporarily store excess dietary TAGs in cytosolic LDs, which are then mobilized for release in the form of CMs to ensure a stable supply of lipids between meals[1]. To assess the contribution of dietary fat, we examined whether feeding with a fat-free diet (FFD, containing <0.5% of fat), as opposed to a NCD (containing 3.4% fat), could prevent lipid accumulation in enterocytes of *Dars2^tamIEC-KO* mice (Extended Data Fig. 8a). Tamoxifen administration induced efficient DARS2 ablation, OXPHOS deficiency and weight loss in *Dars2^tamIEC-KO*

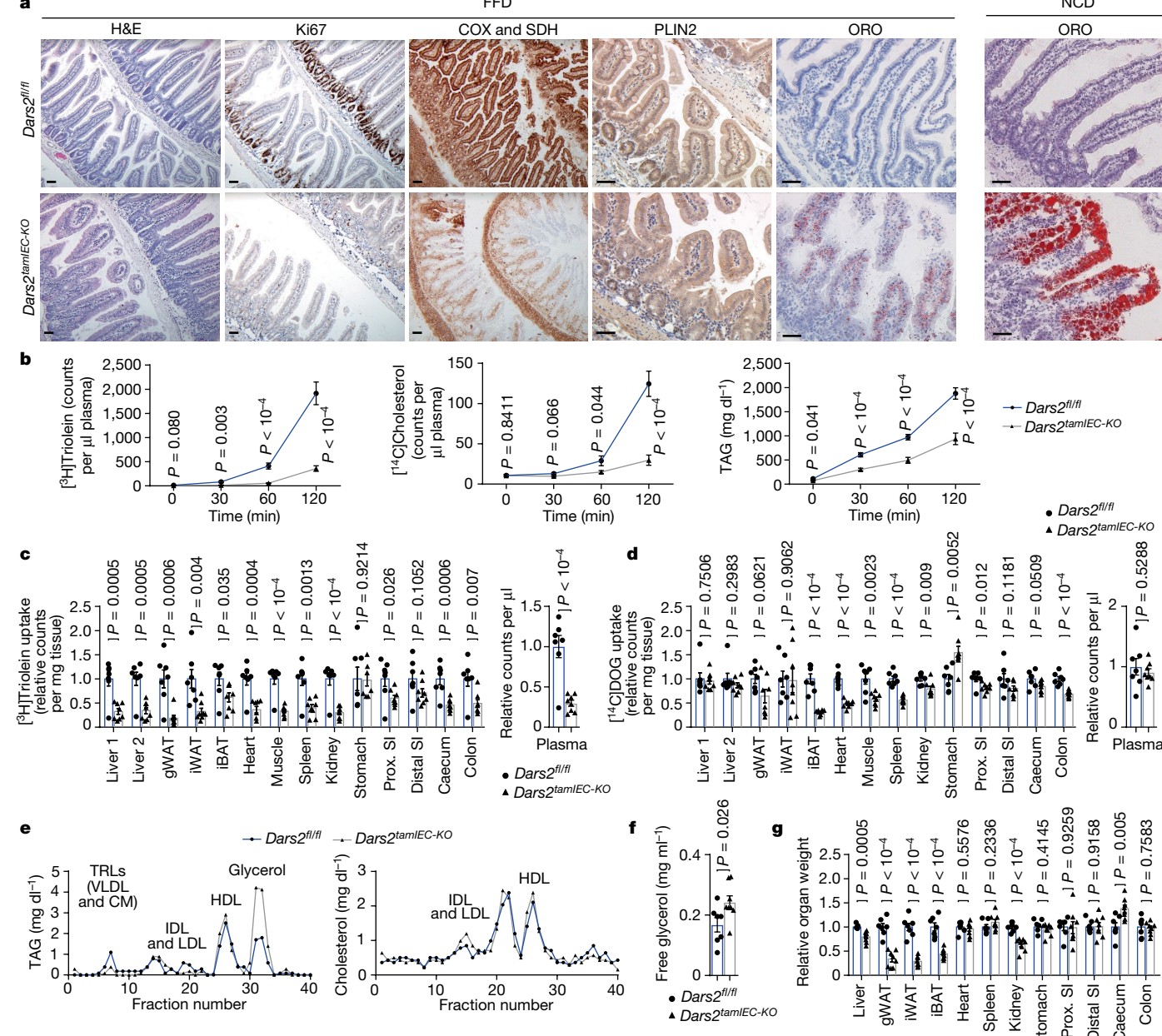

**Fig. 3 | DARS2 deficiency causes impaired transport of dietary lipids by enterocytes. a**, Representative images from the proximal SI of 8–12-week-old *Dars2^fl/fl* and *Dars2^tamIEC-KO* mice fed with a FFD or NCD diet 7 days after the last tamoxifen injection, stained with H&E, ORO and COX and SDH or immunostained with antibodies against Ki67 and PLIN2. Scale bar, 50 μm. **b**, [³H]Triolein, [¹⁴C] cholesterol and TAG content in portal plasma of *Dars2^fl/fl* mice (*n* = 8) and *Dars2^tamIEC-KO* mice (*n* = 8) subjected to oral fat tolerance tests after intravenous injection of tyloxapol. **c,d**, Counts of [³H]triolein (**c**) and [¹⁴C]DOG (**d**) in different organs and plasma from *Dars2^fl/fl* mice (*n* = 7) and *Dars2^tamIEC-KO* mice (*n* = 8) determined 120 min after oral gavage. iBAT, interscapular brown adipose tissue; gWAT, gonadal white adipose tissue; iWAT, inguinal white adipose tissue; Prox. proximal. Liver 1 and liver 2 correspond to two different mice fed the FFD. **e**, Fast-protein liquid chromatography profiles of TAG and cholesterol in pooled portal plasma from fasted *Dars2^fl/fl* mice (*n* = 7) and *Dars2^tamIEC-KO* mice (*n* = 8) 120 min after gavage. IDL, intermediate-density lipoprotein. **f**, Free glycerol levels in plasma from *Dars2^fl/fl* mice (*n* = 7) and *Dars2^tamIEC-KO* mice (*n* = 8). **g**, Relative organ weight of *Dars2^fl/fl* mice (*n* = 7) and *Dars2^tamIEC-KO* mice (*n* = 8) 7 days after the last tamoxifen injection subjected to oral glucose fat tolerance test. In **c,d,f,g**, dots represent individual mice, bar graphs show the mean ± s.e.m. and *P* values were calculated using unpaired two-sided Student's *t*-test with no assumption of equal variance (**b–d,f,g**). In **a**, histological images are representative of the number of mice analysed as indicated in Supplementary Table 4.

mice fed the FFD. These mice also showed impaired IEC proliferation and reduced stem cell gene expression in the SI (Fig. 3a and Extended Data Fig. 8), consistent with our findings in mice fed the NCD. However, *Dars2^tamIEC-KO* mice fed the FFD exhibited only a few small LDs in enterocytes of the proximal SI 7 days after the last tamoxifen injection. This was in contrast to the large and highly abundant LDs observed in mice fed the NCD (Fig. 3a). Therefore, feeding a FFD could strongly reduce

LD formation in DARS2-deficient enterocytes, thereby demonstrating that most accumulating lipids are derived from the diet. Notably, feeding a FFD delayed but could not ultimately prevent the substantial loss of body weight that necessitated the euthanasia of *Dars2^tamIEC-KO* mice. This result shows that lipid accumulation is not the primary cause of weight loss and death in these animals. This finding is in line with a previous study[19] reporting that IEC-specific deficiency of microsomal

triglyceride transfer protein (MTTP) induced lipid accumulation in enterocytes but did not cause death of the mice. Therefore, in addition to lipid accumulation, mitochondrial dysfunction causes defects in enterocytes such as the complete suppression of IEC proliferation, which probably led to the death of the animals.

## Impaired dietary lipid transport in *Dars2*[tamIEC-KO] mice

To directly assess whether DARS2 deficiency impairs the transport of dietary lipids by enterocytes, we performed metabolic tracing experiments. Specifically, we orally administered [$^3$H]triolein and [$^{14}$C]cholesterol to fasted *Dars2*[tamIEC-KO] mice and *Dars2*[fl/fl] littermates 7 days after tamoxifen injection and followed the appearance of the tracers in the plasma in the presence of the lipoprotein lipase inhibitor tyloxapol, which blocks intravascular lipoprotein processing (Extended Data Fig. 9a–c). Levels of [$^3$H]triolein, [$^{14}$C]cholesterol and TAGs were substantially decreased in the plasma of *Dars2*[tamIEC-KO] mice compared with *Dars2*[fl/fl] mice (Fig. 3b), which demonstrated that DARS2 deficiency inhibits the transport of dietary lipids by enterocytes. To further investigate how DARS2 deficiency in enterocytes affects the delivery of dietary lipids and glucose to peripheral organs, we orally administered [$^3$H]triolein and [$^{14}$C]deoxyglucose ([$^{14}$C]DOG) in fasted *Dars2*[tamIEC-KO] mice and *Dars2*[fl/fl] mice 7 days after tamoxifen induction and measured the accumulation of the tracers in different tissues (Extended Data Fig. 9d–f). Compared with *Dars2*[fl/fl] littermates, *Dars2*[tamIEC-KO] mice showed strongly reduced transport of [$^3$H]triolein to the plasma and most peripheral tissues, including the liver (Fig. 3c). By contrast, *Dars2*[tamIEC-KO] mice showed normal [$^{14}$C]DOG transport to the plasma (Fig. 3d), which indicated that loss of DARS2 in enterocytes predominantly affects the handling of dietary lipids. Notably, [$^{14}$C]DOG uptake by the liver was unaffected in *Dars2*[tamIEC-KO] mice but moderately decreased in other peripheral organs, including adipose tissue and heart (Fig. 3d). Moreover, profiling of plasma lipoproteins revealed reduced levels of TAG-rich lipoproteins (TRLs), including CMs and very low-density lipoproteins (VLDLs), in *Dars2*[tamIEC-KO] mice compared with *Dars2*[fl/fl] littermates, whereas HDL and LDL were not affected (Fig. 3e). The amount of plasma glycerol was increased whereas the weight of gonadal and inguinal white adipose tissue was reduced in *Dars2*[tamIEC-KO] mice 7 days after tamoxifen administration (Fig. 3e–g). This result suggested that adipose tissues undergo increased lipolysis, probably as a compensatory response to the impaired supply of dietary lipids. Metabolic tracing studies performed 5 days after the last tamoxifen injection revealed mildly reduced levels of [$^{14}$C]cholesterol and a trend towards reduced [$^3$H]triolein in the plasma of *Dars2*[tamIEC-KO] mice (Extended Data Fig. 9g–n). This finding indicated that already at this stage, the mice showed mild impairment of lipid transport. Uptake of [$^3$H]triolein in the liver and most peripheral tissues, with the exception of white adipose tissue, was not reduced in *Dars2*[tamIEC-KO] mice at 5 days after tamoxifen induction, a result consistent with its mildly impaired transport to the circulation (Extended Data Fig. 9m). Assessment of [$^{14}$C]DOG levels 5 days after tamoxifen injection revealed increased accumulation of the tracer only in the intestine of *Dars2*[tamIEC-KO] mice (Extended Data Fig. 9n), which could be related to the re-programming of cellular metabolism towards glycolysis in DARS2-deficient enterocytes. Taken together, the metabolic tracing experiments revealed a progressive impairment of dietary lipid transport to the circulation after enterocyte-specific ablation of DARS2.

## DARS2 loss impairs CM production and Golgi architecture

Most dietary lipids absorbed by enterocytes are transported to the circulation in the form of CMs[1,20]. CM production requires MTTP-mediated packaging of lipids into pre-CMs with ApoB48 in the endoplasmic reticulum (ER), followed by their transfer within pre-CM transport vesicles to the Golgi for maturation and subsequent extracellular secretion[1]. Primary enterocytes from *Dars2*[tamIEC-KO] mice 5 and 7 days after tamoxifen injection expressed normal levels of ApoB48 (Fig. 4a). However, triglyceride-rich lipoproteins isolated from the plasma of *Dars2*[tamIEC-KO] mice contained reduced levels of ApoB48 in relation to the liver-derived ApoB100 at 5 and 7 days after tamoxifen induction, a result consistent with decreased CM release from the intestine to the circulation (Fig. 4b). Ultrastructural examination of proximal SI sections showed that CMs were prominent within extended Golgi cisternae or were secreted across the basolateral surfaces of the intestinal epithelium of control *Dars2*[fl/fl] mice (Fig. 4c). By contrast, an extensive disorganization of the secretory pathway with a substantial lack of Golgi cisternae containing CMs was observed in enterocytes of *Dars2*[tamIEC-KO] mice (Fig. 4c). Instead, the cytoplasm of DARS2-deficient enterocytes was packed with very large LDs (Fig. 4c, arrowheads). Lipid particles were also often found within the ER lumen of enterocytes in *Dars2*[tamIEC-KO] mice (Fig. 4c, arrows). To further examine the integrity of the Golgi and the secretory pathway in enterocytes, we immunostained proximal SI sections from *Dars2*[fl/fl] and *Dars2*[tamIEC-KO] mice for *trans*-Golgi network integral membrane protein 1 (TGN38), a transmembrane protein localized to the Golgi[21], and E-cadherin, an integral membrane protein that is transported to the plasma membrane through the secretory pathway. TGN38 staining revealed a typical compact juxtanuclear Golgi network in enterocytes from *Dars2*[fl/fl] mice and a predominantly plasma membrane localization of E-cadherin (Extended Data Fig. 10a). By contrast, a substantial dispersal of TGN38 staining was observed in proximal SI enterocytes from *Dars2*[tamIEC-KO] mice at 8 days after tamoxifen injection, which was accompanied by strongly reduced levels of E-cadherin at the plasma membrane (Extended Data Fig. 10a). Time course analyses revealed that the Golgi network was largely unaffected at day 3 and partially fragmented at day 5 after tamoxifen induction, which indicated the occurrence of progressive Golgi disorganization after DARS2 loss (Extended Data Fig. 10a). To address whether Golgi disorganization precedes LD formation, we immunostained intestinal tissue sections with antibodies against TGN38, E-cadherin and PLIN2. Substantial LD formation concomitant with strong Golgi dispersal was observed in DARS2-deficient proximal enterocytes 8 days after tamoxifen injection (Extended Data Fig. 10b). A clear fragmentation of the Golgi network was observed in most proximal enterocytes in *Dars2*[tamIEC-KO] mice 5 days after tamoxifen treatment, whereas only a few LDs were detected in a small fraction of DARS2-deficient enterocytes at this stage (Fig. 4d). Therefore, Golgi dispersal occurs progressively after DARS2 ablation and precedes large LD formation. Notably, enterocytes in distal SI from *Dars2*[tamIEC-KO] mice showed only a mild disorganization of Golgi network and absence of LDs 8 days after tamoxifen injection (Extended Data Fig. 10b). This result suggests that bulk transport and secretion of dietary lipids may accelerate the disorganization of the Golgi network in proximal SI enterocytes. Moreover, proximal enterocytes from FFD-fed *Dars2*[tamIEC-KO] mice showed a partial fragmentation of Golgi network and absence of LDs at 7 days after tamoxifen induction (Extended Data Fig. 10c). Together, these results suggest that impaired production and/or ER-to-Golgi trafficking of CMs is probably an early event associated with Golgi disorganization. Our proteomics data confirmed that several proteins involved in CM production[1,22,23] (CD36, APOA4, APOA1, MTTP and LSR) and COPII vesicle budding[24] (SEC16, SEC23, SEC24 and SEC31) were downregulated in DARS2-deficient enterocytes (Extended Data Fig. 5a). We then assessed whether mitochondrial dysfunction could cause Golgi disorganization in IEC-6 cells, which are derived from rat SI epithelium and display typical characteristics of normal SI enterocytes[25]. Similar to our in vivo findings in *Dars2*[tamIEC-KO] enterocytes, treatment with actinonin, a mitochondrial protein synthesis inhibitor[10] that mimics DARS2 deficiency, or atpenin A5, an inhibitor of SDH[26], induced Golgi dispersal in IEC-6 cells (Extended Data Fig. 11a,b). As a positive control, we treated cells with brefeldin A, an inhibitor of the anterograde transport from the ER to Golgi and that causes Golgi

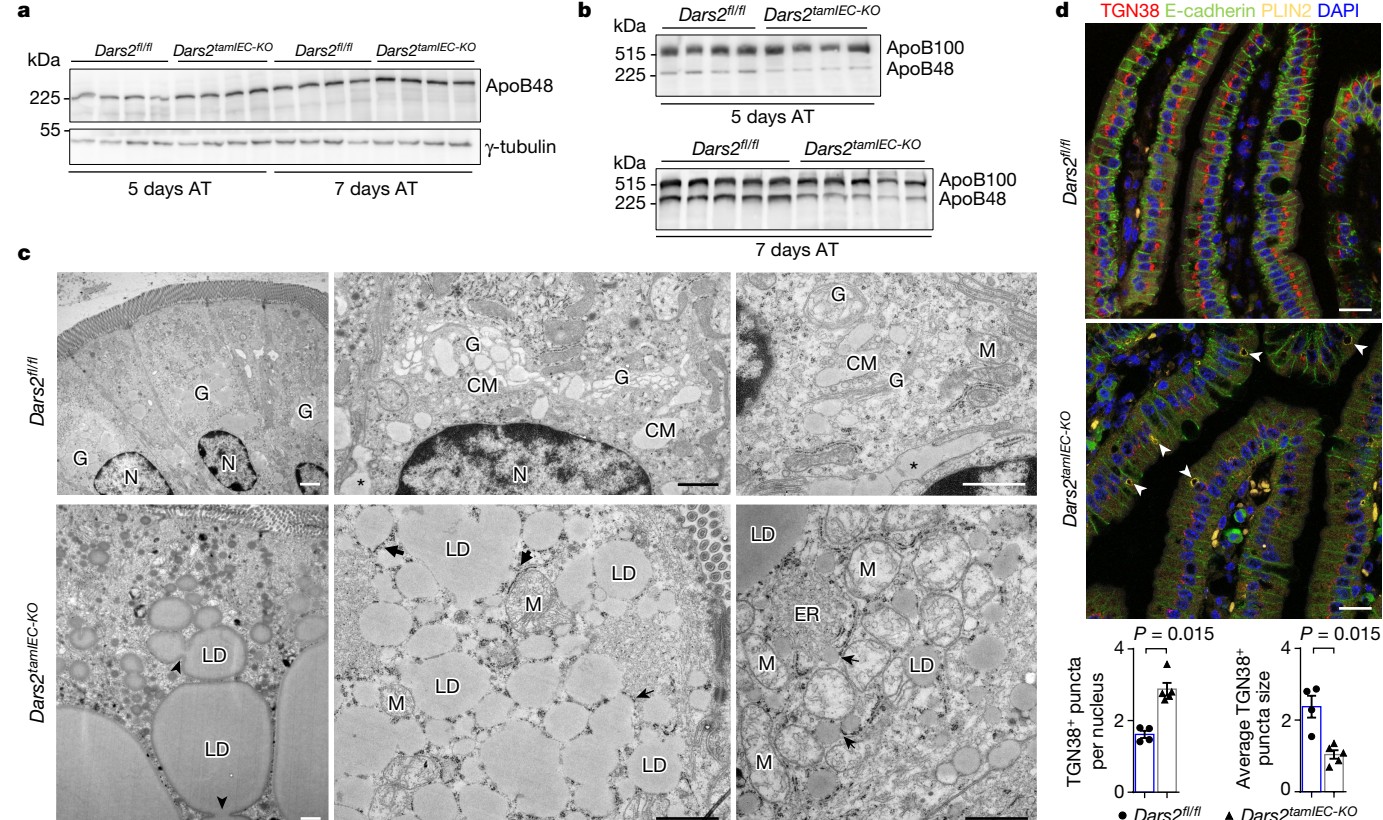

**Fig. 4 | DARS2 deficiency impairs CM production and induces progressive Golgi disorganization that precedes LD accumulation in enterocytes.**
**a**,**b**, Immunoblots depicting expression levels of ApoB48 in SI IECs (**a**) and ApoB48 and ApoB100 on TRLs isolated from plasma by ultracentrifugation (**b**) from *Dars2*[fl/fl] and *Dars2*[tamIEC-KO] mice 5 and 7 days after the last tamoxifen injection (**a**, *n* = 4 mice per genotype, per indicated time point; **b**, *n* = 5 mice per genotype at 7 days after tamoxifen, *n* = 4 mice per genotype at 5 days after tamoxifen). γ-tubulin was used as the loading control (**a**). AT, after tamoxifen. **c**, Representative TEM micrographs from proximal SI sections of *Dars2*[fl/fl] mice (*n* = 4) and *Dars2*[tamIEC-KO] mice (*n* = 4) 7 days after the last tamoxifen injection. Note the lack of CM-containing Golgi complexes and the appearance of aberrant numbers of LDs and damaged mitochondria in DARS2-deficient enterocytes. Asterisks indicate CMs secreted in the basolateral intercellular space. Arrows point at lipid particles within the ER lumen in rough and smooth ER neighbouring areas. Arrowheads point at LD lateral fusion. N, nucleus. **d**, Top, representative fluorescence microscopy images from the proximal SI of 8–12-week-old *Dars2*[fl/fl] mice (*n* = 6) and *Dars2*[tamIEC-KO] mice (*n* = 6) immunostained with antibodies against TGN38, E-cadherin and PLIN2. Nuclei stained with DAPI. Arrowheads point at LDs. Bottom, quantification of the TGN38-positive puncta size and the number of puncta per nucleus from confocal images of proximal SI sections of *Dars2*[fl/fl] mice (*n* = 4) and *Dars2*[tamIEC-KO] mice (*n* = 5) 5 days after the last tamoxifen injection. In **d**, dots represent individual mice, bar graphs show the mean ± s.e.m. and *P* values were calculated using unpaired two-sided Student's *t*-test with no assumption of equal variance. In **a**,**b**, each lane represents one mouse from two independent experiments. Scale bars, 1 μm (**c**) or 50 μm (**d**). For gel source data, see Supplementary Fig. 1.

membranes to be absorbed into the ER[27]. Notably, addition of oleic acid in the medium exacerbated the Golgi disorganization induced by OXPHOS inhibitors, concomitant with lipid accumulation in large LDs (Extended Data Fig. 11b). Thus, inhibition of mitochondrial function induces Golgi disorganization in a cellular system and this effect is exacerbated in the presence of high levels of lipids in the medium.

To assess whether DARS2 depletion affects the Golgi in the intestine of *Caenorhabditis elegans*, we took advantage of a transgenic strain that expresses the Golgi-specific a-mannosidase II fused to GFP under the control of the gut-specific *vha-6* promoter[28]. *Dars-2* RNA-mediated interference considerably reduced the amount of Golgi puncta in the gut of *C. elegans* at adult days 1 and 4 without affecting ER morphology, as evaluated using a strain that expresses the ER-specific SPCS-1 fused to GFP[29] (Extended Data Fig. 11c,d). Next, we investigated whether DARS-2 deficiency would affect the trafficking of lipids. To this end, we followed the transport of vitellogenin 2 (VIT-2) in *C. elegans*[30]. Vitellogenins are large lipo-glyco-phosphoproteins that recruit lipids and require COPII vesicle trafficking for their secretion into the circulation to be ultimately taken up by oocytes through receptor-mediated endocytosis[31]. In *C. elegans*, VIT-2 (visualized using the VIT-2::GFP reporter) is predominantly found in oocytes, but in the absence of the small GTPase

SAR-1, a homologue of SAR1B that is essential for budding of ER-derived COPII vesicles that mediate secretory cargo transport to the Golgi[1,32,33], it accumulates in the intestine[30] (Extended Data Fig. 11e). Notably, *dars-2* depletion strongly prevented VIT-2 transport into oocytes (Extended Data Fig. 11e). By contrast, depletion of fumarate hydratase (FUM-1), a mitochondrial TCA cycle enzyme essential for numerous metabolic processes that also acts as a tumour suppressor[34], did not affect VIT-2 transport (Extended Data Fig. 11e). Further supporting our results and the role of mitochondria in the transport of lipoprotein complexes in *C. elegans*, a screen for factors essential for vitellogenin transport identified 40 mitochondrial proteins, out of which 28 directly regulate mitochondrial protein synthesis, including 8 mitochondrial tRNA amino-acyltransferases, enzymes that belong to the same family as DARS2 (ref. 30).

## Discussion

Taken together, our results revealed an essential and evolutionarily conserved role of mitochondria in dietary lipid processing by enterocytes. The LD accumulation phenotype caused by mitochondrial dysfunction in IECs is reminiscent of the pathology of human patients with

mutations in the gene encoding SAR1B, who suffer from CM retention disease that manifests with chronic diarrhoea, intestinal distension and growth retardation in infancy[1,32,33]. SAR1B deficiency prevents the trafficking of pre-CMs from the ER to the Golgi, which causes impaired transport of dietary lipids to the circulation and their accumulation within large LDs in enterocytes[1,32,33]. Mice with IEC-specific ablation of MTTP, which is essential for CM production, showed impaired transport and accumulation of dietary lipids in enterocytes[19]. Furthermore, brefeldin A treatment in rats suppresses CM production, which results in impaired transport of dietary fat and lipid accumulation in large LDs in enterocytes[35]. These results provide evidence that Golgi disorganization inhibits dietary lipid processing. The precise mechanism by which mitochondrial defects affect secretory pathway organization and function remains to be fully elucidated. However, our findings suggest that mitochondrial dysfunction impairs CM formation and/or trafficking from the ER to the plasma membrane, which results in compromised transport of dietary lipids to peripheral tissues and their accumulation and storage within large cytoplasmic LDs in enterocytes. These findings could be relevant for the understanding of the mechanisms that cause intestinal complications associated with the severe inability to gain weight and failure to thrive in a subset of patients with mitochondrial disease[6,36].

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

# Methods

## Mice

The following mouse lines were used: *Dars2*[fl/fl] (ref. 2), *Cox10*[fl/fl] (ref. 4), *Vil1-cre* (ref. 7) and *Villin-creER*[T2] (ref. 37). *Sdha*[tm2a] mice were obtained from the Knock Out Mouse Project repository (project ID: CSD48939) and bred to FLP deleter mice[38] to delete the FRT-flanked region to generate *Sdha*[fl/fl] mice. IEC-specific knockout mice were generated by intercrossing mice carrying the respective *loxP*-flanked alleles with *Vil1-cre* or *Villin-creER*[T2] transgenic mice. Both female and male mice between 1 and 12 weeks of age were used in all in vivo experiments, whereas metabolic tracing studies were performed exclusively using male mice. All mice were maintained on the C57BL/6N background. Mice were housed at the specific-pathogen-free animal facilities of the CECAD Research Center of the University of Cologne under a 12-h dark–12-h light cycle in individually ventilated cages (Greenline GM500, Tecniplast) at 22 ± 2 °C and a relative humidity of 55 ± 5%. All mice had unlimited access to water and fed a standard chow diet (Harlan diet no. 2918 or Prolab Isopro RMH3000 5P76) ad libitum. For the experiments assessing the role of dietary fat, mice were fed a FFD (E15104-3474, ssniff-Spezialdiäten) containing only traces of fat (<0.5%). All animal procedures were conducted in accordance with European, national and institutional guidelines and protocols were approved by local government authorities (Landesamt für Natur, Umwelt und Verbraucherschutz Nordrhein-Westfalen) and Animal Welfare Officers of the University Medical Center Hamburg-Eppendorf and Behörde für Gesundheit und Verbraucherschutz Hamburg. Animals requiring medical attention were provided with appropriate care and were culled humanely when reaching pre-determined termination criteria to minimize suffering. No other exclusion criteria were applied. Villin-CreER[T2] recombinase activity was induced by five consecutive daily intraperitoneal administrations of 1 mg tamoxifen dissolved in corn oil and DMSO. Littermates not carrying the *Vil1-cre* or *Villin-creER*[T2] transgenes were used as controls in all experiments.

## Tissue preparation

The colon and SI were dissected and washed with PBS. Small pieces (about 0.5 cm) were isolated proximal (after the stomach) and distal (before the caecum) of the SI, snap-frozen on dry ice for RNA expression analysis and stored at −80 °C until further processing. The remaining SI tissue was cut longitudinally and washed in PBS to remove faeces. Intestinal tissue samples were rolled up from proximal to distal to form a Swiss roll and either fixed in 4% paraformaldehyde overnight at 4 °C or embedded in TissueTek for frozen sectioning.

## H&E staining of paraffin-fixed tissues

Paraffin-embedded 3-µm-thick intestinal tissue sections were deparaffinized with xylene and rehydrated with decreasing amounts of ethanol solutions (100% ethanol, 96% ethanol and 75% ethanol). Sections were stained for 2 min in haematoxylin, differentiated in tap water for 15 min and incubated for 1 min in eosin. Stained sections were dehydrated using increasing amounts of ethanol solutions and fixed in xylene for 1 min. Slides were mounted with Entellan.

## COX and SDH and ORO staining of fresh-frozen tissues

Fresh-frozen 7-µm-thick intestinal sections were sequentially stained for COX and SDH activity. Cryosections were dried and incubated for 45 min at 37 °C with COX solution. Then they were briefly washed with PBS and incubated for 40 min with SDH solution at 37 °C. Following dehydration through graded alcohol solutions, the sections were mounted with DPX and stored at room temperature. Fresh-frozen 10-µm-thick sections were fixed in 4% paraformaldehyde for 15 min at room temperature. After fixation, the sections were washed with ddH$_2$O and stained with ORO in isopropanol/water (60:40) for 15 min.

All sections were counterstained with haematoxylin for 5 min and mounted with Aquatex (EMD Millipore).

## Immunohistochemistry and immunofluorescence on intestinal sections

Paraffin sections were rehydrated and heat-induced antigen retrieval was performed in 10 mM sodium citrate, 0.05% Tween-20 at pH 6.2 or with proteinase K treatment. Endogenous peroxidase was blocked in peroxidase blocking buffer for 15 min at room temperature. Sections were blocked in 1% BSA, 0.2% fish-skin gelatin, 0.2% Triton-X-100 and 0.05% Tween-20 in PBS for 1 h at room temperature. After blocking, the sections were incubated overnight at 4 °C with primary antibodies against adipophilin/PLIN2 (Progen, GP46, 1:500), Ki67 (Dako, M724901, clone 1O15, 1:1,000), OLFM4 (Cell Signaling, D6Y5A, clone D6X5A, 1:400), CC3 (Cell Signaling, 9661, 1:1,000), CC8 (Cell Signaling, 8592, 1:1,000), CD45 (BD Bioscience, 560510, clone 30-F11, 1:500) and F4/80 (AbD Serotec, MCA497, clone A3-1, 1:1,000). Sections were incubated with biotinylated anti-mouse IgG (H+L) (Vector Laboratories, BA-9200-1.5, 1:1,000), anti-rabbit IgG (H+L) (Vector Laboratories, BA-1000-1.5, 1:1,000) and anti-rat IgG (H+L) (Vector Laboratories, BA-9400-1.5, 1:1,000) secondary antibodies. Each staining was visualized using ABC Kit Vectastain Elite (Vector, PK6100) and DAB substrate (Dako and Vector Laboratories). Immunofluorescence was performed with primary antibodies against TGN38 (bio-techne, AF8059-SP, 1:200), E-cadherin (BD Biosciences, 610182, 1:1,000) and adipophilin/PLIN2 (Progen, GP46, 1:200). Nuclei were stained using DAPI (Vector Laboratories) and visualized with anti-sheep IgG NorthernLights NL557 (bio-techne, NL010, 1:300), anti-mouse Alexa 488 (Molecular Probes, A1101, 1:300) and anti-guinea pig Alexa 633 (Molecular Probes, A21105, 1:300) fluorescence-conjugated secondary antibodies. Periodic acid–Schiff (PAS) reaction was performed according to standard protocols. Endogenous alkaline phosphatase activity was visualized using a Fast Red Substrate kit according to the manufacturer's instructions (ab64254, Abcam). For image acquisition, the intestinal sections were analysed using a light microscope equipped with a KY-F75U digital camera (JVC) (DM4000B, Leica Microsystems, Diskus 4.50 software), a TCS SP8 confocal laser scanning microscope (Inverse, DMi 8 CS, Leica Microsystems LAS X, Lightning software v.5.1.0) or a LSM Meta 710 confocal laser scanning microscope (Carl Zeiss Technology, ZEN 2009 software). Golgi quantification was performed using ImageJ software (v.2.0.0.-rc-46/1.50g) as previously described[39]. The number and size of TGN38-positive fluorescent objects were quantified using the 'analyse particles' function after applying a fixed threshold on pictures derived from maximal 2D projections of the acquired confocal stacks. Each data point corresponds to the average values from at least three randomly selected intestinal areas of a single mouse. Representative pictures from 4–5 mice per genotype per time point were analysed. More than 100 IEC profiles per mouse with visible nuclei were quantified (*n* = 128–527).

## EM analysis

A piece of 0.5 cm proximal SI tissue was fixed overnight in 2% glutaraldehyde (Merck) and 2% paraformaldehyde (Science Services) in 0.1 M cacodylate buffer (AppliChem). Tissue samples were treated with 1% OsO$_4$ (Science services) in 0.1 M cacodylate buffer for 2 h. After dehydration of the sample with ascending ethanol concentrations followed by propylene oxide, samples were embedded in Epon (Sigma-Aldrich). Ultrathin sections (70 nm thick) were cut, collected onto 100 mesh copper grids (Electron Microscopy Sciences) and stained with uranyl acetate (Plano) and lead citrate (Sigma Aldrich). Images were captured using a transmission electron microscope (Joel JEM2100 Plus) at an acceleration voltage of 80 kV, and pictures were acquired using a 4K-CCD camera, OneView (GATAN). Mitochondrial morphological integrity quantification was performed on randomly selected pictures of the proximal SI areas from four *Dars2*[fl/fl] and four *Dars2*[tamIEC-KO] mice. Each mitochondrial profile was classified as normal, partly affected or

severely damaged based on its electron density, the appearance of the cristae and the extent of matrix loss (Fig. 2c). The relative distribution of the analysed mitochondria per mouse into the three morphological groups is presented. A total of 663 mitochondrial profiles from 69 IECs versus 707 mitochondrial profiles from 80 IECs were quantified.

## Cell culture conditions and drug treatments

IEC-6 cells (ACC 111) were purchased from the Leibniz Institute DSMZ–German Collection of Microorganisms and Cell Cultures and maintained in standard conditions at 37 °C and 5% $CO_2$. The cell culture medium was composed of 45% Dulbecco's modified Eagle medium (ThermoFisher, 41965-039), 45% RPMI 1640 (ThermoFisher, 11875093) and 0.1 U ml$^{-1}$ human insulin solution (Sigma, I9278) supplemented with 10% FCS (Bio&SELL). IEC-6 cells were routinely checked for mycoplasma contamination and tested negative. For induction of mitochondrial dysfunction, 70–80% confluent cells were treated for 48 h with 100 µM actinonin (A6671, Sigma-Aldrich) or 1 µM atpenin A5 (ab144194, Abcam). All compounds were solubilized in dimethyl sulfoxide (DMSO) (A3672, PanReac AppliChem). Control cells were treated with corresponding amounts of DMSO, which did not exceed 1% in culture medium. Treatments were renewed every 24 h. IEC-6 cells were incubated with 5 µg ml$^{-1}$ brefeldin A (B6542, Abcam) for 6 h. To induce LD formation, oleic acid (O1008, Sigma-Aldrich) was complexed to fatty acid-free BSA (A6003, Sigma-Aldrich) at a ratio of 6:1 and used at a concentration of 600 µM after titration for 24 h.

## Immunofluorescence of cultured cells

Immunofluorescence staining was performed on IEC-6 cells cultured on coverslips and fixed in 4% paraformaldehyde for 15 min. Reactive aldehydes were quenched with 50 mM $NH_4Cl$ for 10 min and the cells were permeabilized with 0.1% Triton-X-100 in PBS for 5 min. After 20 min in blocking solution (0.2% fish-skin gelatin diluted in PBS), IEC-6 cells were incubated with primary antibodies against TGN38 (bio-techne, AF8059-SP, 1:200) and MTCO1/COX1 (Molecular Probes, 459600, 1D6E1A8, 1:100) for 30 min at room temperature, followed by incubation with anti-sheep IgG NorthernLights NL557 (bio-techne, NL010, 1:300) or anti-mouse Alexa 488 (Molecular Probes, A1101, 1:300) fluorescence-conjugated secondary antibodies for 30 min at room temperature. When LDs were stained, 5 µM of BODIPY 493/503 (D3922, Invitrogen) diluted in PBS was applied for 30 min. Finally, IEC-6 cells were mounted in Vectashield containing DAPI. For image acquisition, a TCS SP8 confocal laser scanning microscope (Inverse, DMi 8 CS, Leica Microsystems LAS X, Lightning software v.5.1.0) was used. Quantification of Golgi morphology was performed using ImageJ software (v.2.0.0.-rc-46/1.50g) on 2D projections from Z-stack images. A total of 4–6 randomly selected viewing fields per condition, capturing at least 30 cells per image, were used. Golgi morphology was classified into five distinct categories based on TGN38-positive fluorescent objects (Extended Data Fig. 11a,b) as follows: (1) normal (juxtanuclear Golgi ribbon composed of connected stacks); (2) ring (ring-like Golgi structures surrounding the entire nucleus); (3) condensed (bulb-shaped juxtanuclear Golgi structure); (4) fragmented (Golgi ribbon replaced by more and smaller tubules and vesicles positive for TGN38); and (5) dispersed (complete loss of Golgi ribbon and dispersal of the TGN38 signal). Quantification was performed by manually classifying the TGN38 pattern in each cell in one of the five Golgi phenotypes by the same observer, who was blinded to the experimental conditions. Three independent experiments were quantified.

## Measurement of serum parameters

Glucose (GLU2), total cholesterol (CHOL2), triacylglycerol (TRIGL), high-density lipoprotein (HDLC4) and low-density lipoprotein (LDLC3) levels in the blood serum from mice aged 1–12 weeks old were measured using standard assays in a Cobas C111 Biochemical Analyzer (Roche Diagnostics).

## Isolation of mitochondria and analysis of mitochondrial respiratory complexes with blue native electrophoresis

**Mitochondria isolation.** The SI was chopped into small pieces and homogenized with a rotating Teflon potter (Potter S, Sartorius; 20 strokes, 1,000 r.p.m.) in a buffer containing 100 mM sucrose, 50 mM KCl, 1 mM EDTA, 20 mM TES and 0.2% fatty acid-free BSA, pH 7.6 followed by differential centrifugation at 850$g$ and 8,500$g$ for 10 min at 4 °C. Mitochondria were washed with BSA-free buffer, and protein concentrations were determined using Bradford reagent. Mitochondria were subjected to blue native polyacrylamide gel electrophoresis (BN-PAGE) followed by western blot analysis or determination of the in gel activity of respiratory complexes.

**BN-PAGE.** Mitochondrial protein concentrations were determined using Bradford reagent (Sigma). A total of 20 µg of mitochondria was lysed for 15 min on ice in dodecylmaltoside (5 g g$^{-1}$ of protein) for individual respiratory complexes, or digitonin (6.6 g g$^{-1}$ protein) for supercomplexes, and cleared from insoluble material for 20 min at 20,000$g$, 4 °C. Lysates were combined with Coomassie G-250 (0.25% final). Mitochondrial complexes were resolved by BN-PAGE using 4–16% NativePAGE Novex Bis-Tris mini gels (Invitrogen) in a Bis-Tris/Tricine buffering system with cathode buffer initially supplemented with 0.02% G-250 followed by the 0.002% G-250.

**Complex I in-gel activity.** Gels were incubated in a buffer containing 0.01 mg ml$^{-1}$ NADH and 2.5 mg ml$^{-1}$ nitrotetrazolium blue in 5 mM Tris-HCl pH 7.4.

**Western blot analysis.** Separated mitochondrial complexes were transferred onto a polyvinylidene fluoride membrane using a wet transfer methanol-free system. Membranes were immunodecorated with indicated antibodies followed by ECL-based signal detection. The following antibodies were used: anti-MTCO1 (Molecular Probes, 459600, clone 1D6E1A8, 1:5,000), anti-COX4L1 (Molecular Probes, A21348, clone 20E8C12, 1:1,000), anti-UQCRC1 (Molecular Probes, 459140, clone 16D10AD9AH5, 1:4,000), anti-NDUFS1 (Proteintech, 12444-1-AP, 1:1,000), anti-NDUFS2 (Abcam, ab96160, 1:1,000), anti-NDUFV2 (Proteintech, 15301-1-AP, 1:1,000), anti-UQCRFS1/RISP[5A5] (Abcam, ab14746, clone 5A5, 1:1,000), anti-ATP5A (Abcam, ab14748, 1:3,000), anti-SDHA (Molecular Probes, 459200, clone 2EGC12FB2AE2, 1:5,000) and anti-NDUFA9 (Molecular Probes, 459100, clone 20C11B11B11, 1:1,000).

## Isolation of IECs

SI tissue was collected from mice, washed in DPBS (14190-094, Gibco) to remove faeces and cut longitudinally. IECs were isolated by sequential incubation of intestinal tissue in pre-heated 1 mM dithiothreitol and 1.5 mM EDTA solutions at 37 °C while shaking. Pellets of IECs were frozen at −80 °C for further processing.

## Protein lysate preparation

IEC pellets were lysed in RIPA lysis buffer (10 mM Tris-Cl (pH 8), 140 mM NaCl, 1 mM EDTA, 0.5 mM EGTA, 1% Triton X-100, 0.1% sodium deoxycholate, 0.1% Triton X-100 and 0.1% SDS). Lysis buffer was supplemented with protease and phosphatase inhibitor tablets (Roche). The protein concentration was measured using Pierce 660 nm Protein Assay reagent (22660, Thermo Scientific) and a BSA standard pre-diluted set ranging from 0 to 2,000 µg ml$^{-1}$ (23208, Thermo Scientific). Cell lysates were separated on SDS–PAGE and transferred to polyvinylidene fluoride membranes (IPVH00010, Millipore). A protein size ladder (26620, Thermo Scientific) was used for size comparison. Membranes were blocked with 5% milk and 0.1% PBST and were probed overnight with primary antibodies against the following antibodies: DARS2 (Proteintech, 13807-1-AP, 1:1,200); total OXPHOS rodent WB antibody

cocktail (Abcam, ab110413, 1:1,000); MTCO1 (Molecular Probes, 459600, clone 1D6E1A8, 1:5,000); COX4L1 (Molecular Probes, A21348, clone 20E8C12, 1:1,000); UQCRC1 (Molecular Probes, 459140, clone 16D10AD9AH5, 1:4,000); NDUFS1 (Proteintech, 12444-1-AP, 1:1,000); NDUFS2 (Abcam, ab96160, 1:1,000); NDUFV2 (Proteintech, 15301-1-AP, 1:1,000); UQCRFS1/RISP[5A5] (Abcam, ab14746, Clone 5A5, 1:1,000); ATP5A (Abcam, ab14748, 1:3,000); SDHA (Molecular Probes, 459200, clone 2EGC12FB2AE2, 1:5,000); NDUFA9 (Molecular Probes, 459100, clone 20C11B11B11, 1:1,000); α-tubulin (Sigma Aldrich, T6074, clone TUBA4A, 1:1,000); TOMM70 (Sigma, HPA014589, 1:500); β-actin (Santa Cruz, sc-1616, clone I-19, 1:1,000); adipophilin/PLIN2 (Progen, GP46, 1:500); FABP2 (Proteintech, 21252-1-AP, 1:500); FASN (Cell Signaling, 3189S, 1:1,000); vinculin (Cell Signaling, 13901, 1:1,000); and ApoB (Beckman Coulter, 467905, 1:500). Membranes were incubated for 1 h at room temperature with anti-rabbit IgG (GE Healthcare, NA934V, 1:5,000), anti-mouse IgG (GE Healthcare, NA931, 1:5,000), anti-goat IgG (Jackson Laboratories, 705-035-003, 1:5,000) or anti-guinea pig IgG (Progen, 90001, 1:5,000) secondary HRP-coupled antibodies and Amersham ECL Western Blotting Detection reagent (GE Healthcare) were used. The membranes were re-probed after incubation in Restore Western Blot stripping buffer (21059, ThermoFisher). The signal was measured with a Curix 60 Processor and a western blot imager (FUSION Solo X, Vilber).

### RNA isolation from tissues
SI tissue samples were disrupted using a Precellys 24 tissue homogenizer (Bertin technologies). Isolation of RNA was performed using a NucleoSpin RNA isolation kit (Macherey Nagel ref. 740955.250) according to the manufacturer's instructions.

### RT–qPCR
cDNA was prepared using a Superscript III cDNA-synthesis kit (18080-044, Thermo Scientific). RT–qPCR was performed using TaqMan probes (Life Technologies) and SYBR Green (Thermo Scientific). The mRNA expression of each gene was normalized to the expression of the housekeeping genes *Tbp* or *Hprt1*. Relative expression of gene transcripts was analysed using the $2^{-\Delta\Delta Ct}$ method. The RT–PCR data were collected using QuantStudio 12K Flex Software v.1.6 (Applied Biosystems). The following Taqman probes were used: *Olfm4* (Mm01320260_m1, Thermo Scientific), *Lgr5* (Mm00438890_m1, Thermo Scientific), *Ascl2* (Mm01268891_g1, Thermo Scientific), *Tbp* (Mm00446973_m1, Thermo Scientific), *Prominin-1* (Mm00477115_m1, Thermo Scientific) and *Lrig-5* (Mm00456116_m1, Thermo Scientific). Primer sequences for SYBR Green are described in Supplementary Table 5.

### *C. elegans* strains, maintenance and imaging
Strains were cultured on OP50 *Escherichia coli*-seeded NGM plates, according to standard protocols[40]. Strains used in this study are Bristol N2, RT1315 *unc-119(ed3); pwIs503*[*pvha-6::mans::gfp;cbr-unc-119*], VS25 hjIs14 [vha-6p::GFP::C34B2.10(SP12) + unc-119(+)] and RT130 pwIs23 [vit-2::GFP]. RNAi knockdown was performed as previously described[41]. All the experiments were performed with hermaphrodite worms at days 1 and 4 of adulthood that were randomly selected and were not allocated into groups. *dars-2*, *sar-1*, *sec-13* and *fum-1* clones were obtained from the Ahringer RNAi library[41] and confirmed by sequencing. As a control, empty L4440 vector was used. For confocal imaging, animals were immobilized on 2% agarose pads in 5 mM levamisole buffer and imaging was performed using a spinning disc confocal microscope (Inverse, Nikon TiE, UltraView VoX, Perkin Elmer, Volocity software). For fluorescence imaging, worms were immobilized on 2% agarose pads in 50 mM sodium azide buffer and imaged using the optical Zeiss Axio Imager Z1 microscope (ZEN 2009 software). Images were analysed using the open-source software Fiji (ImageJ, v.1.53c).

### RNA isolation and RT–qPCR in *C. elegans*
Worms were collected from a 9 cm plate and total RNA was isolated using Trizol (Invitrogen). DNAse treatment was performed using DNA-free, DNAse and removal (Ambion, Life technologies) according to the manufacturer's protocol. RNA was quantified by spectrophotometry and 0.8 µg of total RNA was reverse transcribed using a High-Capacity cDNA Reverse Transcription kit (Applied Biosystems). For each condition, six independent samples were prepared. qPCR was performed using a Step One Plus Real-Time PCR system (Applied Biosystems) with the following PCR conditions: 3 min at 95 °C, followed by 40 cycles of 5 s at 95 °C and 15 s at 60 °C. Amplified products were detected using SYBR Green (Brilliant III Ultra-Fast SYBR Green qPCR Master Mix, Agilent Technologies). Relative quantification was performed against Y45F10D.4.

The following primers were used: *dars-2* FW1 (5′-GTTTGCTGGGG AAATTCAGA-3′); *dars-2* RV1 (5′-AGTGGAGCCGTAAATGGATG-3′); Y45F10D.4 FW (5′-GTCGCTTCAAATCAGTTCAGC-3′); and Y45F10D.4 RV (5′-GTTCTTGTCAAGTGATCCGACA-3′). Data were analysed using ΔΔCt analysis.

### Lipidomics
For lipid analyses, mouse tissue samples were homogenized in deionized water (10 µl per 1 mg wet weight) using a Precellys 24 homogenizer (Peqlab) at 6,500 r.p.m. for 30 s. The protein content of the homogenate was routinely determined using bicinchoninic acid.

**Liquid chromatography coupled to electrospray ionization tandem mass spectrometry.** Sphingolipid (ceramides and sphingomyelins) and cholesterol levels in mouse SI tissue were determined by liquid chromatography coupled to electrospray ionization tandem mass spectrometry (LC–ESI-MS/MS). For sphingolipid analyses, 50 µl of tissue homogenate was used. Lipid extraction and LC–ESI-MS/MS analysis were performed as previously described[42,43]. For the determination of cholesterol levels, 25 µl of tissue homogenate was extracted and processed as previously described[44].

**Nano-ESI-MS/MS.** Levels of cholesteryl esters (CEs), diacylglycerols (DAGs), TAGs and glycerophospholipids in mouse SI tissue were determined by nano-ESI-MS/MS. Next, 10 µl (for DAGs) or 5 µl (for TAGs and CEs) of tissue homogenate was diluted to 500 µl with Milli-Q water and mixed with 1.875 ml of chloroform, methanol and 37% hydrochloric acid 5:10:0.15 (v/v/v). Next, 20 µl of 4 µM d5-TG internal standard mixture I (for TAGs), 15 µl of 256 µM CE 19:0 (for CEs) or 20 µl of 4 µM d5-DG internal standard mixtures I and II (for DAGs) (Avanti Polar Lipids) were added. Lipid extraction and nano-ESI-MS/MS analyses of DAGs and TAGs were performed as previously described[45]. The detection of CE species was conducted in positive-ion mode by scanning for precursors of *m/z* 369 Da at a collision energy of 15 eV and with a declustering potential of 100 V, an entrance potential of 10 V and a cell exit potential of 14 V. Levels of glycerophospholipids (that is, phosphatidylcholines, phosphatidylethanolamines, phosphatidylinositols, phosphatidylserines and phosphatidylglycerols) were determined by performing extraction and nano-ESI-MS/MS measurement of 10 µl of tissue homogenate as previously described[46].

### Metabolomics
**Metabolite extraction.** Metabolite extraction solution (50% methanol, 30% acetonitrile, 20% water and 5 µM valine-d8 as internal standard) was added to 10–20 mg frozen SI tissue samples at an extraction ratio of 25 µl mg$^{-1}$ on dry ice. Samples were then homogenized using a Precellys 24 tissue homogenizer (Bertin Technologies). The resulting sample suspension was vortexed, mixed at 4 °C in a Thermomixer for 15 min at 1,500 r.p.m. and then centrifuged at 16,000*g* for 20 min at 4 °C. The supernatant was collected for LC–MS analysis.

**Metabolite measurement by LC–MS.** LC–MS chromatographic separation of metabolites was achieved using a Millipore Sequant ZIC-pHILIC analytical column (5 μm, 2.1 × 150 mm) equipped with a 2.1 × 20 mm guard column (both 5 mm particle size) with a binary solvent system. Solvent A was 20 mM ammonium carbonate and 0.05% ammonium hydroxide. Solvent B was acetonitrile. The column oven and autosampler tray were held at 40 °C and 4 °C, respectively. The chromatographic gradient was run at a flow rate of 0.200 ml min$^{-1}$ as follows: 0–2 min: 80% solvent B; 2–17 min: linear gradient from 80% solvent B to 20% solvent B; 17–17.1 min: linear gradient from 20% solvent B to 80% solvent B; 17.1–22.5 min: hold at 80% solvent B. Samples were randomized and analysed with LC–MS in a blinded manner with an injection volume of 5 μl. Pooled samples were generated from an equal mixture of all individual samples and analysed interspersed at regular intervals within the sample sequence as a quality control. Metabolites were measured using a Thermo Scientific Q Exactive Hybrid Quadrupole-Orbitrap mass spectrometer (HRMS) coupled to a Dionex Ultimate 3000 UHPLC. The mass spectrometer was operated in full-scan, polarity-switching mode, with the spray voltage set to +4.5 kV/−3.5 kV, the heated capillary held at 320 °C and the auxiliary gas heater held at 280 °C. The sheath gas flow was set to 25 units, the auxiliary gas flow was set to 15 units and the sweep gas flow was set to 0 unit. HRMS data acquisition was performed in a range of $m/z$ = 70–900, with the resolution set at 70,000, the automatic gain control (AGC) target at $1 \times 10^6$ and the maximum injection time at 120 ms. Metabolite identities were confirmed using two parameters: (1) precursor ion $m/z$ was matched within 5 ppm of theoretical mass predicted by the chemical formula; (2) the retention time of metabolites was within 5% of the retention time of a purified standard run with the same chromatographic method.

**Data analysis.** Chromatogram review and peak area integration were performed using the Thermo Fisher software Tracefinder (v.5.0). The peak area for each detected metabolite was subjected to the 'Filtering 80% Rule', half minimum missing value imputation and normalized against the total ion count of that sample to correct any variations introduced from sample handling through instrument analysis. Samples were excluded after performing testing for outliers based on geometric distances of each point in the PCA score plot as part of the muma package (v.1.4)[47]. Afterwards, differential metabolomics analysis was performed. In detail, the R package 'gtools' (v.3.8.2) (cran.r-project.org/web/packages/gtools/index.html) was used to calculate the log$_2$(fold change) using the functions 'foldchange' and 'foldchange2logratio' (parameter base = 2). The corresponding $P$ value was calculated using the R base package 'stats' (v.4.0.5) (www.r-project.org) with the function 't.test' (SIMPLIFY = F). The $P$ value was adjusted using the stats base function 'p.adjust' (method = "bonferroni"). Volcano plots were generated using the EnhancedVolcano package[48] (v.1.8.0).

## QuantSeq 3′ mRNA sequencing

RNA quality was evaluated based on the RNA integrity number (RIN) and OD260/280 and OD260/230 ratios. RIN values were determined using TapeStation4200 and RNA Screen Tapes (Agilent Technologies). Gene expression was determined using a QuantSeq 3′ mRNA-Seq Library Prep kit FWD for Illumina (Lexogen). QuantSeq libraries were sequenced on an Illumina NovaSeq 6000 sequencer using Illumina RTA v.3.4.4 base-calling software. Sample exclusion criteria were OD260/280 < 1.8, OD260/230 < 1.5 and RIN < 7. Illumina adapters were clipped off the raw reads using Cutadapt with standard parameters and a minimum read length of 35 after trimming (shorter reads were discarded). QuantSeq-specific features were deliberately not removed to avoid loss of reads. Trimmed reads were mapped to a concatenation of the mouse genome (Mus_musculus.GRCm38.dna.chromosome.*.fa.gz, downloaded from ftp.ensembl.org/pub/release-100/fasta/mus_musculus/dna/) and the ERCC92 Spike In sequences (downloaded

from assets.thermofisher.com/TFS-Assets/LSG/manuals/ERCC92.zip) using subread-align version v.2.0.1 with parameters -t 0 -d 50 -D 600 --multiMapping -B 5. Genomic matches were counted using featureCounts with parameters -F "GTF" -t "exon" -g "gene_id" --minOverlap 20 -M --primary -O --fraction -J -Q 30 -T 4. The genome annotation used was Mus_musculus.GRCm38.100.gtf (downloaded from ftp.ensembl.org/pub/release-100/gtf/mus_musculus/), augmented by entries for the ERCC92 Spike Ins.

All analyses were done in R-4.0.0, using the functionality of Bioconductor v.3.11. For differential gene expression analysis, the package DESeq2 was used (bioconductor.org/packages/release/bioc/html/DESeq2.html).

Pairwise comparisons were performed between genotypes TG and WT (differential_expression_DESeq2_tg_VS_wt.xlsx). Genes were excluded from a DESeq2 run if they had a zero count in more than half of the samples in either of the conditions compared. Note that DESeq2 sets the $P$ value and the adjusted $P$ value to NA for genes with too few counts or with extreme outlier counts. Such genes were removed after analysis from the DESeq2 output.

The output tables were augmented by gene symbols and descriptions, which were derived from the org.Mm.eg.db annotation package using the function AnnotationDbi::mapIDs (bioconductor.org/packages/release/bioc/html/AnnotationDbi.html). In addition, the raw read counts per gene and sample, as returned by featureCounts, were appended to the rows of each output table. The statistical test producing the $P$ values is the Wald test, and $P$-adjusted values were calculated using the false discovery rate and Benjamini–Hochberg approach. It was computed using the function nbinomWaldTest of the Bioconductor R package DESeq2, based on a negative binomial general linear model of the gene counts from a previously described method[49].

## Proteomics

**In-solution digestion for MS.** Samples for MS analysis were prepared by in-solution digestion. Protein (20 μg) was precipitated for at least 1 h in four volumes (v/v) of ice-cold acetone and protein pellets were extracted by centrifugation at 13,000$g$ for 10 min and dissolved in urea buffer (6 M urea, 2 M thiourea in 10 mM HEPES, pH 8.0). Urea-containing samples were reduced by applying tris(2-carboxyethyl)phosphine at a final concentration of 10 mM, alkylated with chloroacetamide at a final concentration of 40 mM and incubated for 1 h at room temperature. Samples were then digested with 1 μl LysC for 2 h at room temperature, diluted with 50 mM ammonium bicarbonate to a urea concentration of 2 M, incubated with 1 μl 0.5 mg ml$^{-1}$ trypsin overnight at room temperature, acidified to 1% formic acid and purified using Stop and Go extraction tips (StageTips)[50].

**MS-based proteome analysis.** Proteome samples were analysed using LC–MS/MS on an Orbitrap Eclipse Tribrid mass spectrometer (Thermo Fisher) with a FAIMS Pro device using a combination of two compensation voltages of −50 V and −70 V. Chromatographic peptide separation was achieved on 50 cm reverse-phase nanoHPLC-columns (ID 75 μm, PoroShell C18 120, 2.4 μm) coupled to an EASY-nLC 1200 HPLC system and a binary buffer system A (0.1% formic acid) and B (80% acetonitrile/0.1% formic acid). Samples derived from in-solution digestion were measured over a 120 min gradient, raising the content of buffer acetonitrile from 3.2 to 22% over 102 min, from 22 to 45% over 8 min and from 45 to 76% over 2 min. The column was washed with 76% acetonitrile for 8 min. Full MS spectra (300–1,750 $m/z$) were recorded at a resolution of 60,000, maximum injection time of 20 ms and automatic gain control target of $6 \times 10^5$. The 20 most abundant ion peptides in each full MS scan were selected for higher-energy collisional dissociation fragmentation at nominal collisional energy of 30. MS2 spectra were recorded at a resolution of 15,000, a maximum injection time of 22 ms and an automatic gain control target of $1 \times 10^5$. This MS acquisition

program was alternatively run for both FAIMS compensation voltages to cover different peptide fractions.

**MS data processing and analysis.** The generated MS raw data were analysed using MaxQuant analysis software and the implemented Andromeda software (v.1.6.14)[51,52]. Peptides and proteins were identified using the canonical mouse UniProt database (downloaded August 2019) with common contaminants. All parameters in MaxQuant were set to the default values. Trypsin was selected as the digestion enzyme, and a maximum of two missed cleavages was allowed. Methionine oxidation and amino-terminal acetylation were set as variable modifications, and carbamidomethylation of cysteines was chosen as a fixed modification. The label-free quantification algorithm was used to quantify the measured peptides and the 'match between runs' option was enabled to quantify peptides with a missing MS2 spectrum. Subsequent statistical analysis was performed using Perseus (1.5.8.5) software. Potential contaminants and reverse peptides were excluded, and values were $\log_2$ transformed. Raw files were assigned to two groups (TG and WT) and protein groups were filtered for four valid values in at least one group before missing values were replaced from normal distribution (width of 0.3; down shift of 1.3). Welch's Student $t$-test with S0 = 0.1 and a permutation-based false discovery rate of 0.01 with 500 randomizations was performed to obtain differentially regulated proteins between the two groups. Identified proteins were annotated with the gene ontology terms biological process, molecular function, and cellular compartment, and the Reactome Pathway database. Finally, graphical visualization was achieved using Instant Clue software[53] (v.0.5.3).

**GSEA and data visualization.** Gene set enrichment methods were applied using GSEA and over-representation analysis (ORA). In detail, GSEA was performed by using gene sets published on the MsigDB (Reactome, KEGG, Biocarta and Hallmarks)[54] and from a published study[55] (ATF4) using the packages fgsea[56] (v.1.16.0) and GSEABase[57] (v.1.52.1). Volcano plots were generated using the EnhancedVolcano package[48] (v.1.8.0). The ORA was performed using the 'enrich_GO' function (parameters: keyType = "ENTREZID", OrgDb = org.Mm.eg.db, ont = "ALL", pAdjustMethod = "BH", qvalueCutoff = 0.1) of the clusterProfiler package[58] (v.3.16.1). The output data were plotted using the 'emapplot' function of the enrichplot package (v. 1.8.1) (www.bioconductor.org/packages/release/bioc/html/enrichplot.html) (parameters: pie_scale = 1, showCategory = 40, layout = "nicely").

### Metabolic tracer studies

**Postprandial glucose and fat tolerance tests.** Mice were fasted for 2 h before receiving an oral gavage of 300 μl of a glucose–lipid emulsion containing triolein (3.6 g $kg^{-1}$ body weight), lecithin (0.36 g $kg^{-1}$ body weight) and glucose (2 g $kg^{-1}$ body weight), traced with [$^3$H]triolein (1.4 MBq $kg^{-1}$ body weight) and [$^{14}$C]DOG (1.7 MBq $kg^{-1}$ body weight). After 2 h, mice were anaesthetized and transcardially perfused with PBS containing 10 U $ml^{-1}$ heparin. Organs were collected, weighed and dissolved in 10× (v/w) Solvable (Perkin Elmer), and radioactivity (in d.p.m.) was measured by scintillation counting using a Perkin Elmer Tricarb scintillation counter. Uptake of radioactive tracers was calculated per total organ weight.

**CM production.** Mice were injected with tyloxapol (500 mg in 0.9% NaCl per kg body weight) to block vascular lipolysis. Mice received an oral gavage of a lipid emulsion with triolein (3.6 g $kg^{-1}$ body weight) and lecithin (0.36 g $kg^{-1}$ body weight) that were traced with [$^{14}$C]cholesterol (1.4 MBq $kg^{-1}$ body weight) and [$^3$H]triolein (1.7 MBq $kg^{-1}$ body weight). Blood was collected from the tail vein at 0, 30, 60 and 120 min after gavage. Plasma triglycerides were determined by standard colorimetric assays (Roche) and radioactivity was measured by scintillation counting.

### Plasma parameters

Plasma was generated by centrifugation of EDTA-spiked blood for 10 min at 10,000 r.p.m. at 4 °C in a bench top centrifuge. Free glycerol was determined photometrically using Free Glycerol reagent (F6428, Sigma). For lipoprotein profiling, 150 μl pooled plasma was diluted with an equal amount of FPLC buffer (total 300 μl), which was separated by fast-performance liquid chromatography (FPLC) on a Superose 6 10/300 GL column (GE Healthcare) with a flow rate of 0.5 ml $min^{-1}$. Forty fractions (0.5 ml each) were collected, and cholesterol and triglyceride concentrations were measured in each one.

To isolate the TRL fractions, 200 μl of plasma was mixed with 200 μl density solution 1 (0.9% NaCl, 10 mM EDTA, 10 mM Tris-Cl pH 8.6 and 0.49 g $ml^{-1}$ KBr; density 1.3 g $l^{-1}$). The density solution 2 (0.9% NaCl, 10 mM EDTA, 10 mM Tris-Cl pH 8.6; density 1.006 g $l^{-1}$) was placed into a Beckman TL100 centrifuge tube (Beckman, 343778) and then the plasma carefully under layered. Ultracentrifugation was performed in a Beckman Optima MAX-XP ultracentrifuge for 2 h at 4 °C and 40,000 r.p.m. in a Beckman TL100 rotor. After centrifugation, 200 μl of the top containing TRL particles were collected using a syringe.

### Statistical analysis

Data shown in column graphs represent the mean ± s.e.m., as indicated in the figure captions. The D'Agostino–Pearson omnibus normality test was applied to test normal (Gaussian) distribution. When data fulfilled the criteria for normality, unpaired two-sided Student's $t$-tests with no assumption of equal variance were performed; otherwise, the non-parametric Mann–Whitney $U$-test was chosen. Multiple pairwise comparisons of groups over time by repeated measures were evaluated by two-way ANOVA with Bonferroni's correction for multiple comparison (the corrected $P$ values are given for comparison between genotypes at each time point). Survival curves were compared using Gehan–Breslow–Wilcoxon test. The chi-squared test was used for the comparison of the mitochondria integrity distribution between two groups and the assessment of the Golgi pattern distribution after various inhibitor treatments in IEC-6 cells. The number of mice analysed in each experiment is described in the respective figure captions. Statistical analyses were performed with GraphPad Prism 6 (v.6.01) and 9 (v.9.4.1).

### Reporting summary

Further information on research design is available in the Nature Portfolio Reporting Summary linked to this article.

## Data availability

The MS proteomics data have been deposited into the ProteomeXchange Consortium through the PRIDE[59] partner repository with the dataset identifier PXD026934. Metabolomics data have been deposited into Metabolomics Workbench (www.metabolomicsworkbench.org) under the study ID ST002184 (datatrack_id: 3289). The RNA-seq data generated in this study have been deposited into NCBI's Gene Expression Omnibus (GEO) and are accessible through GEO series accession number GSE207803. Numerical source data underlying the graphical representations and statistical descriptions presented in Figs. 1–4 and Extended Data Figs. 1–3,5–9 and 11 are provided as source data files. Uncropped images of immunoblots presented in the figures are included in Supplementary Fig. 1. The number of mice analysed for histological purposes are presented in Supplementary Table 4. Source data are provided with this paper.

## Code availability

Detailed code for GSEA and metabolomics analysis, corresponding data tables and figures can be found at github.com/ChristinaSchmidt1/Mitochondria_in_Intestinal_Lipid-transport.

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

**Acknowledgements** We are grateful to E. Gareus, J. Kuth, E. Stade, C. Uthoff-Hachenberg and J. von Rhein for their technical assistance; A. Schauss and staff at the CECAD Imaging Facility for microscopy support; A. Dilthey and U. Goebel from the CECAD Bioinformatics Facility for RNA-seq data analysis; S. Robine for *Villin-creER^T2* mice, D. Gumucio for *Vil1-cre* mice and C. Moraes for *Cox10^fl/fl* mice; and T. Langer and E. Rugarli for valuable discussions. Research reported in this publication was supported by funding from the European Research Council (grant agreement no. 787826 to M.P. and 819920 to C.F), the Deutsche Forschungsgemeinschaft (DFG, German Research Foundation), projects SFB1403 (project no. 414786233) and TRR259 (project no. 397484323) to M.P., projects SFB1218 (project no. 269925409), GRK2407 (project no. 360043781) to M.P. and A.T., project TR 1018/8-1 to A.T., as well as project SFB841 (Liver inflammation: Infection, immune regulation and consequences) and project SFB1328 (project number 335447717) to J.H. C.F. was also funded by the MRC Core award grant MRC_MC_UU_12022/6, the CRUK Programme Foundation award C51061/A27453 and by the Alexander von Humboldt Foundation in the framework of the Alexander von Humboldt Professorship endowed by the Federal Ministry of Education and Research.

**Author contributions** C.M., A.T. and M.P. conceived the study and designed the experiments. C.M. performed and analysed most experiments. V.K., I.E., F.S., K.S., R.S., S.B., S.E., M. Heine and M.Y.J. performed and analysed experiments. M.Y., E.N. and C.S. performed and analysed the metabolomics experiments. T.B. performed the proteomics experiments. C.S., T.B. and M.K. analysed proteomics data. C.S. performed omics data visualization. M. Herholz. and L.B. performed and analysed the *C. elegans* experiments. V.K., C.F., J.H., A.T. and M.P. supervised the experiments. C.M., V.K., A.T. and M.P. interpreted data and wrote the paper.

**Funding** Open access funding provided by Universität zu Köln.

**Additional information**
**Correspondence and requests for materials** should be addressed to Aleksandra Trifunovic or Manolis Pasparakis.

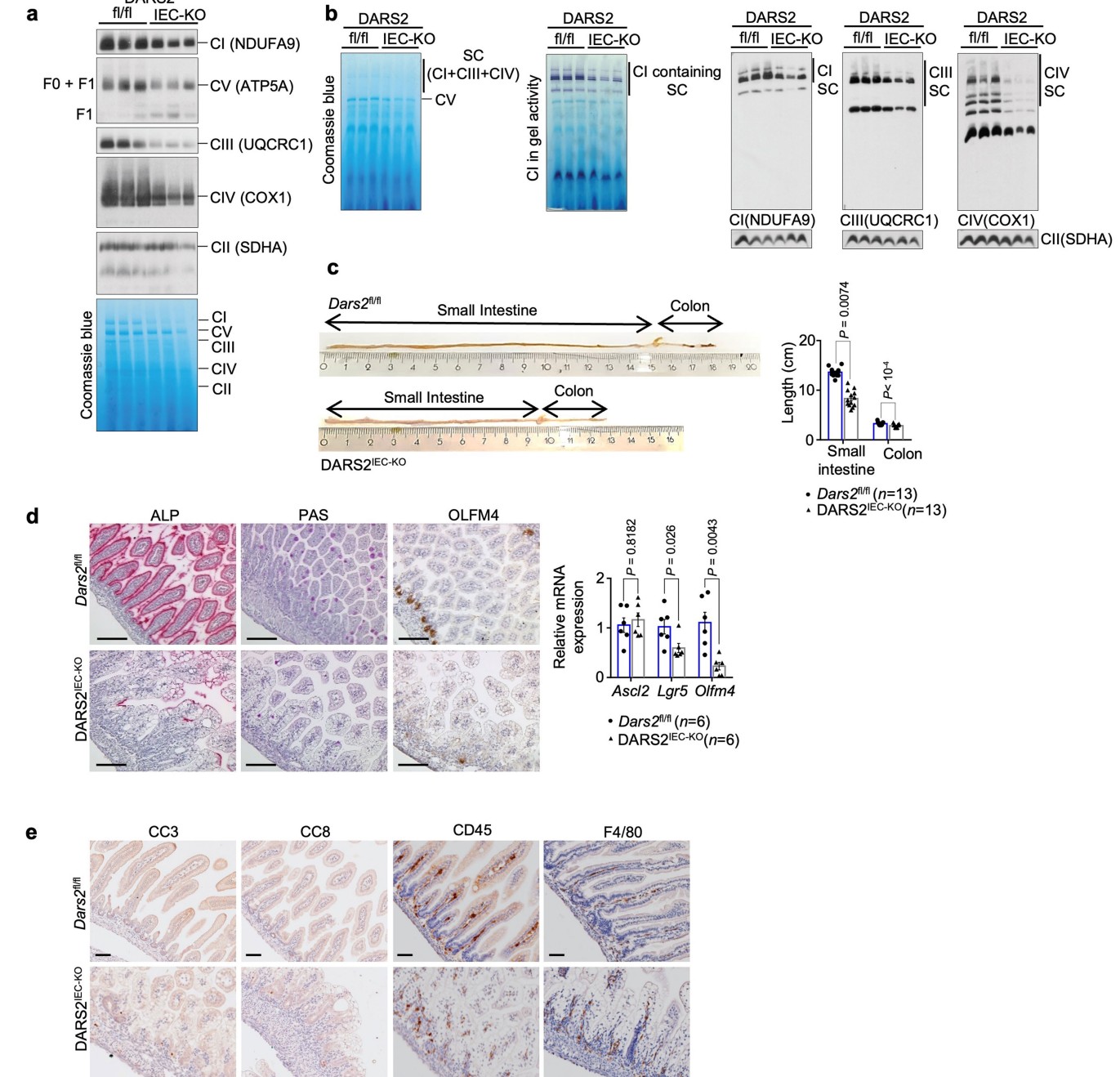

**Extended Data Fig. 1 | Depletion of respiratory complex subunits and intestinal pathology in DARS2[IEC-KO] mice.** BN-PAGE analysis of (**a**) individual respiratory complexes, or (**b**) supercomplexes in mitochondria isolated from the SI of 7-day-old *Dars2*[fl/fl] (fl/fl) and DARS2[IEC-KO] (IEC-KO) mice. Respiratory complexes were visualized by immunoblotting with indicated antibodies (**a, b**). The activity of supercomplex-associated Complex I was determined with *in-gel* assay (**b**). Coomassie blue stains and Complex II levels (anti-SDHA) were used as the loading controls (**a, b**). **c**, Representative pictures and quantification of the length of the SI and colon in 7-day-old DARS2[IEC-KO] (*n* = 13) and *Dars2*[fl/fl] littermates (*n* = 13). **d**, Representative images of SI sections from *Dars2*[fl/fl] and DARS2[IEC-KO] mice stained with PAS and ALP or immunostained against OLFM4 and graph depicting relative mRNA expression of the indicated genes measured by

RT-qPCR in the SI from 7-day-old *Dars2*[fl/fl] (*n* = 6) and DARS2[IEC-KO] (*n* = 6) mice normalized to *Tbp*. **e**, Representative images of SI sections from *Dars2*[fl/fl] and DARS2[IEC-KO] mice immunostained against CC3, CC8, CD45 and F4/80. Scale bars, 50 μm (**d, e**). PAS, Periodic acid- Schiff; ALP, Alkaline Phosphatase; *OLFM4*, Olfactomedin 4, CC3, Cleaved Caspase 3, CC8, Cleaved Caspase 8. In **c** and **d**, dots represent individual mice, bar graphs show mean ± s.e.m. and *P* values were calculated by two-sided nonparametric Mann-Whitney *U*-test. In **a** and **b**, each individual lane represents mitochondria isolated from one mouse (*n* = 3 per genotype). For gel source data, see Supplementary Fig. 1. In **d** and **e**, histological images shown are representative of the number of mice analysed as indicated in Supplementary Table 4.

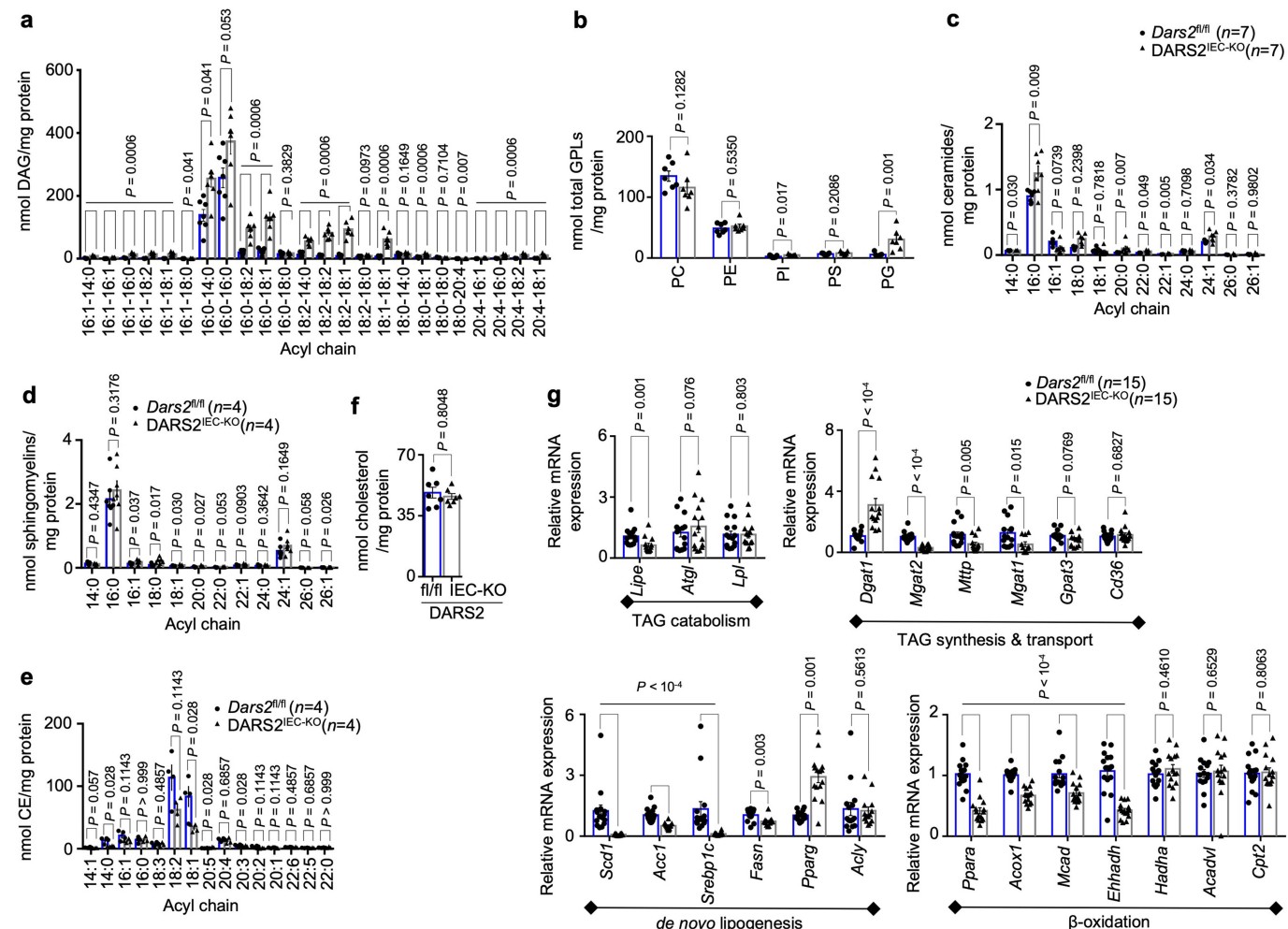

**Extended Data Fig. 2 | Impaired lipid homeostasis in the intestines of DARS2[IEC-KO] mice. a-f**, Graphs depicting quantification of DAG (**a**), GPLs (**b**), ceramides (**c**), sphingomyelins (**d**), CEs (**e**), and total cholesterol levels (**f**) in SI tissues from 7-day-old *Dars2*[fl/fl] and DARS2[IEC-KO] mice (*n* = 7, DAG, GLPs, ceramides, sphingomyelins, total cholesterol and *n* = 4, CE per genotype). **g**, Graph depicting relative mRNA expression of lipid-regulating genes associated with the indicated processes and measured by RT-qPCR in the total distal SI of 7-day-old

*Dars2*[fl/fl] (*n* = 15) and DARS2[IEC-KO] (*n* = 15) mice normalized to *Hprt1*. In all graphs, dots represent individual mice, bar graphs show mean ± s.e.m. and *P* values were calculated by two-sided nonparametric Mann-Whitney *U*-test. DAG, Diacylglycerol, GLPs, Glycerophospholipids, PC, phosphatidylcholine, PE, phosphatidylethanolamine, PI, phosphatidylinositol, PS, phosphatidylserine, PG, phosphatidylglycerol, CE, cholesterol esters.

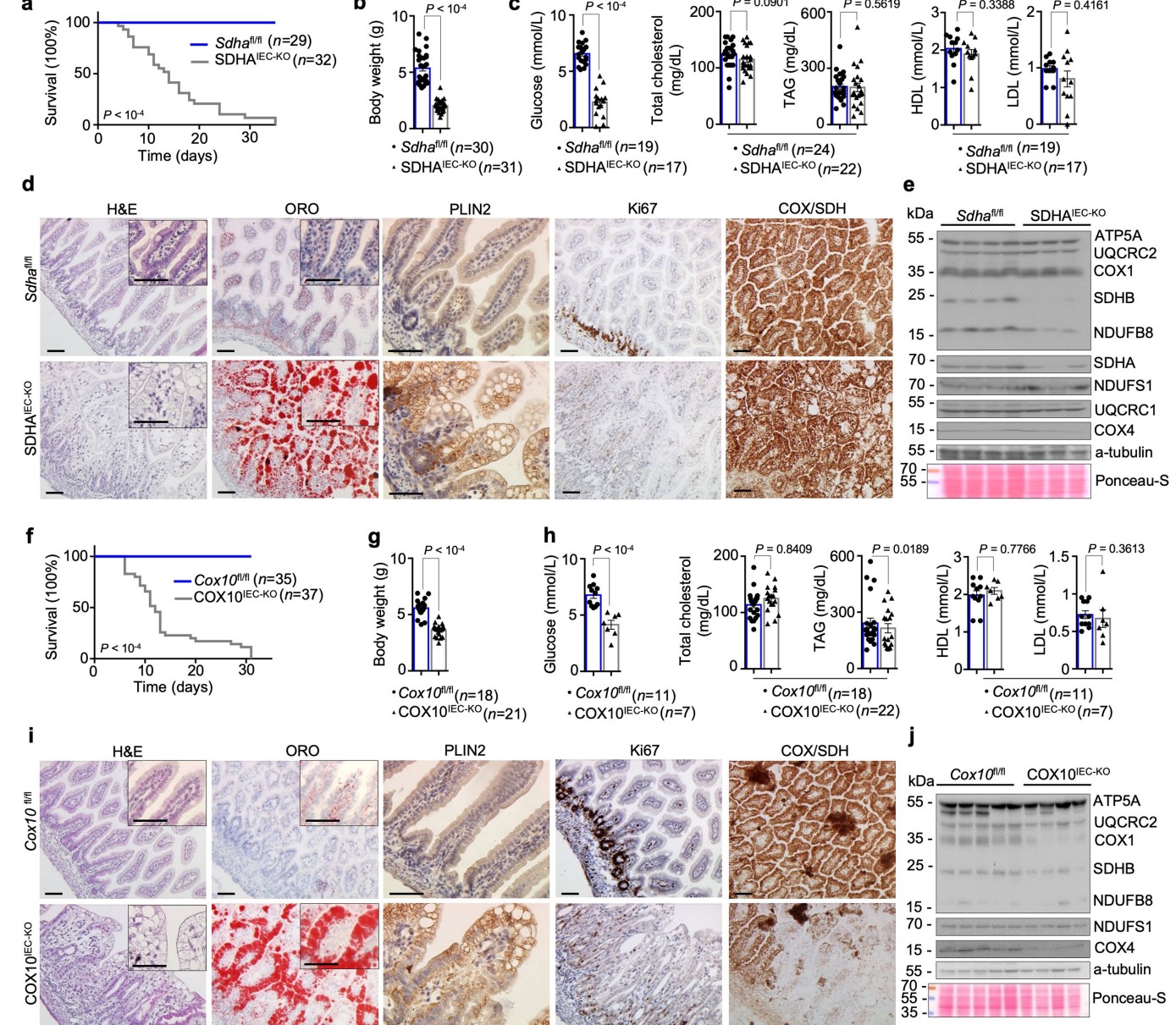

**Extended Data Fig. 3 | IEC-specific ablation of SDHA or COX10 causes lipid accumulation in large LDs in enterocytes. a-c,** Graphs depicting Kaplan-Meier survival curve **(a)**, body weight **(b)** serum levels of glucose, total cholesterol, TAGs, HDL- and LDL-cholesterol **(c)** of *Sdha*^fl/fl^ (*n* = 29 **(a)**, *n* = 30 **(b)**, *n* = 19 (glucose, HDL- and LDL- cholesterol), *n* = 24 (total cholesterol, TAG) **(c)**) and SDHA^IEC-KO^ (*n* = 32 **(a)**, *n* = 31 **(b)**, *n* = 17 (glucose, HDL- and LDL-cholesterol), *n* = 22 (total cholesterol, TAG) **(c)**) mice at the age of 7 days. **d**, Representative images of SI sections from *Sdha*^fl/fl^ and SDHA^IEC-KO^ mice stained with H&E, COX-SDH and ORO or immunostained for PLIN2 and Ki67. Scale bars, 50 μm. **e**, Immunoblot analysis of IEC protein extracts from 7-day-old *Sdha*^fl/fl^ (*n* = 4) and SDHA^IEC-KO^ (*n* = 3) mice with the indicated antibodies. **f, g, h** Graphs depicting Kaplan-Meier survival curve **(f)**, body weight **(g)** serum levels of glucose, total cholesterol, TAGs, HDL- and LDL-cholesterol **(h)** of 7-day-old *Cox10*^fl/fl^ (*n* = 35 **(f)**,

*n* = 18 **(g)**, *n* = 11 (glucose, HDL- and LDL-cholesterol), *n* = 18 (total cholesterol, TAG) **(h)**) and COX10^IEC-KO^ (*n* = 37 **(f)**, *n* = 21 **(g)**, *n* = 7 (glucose, HDL- and LDL-cholesterol), *n* = 22 (total cholesterol, TAG) **(h)**) mice. **i,** Representative images of SI sections from *Cox10*^fl/fl^ and COX10^IEC-KO^ mice stained with H&E, COX-SDH and ORO or immunostained for PLIN2 and Ki67. Scale bars, 50 μm. **j,** Immunoblot analysis of IEC protein extracts from 7-day-old *Cox10*^fl/fl^ (*n* = 5) and COX10^IEC-KO^ (*n* = 4) mice with the indicated antibodies. In **b**, **c**, **g** and **h**, dots represent individual mice, bar graphs show mean ± s.e.m. and *P* values were calculated by two-sided nonparametric Mann-Whitney *U*-test. In **a** and **f**, *P* values were calculated by two-sided Gehan-Breslow-Wilcoxon test. In **d** and **i**, histological images shown are representative of the number of mice analysed as indicated in Supplementary Table 4. In **i** and **j**, each lane represents one mouse and α-tubulin was used as loading control. For gel source data, see Supplementary Fig. 1.

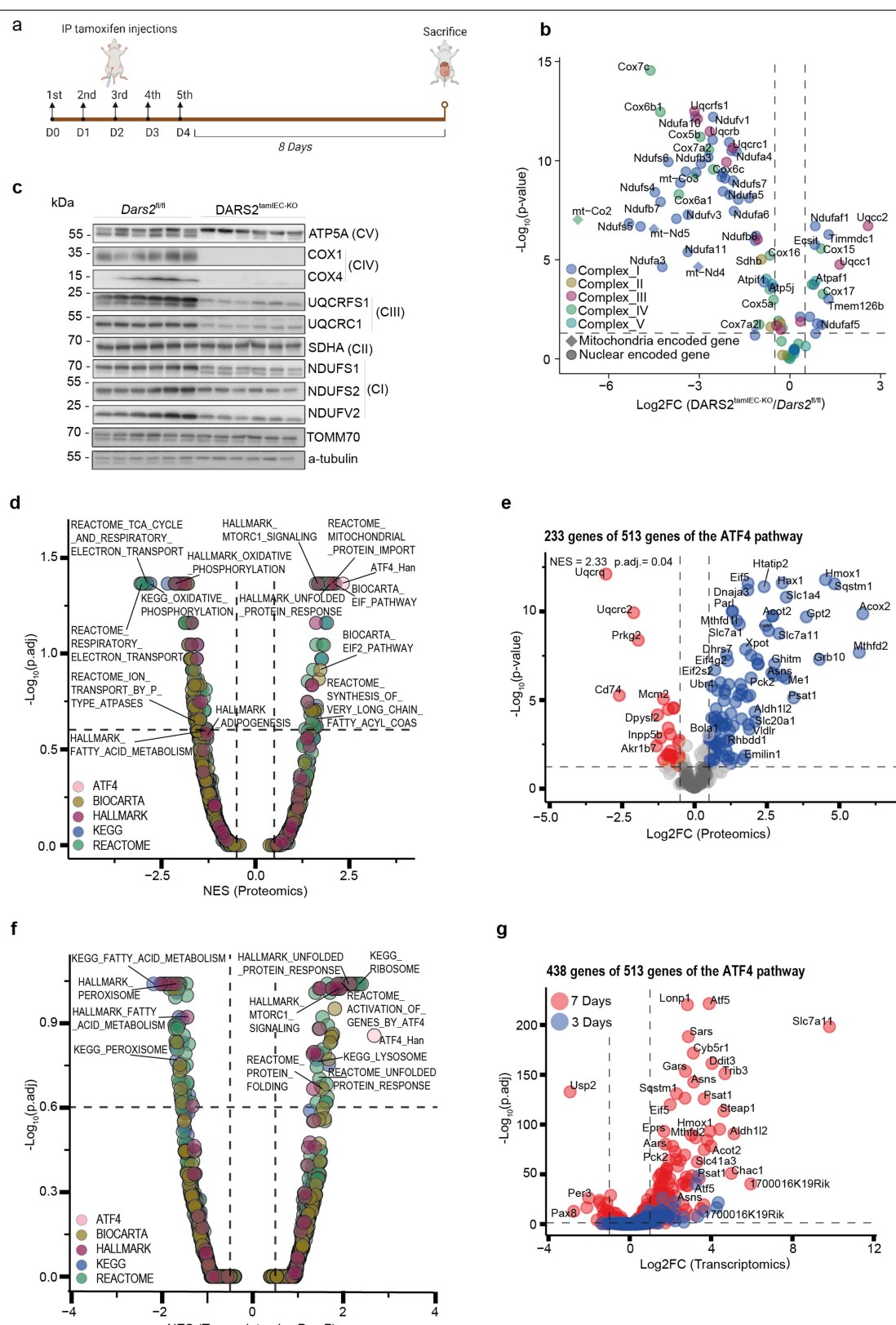

**Extended Data Fig. 4** | See next page for caption.

**Extended Data Fig. 4 | Proteomics and transcriptomics analyses of intestinal tissue and enterocytes from DARS2[tamIEC-KO] mice. a**, Schematic depicting the experimental design for inducible DARS2 deletion created with BioRender.com. Mice received daily intraperitoneal injections of tamoxifen (1 mg) for 5 consecutive days and were sacrificed 8 days upon the last injection as indicated. **b**, Volcano plot illustrating the protein expression profile of the mitochondria respiratory chain complex proteins detected in proximal IECs isolated from DARS2[tamIEC-KO] ($n$ = 11) compared to *Dars2*[fl/fl] ($n$ = 9) 7 days upon the last tamoxifen injection. **c**, Immunoblot analysis of protein extracts from proximal SI IECs from *Dars2*[fl/fl] and DARS2[tamIEC-KO] mice 7 days after the last tamoxifen injection with the indicated antibodies. α-tubulin was used as loading control. **d**, Volcano plot the profile of the different gene sets (colour coded) after performing GSEA analysis on the proteomics landscape of the DARS2[tamIEC-KO] ($n$ = 11) mice compared to *Dars2*[fl/fl] ($n$ = 9) 7 days upon the last tamoxifen injection. Adjusted p-value (p.adj) and normalized enrichment score (NES) are the result of the GSEA analysis. **e**, Volcano plot illustrating the protein expression profile of genes that are part of the ATF4 signature based on Han et al[44] comparing proximal small intestinal IECs from DARS2[tamIEC-KO] ($n$ = 11)

to *Dars2*[fl/fl] mice ($n$ = 9). **f**, Volcano plot illustrating the profile of the different gene sets (colour coded) after performing GSEA analysis on the transcriptomic profile of DARS2[tamIEC-KO] ($n$ = 6) mice compared to *Dars2*[fl/fl] ($n$ = 6) 7 days upon the last tamoxifen injection. NES and p.adj are the result of the GSEA analysis. **g**, Volcano plot illustrating the mRNA expression profile of genes that are part of the ATF4 signature based on Han et. al[44] comparing the proximal small intestine from DARS2[tamIEC-KO] mice to *Dars2*[fl/fl] 7 days (red, $n$ = 6) and 3 days (blue, $n$ = 7) upon the last tamoxifen injection, respectively. In **c**, each lane represents one mouse ($n$ = 6 per genotype). For gel source data, see Supplementary Fig. 1. In **b** and **e**, unpaired two-sided Welch's Student *t*-test with S0 = 0.1 and a permutation-based FDR of 0.01 with 500 randomizations was performed to obtain differentially regulated proteins between the two groups. In **d** and **f**, normalized enrichment score (NES) and the statistics (p.adj) were calculated based on an algorithm described in Subramanian et al[54]. In **g**, the statistical test producing the *P* values is Wald test and P-adjusted values are calculated using the FDR/Benjamini-Hochberg approach. It is computed by function nbinomWaldTest of the Bioconductor DESeq2 package, based on a negative binomial general linear model of the gene counts[49].

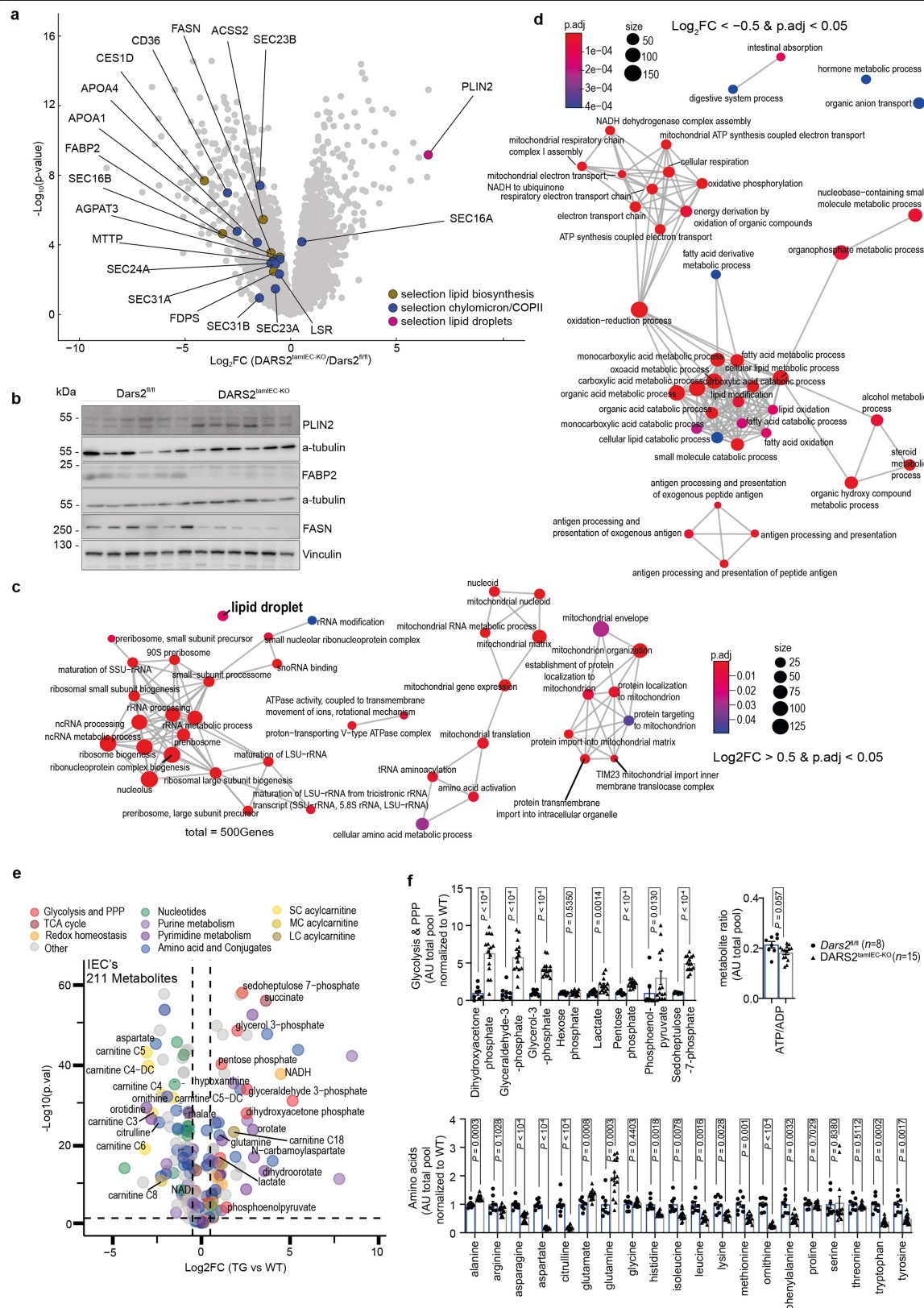

**Extended Data Fig. 5** | See next page for caption.

**Extended Data Fig. 5 | Proteomics and metabolomics analyses reveal downregulation of lipid biosynthesis and chylomicron production, increased lipid droplet formation and suppression of mitochondrial metabolism in DARS2-deficient enterocytes. a**, Volcano plot presenting the proteome landscape comparing DARS2$^{tamIEC-KO}$ ($n = 11$) to $Dars2^{fl/fl}$ ($n = 9$) mice 7 days after the last tamoxifen injection. **b**, Immunoblot analysis with the indicated antibodies of protein extracts from proximal SI IECs isolated from $Dars2^{fl/fl}$ and DARS2$^{tamIEC-KO}$ mice 7 days after the last tamoxifen injection. α-tubulin and vinculin were used as loading controls. **c-d**, Emapplots of the Over Representation Analysis (ORA) performed on the significantly upregulated (Log2FC > 0.5 and p.adj < 0.05) proteins (**c**) and on the significantly downregulated (Log2FC < −0.5 and p.adj<0.05) proteins (**d**) when comparing DARS2$^{tamIEC-KO}$ ($n = 11$) to $Dars2^{fl/fl}$ ($n = 9$) mice 7 days upon tamoxifen injection.

**e**, Volcano plot illustrating the differential intracellular metabolite levels comparing SI IECs of DARS2$^{tamIEC-KO}$ ($n = 15$) to $Dars2^{fl/fl}$ ($n = 9$) mice 7 days upon the last tamoxifen injection. **f**, Each data point in the bar graph represents the mean ± s.e.m of three technical replicates of one animal and is expressed as AUs relative to the average value of all control mouse samples for each metabolite detected in IECs from DARS2$^{tamIEC-KO}$ ($n = 15$) to $Dars2^{fl/fl}$ ($n = 8$) mice 7 days upon the last tamoxifen injection. $P$ values were calculated by unpaired two-sided Welch's Student $t$-test with S0 = 0.1 and a permutation-based FDR of 0.01 with 500 randomizations (**a**), one-sided Fisher's exact test with Benjamini-Hochberg multiple-testing correction (**c**, **d**), one-sided Student $t$-test with Benjamini-Hochberg multiple-testing correction (**e**) and unpaired two-sided Student's $t$-test with no assumption of equal variance or two-sided nonparametric Mann-Whitney $U$-test (**f**).

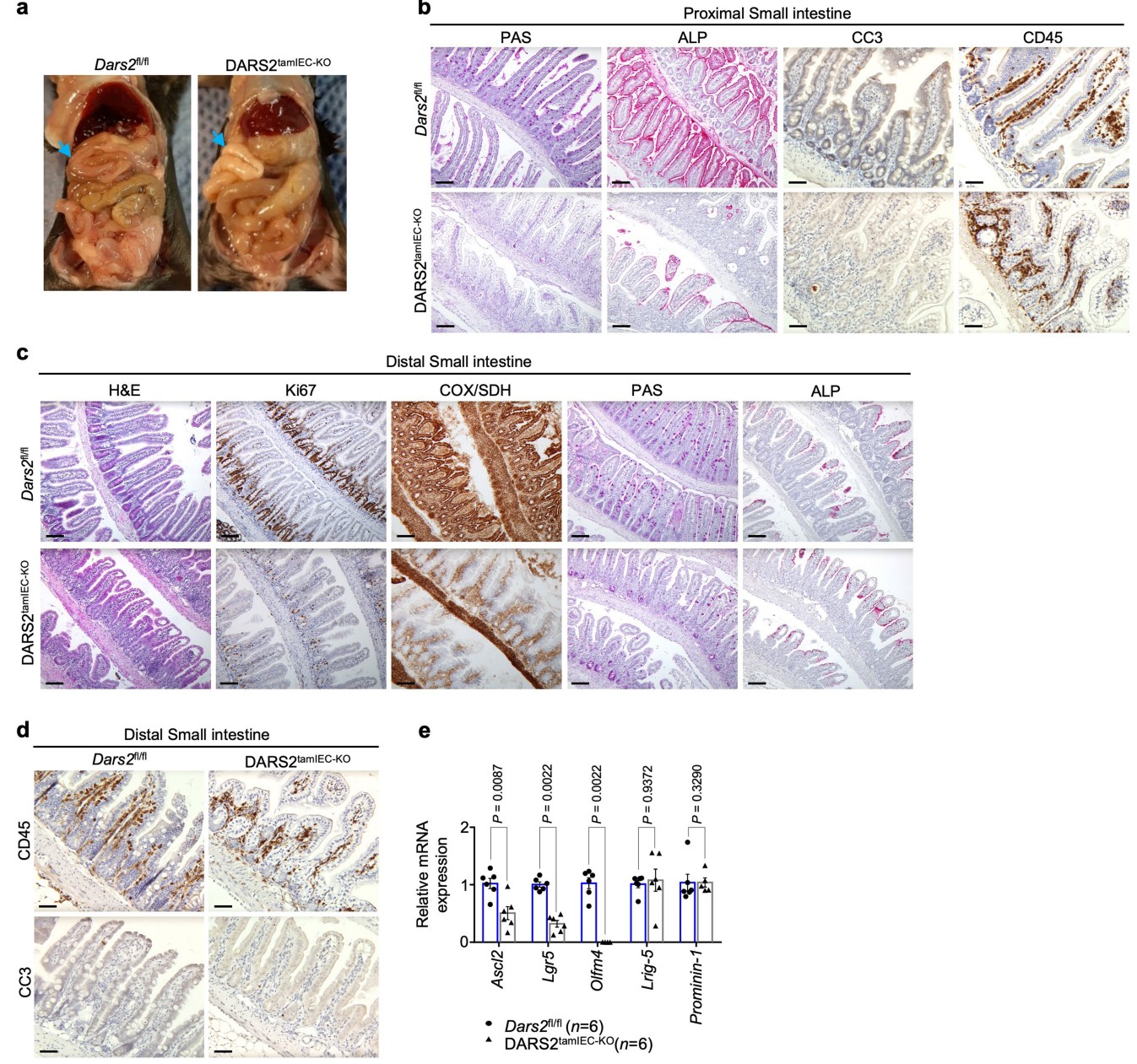

**Extended Data Fig. 6 | Impaired IEC proliferation, stemness and differentiation upon tamoxifen-inducible DARS2 ablation in IECs of adult mice. a**, Representative pictures from necropsy examination of *Dars2*<sup>fl/fl</sup> (*n* = 4) and DARS2<sup>tamIEC-KO</sup> (*n* = 4) mice sacrificed 8 days after the last tamoxifen injection. Blue arrows indicate the proximal SI appearing white in DARS2<sup>tamIEC-KO</sup> mice. **b, c, d** Representative microscopic pictures of proximal SI sections stained with PAS and ALP and immunostained with CC3 and CD45 (**b**) distal SI sections stained with H&E, COX/SDH, PAS, ALP or immunostained with Ki67 (**c**) and distal SI sections immunostained with CC3 and CD45 from *Dars2*<sup>fl/fl</sup> and DARS2<sup>tamIEC-KO</sup> (**d**). Scale bar, 50 µm (**b**, **c**, **d**). **e**, Graph depicting mRNA expression levels of stem cell genes in distal SI of *Dars2*<sup>fl/fl</sup> (*n* = 6) and DARS2<sup>tamIEC-KO</sup> (*n* = 6) mice measured with RT-qPCR and normalized to *Tbp*. In **e**, dots represent individual mice, bar graphs show mean ± s.e.m. and *P* values were calculated by two-sided non-parametric Mann-Whitney U test. In **b**, **c** and **d**, histological images shown are representative of the number of mice analysed as indicated in Supplementary Table 4.

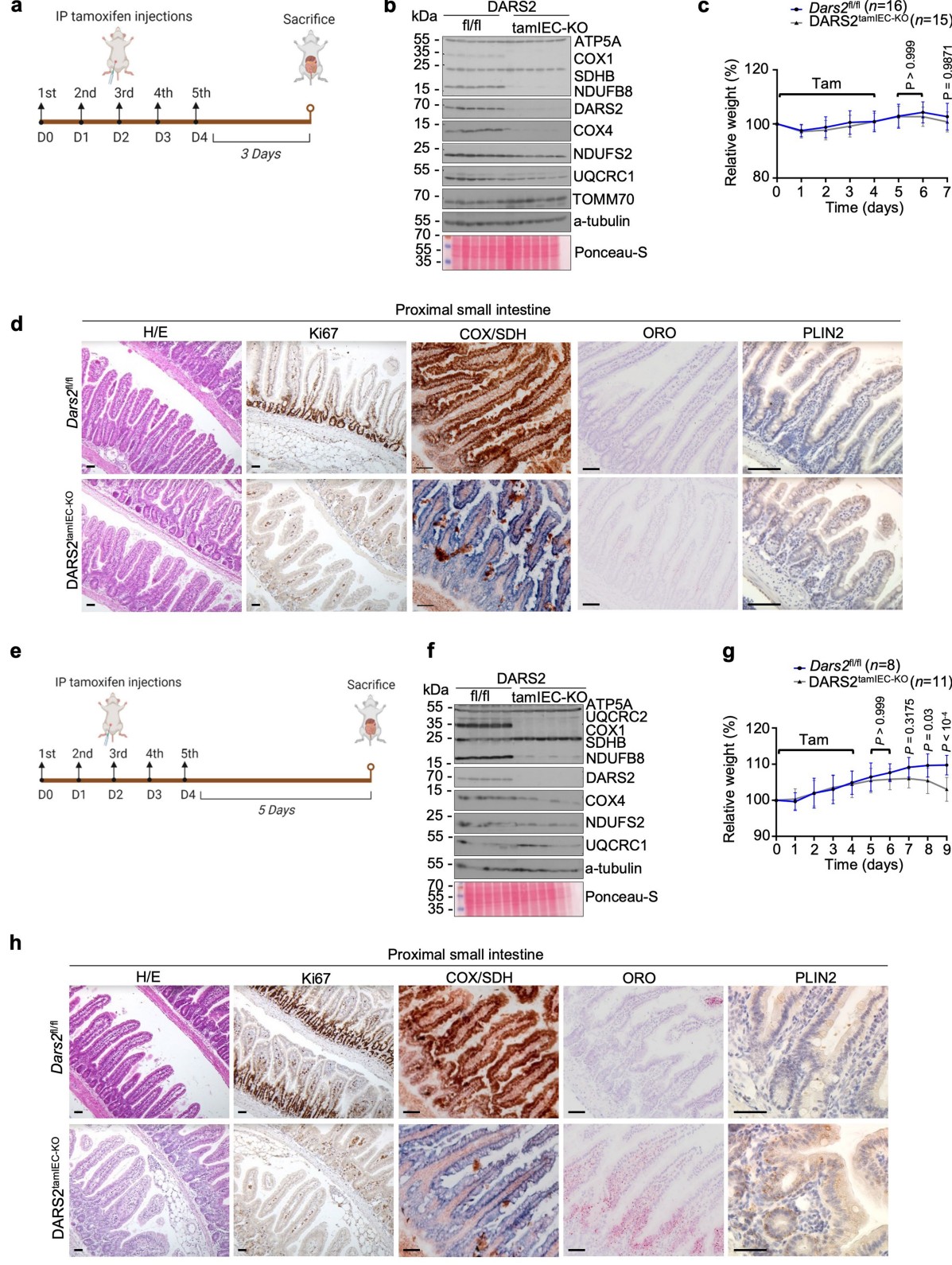

**Extended Data Fig. 7 |** See next page for caption.

**Extended Data Fig. 7 | Analysis of respiratory subunit expression, IEC proliferation and lipid accumulation in DARS2$^{tamIEC-KO}$ mice 3 and 5 days after tamoxifen injection. a**, Schematic depiction of the experimental design created with BioRender.com. **b**, Immunoblot analysis with the indicated antibodies of proximal SI IEC protein extracts from *Dars2*$^{fl/fl}$ ($n = 6$) and DARS2$^{tamIEC-KO}$ ($n = 6$) mice 3 days after the last tamoxifen injection. **c**, Graph depicting relative body weight of *Dars2*$^{fl/fl}$ ($n = 16$) and DARS2$^{tamIEC-KO}$ ($n = 15$) mice after tamoxifen injection. **d**, Representative microscopic images of proximal SI sections from *Dars2*$^{fl/fl}$ and DARS2$^{tamIEC-KO}$ mice sacrificed 3 days upon the last tamoxifen injection stained with H&E, COX/SDH and ORO or immunostained with Ki67 and PLIN2. Scale bars, 50 μm. **e**, Schematic depiction of the experimental design created with BioRender.com. **f**, Immunoblot analysis with the indicated antibodies of proximal SI IEC protein extracts from *Dars2*$^{fl/fl}$ ($n = 5$) and DARS2$^{tamIEC-KO}$ ($n = 6$) mice 5 days after the last tamoxifen injection. **g**, Graph depicting relative body weight of *Dars2*$^{fl/fl}$ ($n = 8$) and DARS2$^{tamIEC-KO}$ ($n = 11$) mice after tamoxifen injection. **h**, Representative microscopic images of proximal SI sections from *Dars2*$^{fl/fl}$ and DARS2$^{tamIEC-KO}$ mice sacrificed 5 days upon the last tamoxifen injection stained with H&E, COX/SDH and ORO or immunostained with Ki67 and PLIN2. Scale bars, 50 μm. In **c** and **g**, data are represented as mean ± s.e.m and *P* values were calculated by two-way ANOVA with Bonferroni's correction for multiple comparison. In **b** and **f**, each individual lane represents one mouse and α-tubulin was used as loading control. For gel source data, see Supplementary Fig. 1. In **d** and **h**, histological images shown are representative of the number of mice analysed as indicated in Supplementary Table 4.

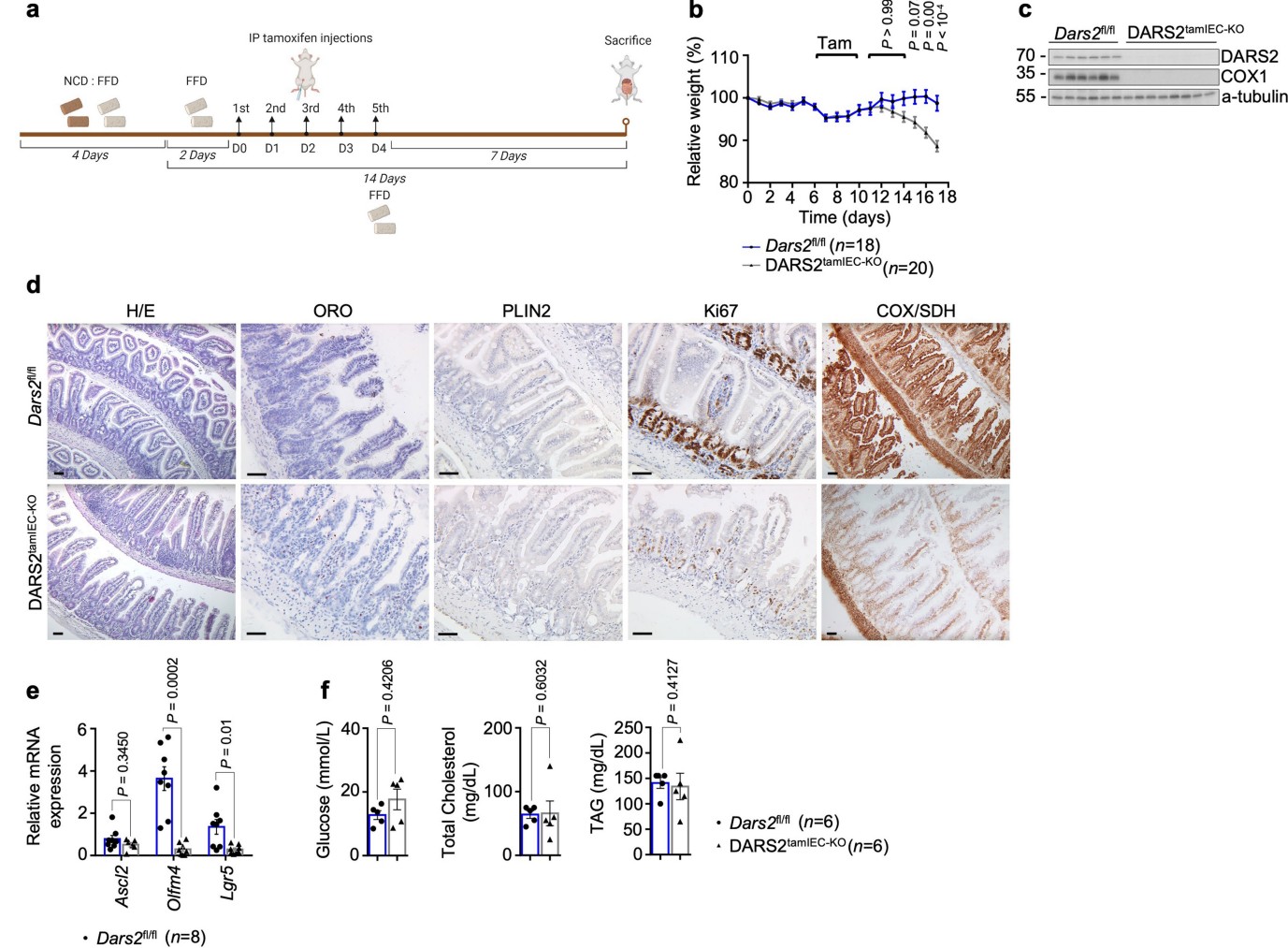

**Extended Data Fig. 8 | Tamoxifen-inducible ablation of DARS2 in adult mice fed with a fat-free diet (FFD). a**, Schematic depiction of the experimental design for inducing DARS2 deletion in mice fed with a FFD created with BioRender.com. 8-12-week-old mice were fed with an equal mixture of normal chow diet (NCD) and FFD for 4 days, which was followed by 14 days FFD feeding and sacrifice 7 days after the last tamoxifen injection. **b**, Graph depicting relative body weight change in FFD-fed *Dars2*^fl/fl ($n$ = 18) and DARS2^tamIEC-KO ($n$ = 20) mice after tamoxifen injection. **c**, Immunoblot analysis with the indicated antibodies of small intestinal IEC protein extracts from FFD-fed *Dars2*^fl/fl ($n$ = 6) and DARS2^tamIEC-KO ($n$ = 8) mice. α-tubulin was used as loading control. **d**, Representative images of sections from the distal SI of FFD-fed *Dars2*^fl/fl and DARS2^tamIEC-KO mice 7 days after the last tamoxifen injection stained with H&E, ORO and COX/SDH or immunostained for PLIN2 and Ki67.

Scale bar, 50 μm. **e**, Graph depicting relative mRNA levels of stem cell genes analysed by RT-qPCR and normalized to *Tbp* in the SI of *Dars2*^fl/fl ($n$ = 8) and DARS2^tamIEC-KO ($n$ = 8) mice 7 days after tamoxifen injection. **f**, Graphs depicting the concentration of glucose, total cholesterol and TAGs in sera from *Dars2*^fl/fl ($n$ = 6) and DARS2^tamIEC-KO ($n$ = 6) mice fed with FFD 7 days upon the last tamoxifen injection. In **e** and **f**, dots represent individual mice. In **b**, **e** and **f** bar graphs data are represented as mean ± s.e.m. and $P$ values were calculated by two-way ANOVA with Bonferroni's correction for multiple comparison (**b**) and two-sided nonparametric Mann-Whitney $U$-test (**e**, **f**). In **c**, each individual lane represents one mouse. For gel source data, see Supplementary Fig. 1. In **d**, histological images shown are representative of the number of mice analysed as indicated in Supplementary Table 4.

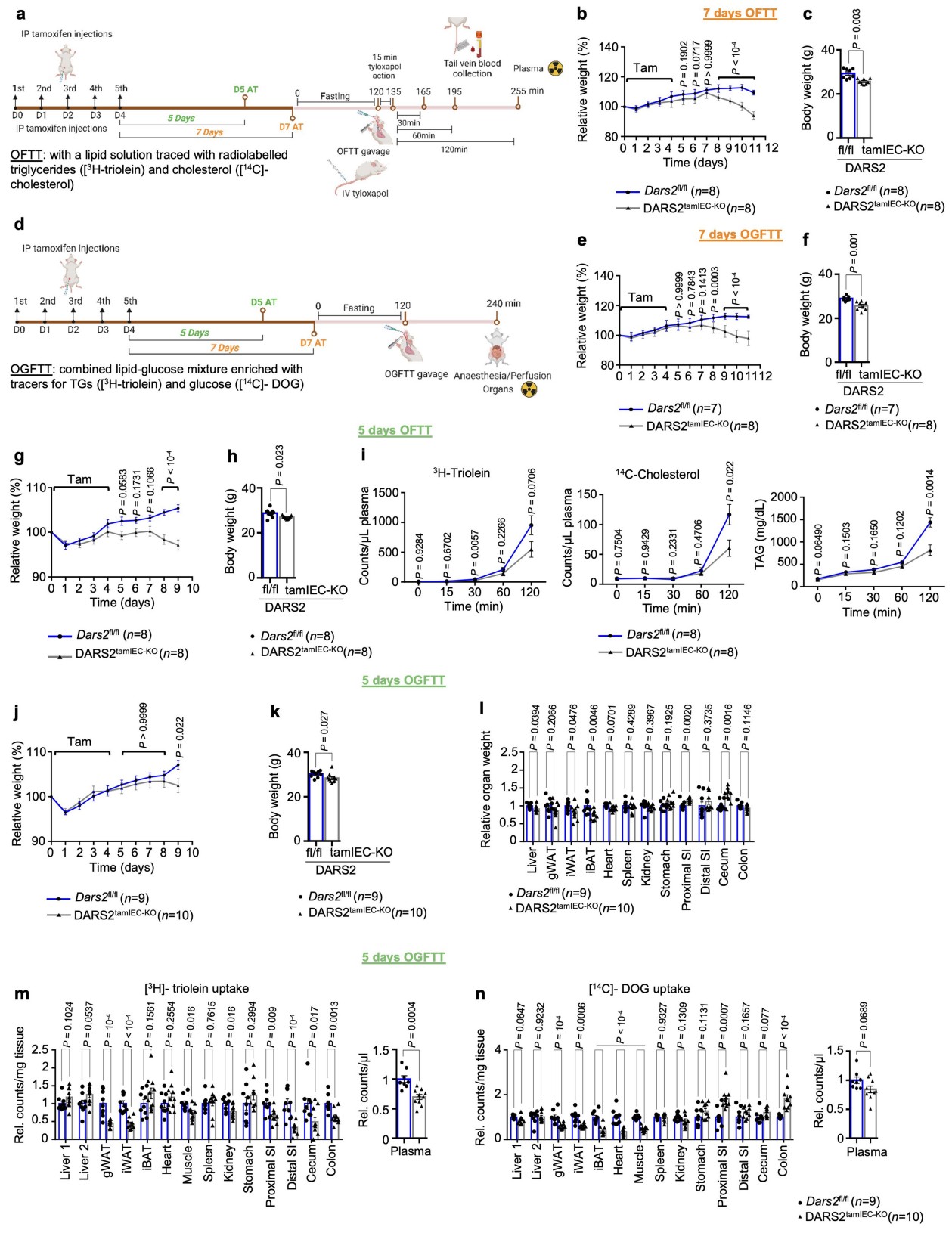

**Extended Data Fig. 9** | See next page for caption.

**Extended Data Fig. 9 | Oral glucose fat tolerance and metabolic tracing in DARS2$^{tamIEC-KO}$ mice 5 and 7 days upon the last tamoxifen injection.**
**a-n**, Metabolic tracing studies performed in DARS2$^{tamIEC-KO}$ mice 7 (**a-f**) and 5 days (**a**, **d**, **g-n**) upon the last tamoxifen injection. **a**, Schematic depiction of the experimental design of the oral fat tolerance test (OFTT) created with Biorender. com. On day 5 or 7 after the last tamoxifen injection, mice received an intravenous injection of the lipoprotein lipase inhibitor tyloxapol and were fasted for 2 h, followed by oral gavage with a lipid solution containing $^3$H-triolein and $^{14}$C-cholesterol. Afterwards, blood was collected from the tail vein for the indicated time points and the plasma appearance of the tracers was measured. **b, c, g, h**, Relative body weight change over the indicated time period (**b**, **g**) and body weight recorded on the day of sacrifice (**c**, **h**) of *Dars2*$^{fl/fl}$ (*n* = 8) and DARS2$^{tamIEC-KO}$ (*n* = 8) mice subjected to OFTT 7 days (**b**, **c**) or 5 days (**g**, **h**) upon tamoxifen. **e, f, j, k**, Relative body weight change over the indicated time period (**e**, **j**) and body weight recorded on the day of sacrifice (**f**, **k**) of *Dars2*$^{fl/fl}$ (*n* = 7) and DARS2$^{tamIEC-KO}$ (*n* = 8) mice subjected to OGFTT 7 days (**e**, **f**) or *Dars2*$^{fl/fl}$ (*n* = 9) and DARS2$^{tamIEC-KO}$ (*n* = 10) mice subjected to OGFTT 5 days (**j**, **k**) upon

tamoxifen. **d**, Schematic depiction of the experimental design of the oral glucose fat tolerance test (OGFTT) created with BioRender.com. On day 5 or 7 after the last tamoxifen injection, mice were fasted for 2 h followed by oral gavage with $^3$H-triolein and $^{14}$C-DOG. Tissues were harvested 2 h after the oral gavage. **i**, Graphs depicting $^3$H-triolein, $^{14}$C-cholesterol and TAG content in portal plasma of *Dars2*$^{fl/fl}$ (*n* = 8) and DARS2$^{tamIEC-KO}$ (*n* = 8) mice subjected to OFTT after intravenous tyloxapol injection. **l**, Graph depicting relative organ weight of *Dars2*$^{fl/fl}$ (*n* = 9) and DARS2$^{tamIEC-KO}$ (*n* = 10) mice subjected to OGFTT. **m, n**, Graphs depicting counts of $^3$H-triolein (**m**) or $^{14}$C-DOG (**n**) in different organs and plasma from *Dars2*$^{fl/fl}$ (*n* = 9) and DARS2$^{tamIEC-KO}$ (*n* = 10) mice determined 120 min after oral gavage 5 days upon the last tamoxifen injection. In **c, f, h, k, l, m** and **n**, dots represent individual mice. In bar graphs data are represented as mean ± s.e.m. and *P* values were calculated by two-way ANOVA with Bonferroni's correction for multiple comparison (**b**, **e**, **g**, **j**), two-sided nonparametric Mann-Whitney *U*-test (**c**, **f**, **h**, **k**), and unpaired two-sided Student's *t*-test with no assumption of equal variance (**i**, **l**, **m**, **n**).

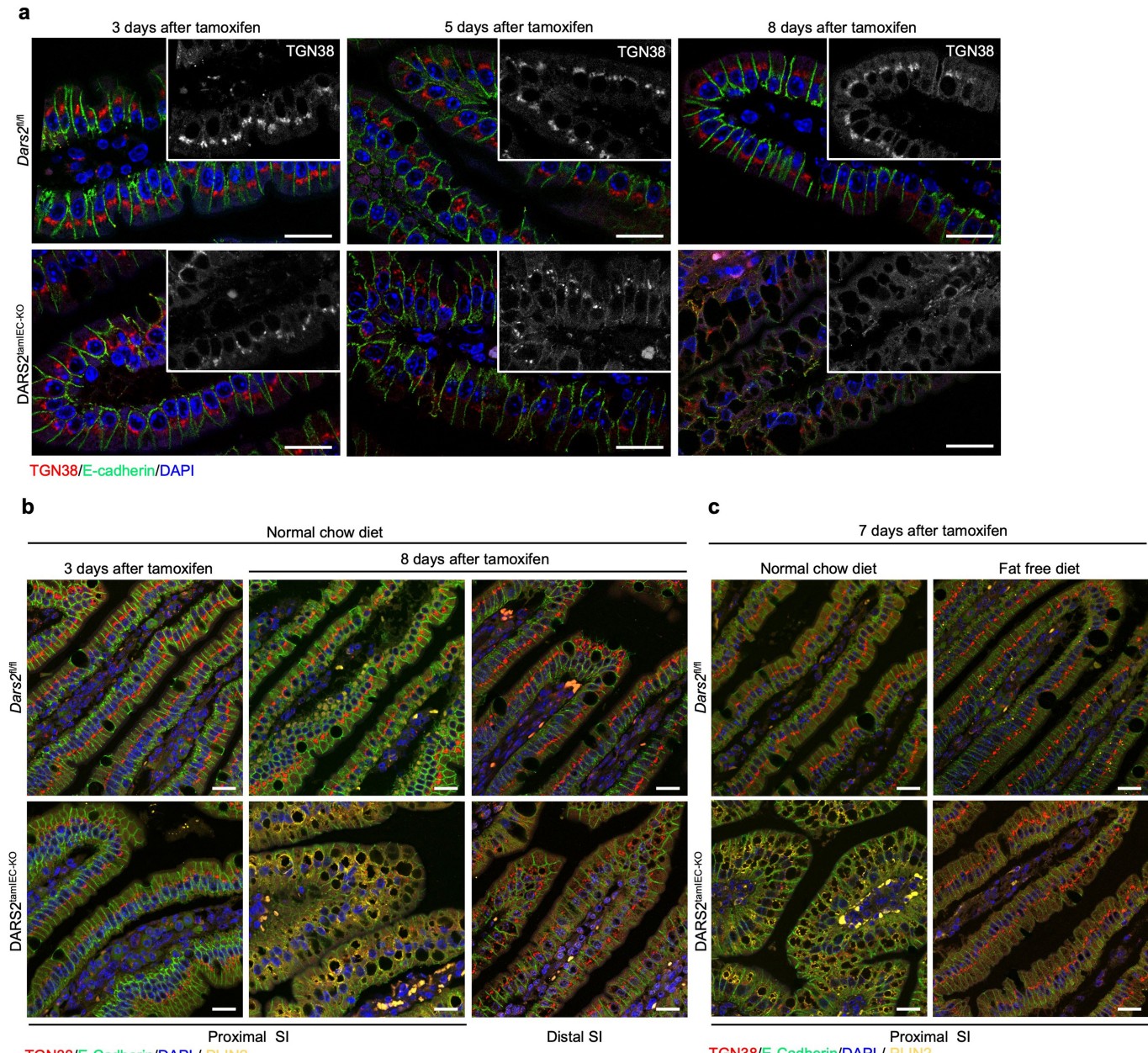

**Extended Data Fig. 10 | DARS2 depletion causes gradual Golgi disorganisation in proximal enterocytes that precedes LD formation and requires the presence of fat in the diet. a**, Representative fluorescence microscopy images from the proximal SI of 8-12-week-old *Dars2*^fl/fl (*n* = 6) and DARS2^tamIEC-KO (*n* = 6) mice sacrificed 3, 5 and 8 days upon the last tamoxifen injection and immunostained with antibodies against TGN38 (red) and E-cadherin (green). Insets shows only TGN38 staining in white. **b**, Representative fluorescence microscopy images from the proximal and distal SI of 8-12-week-old *Dars2*^fl/fl (*n* = 6) and DARS2^tamIEC-KO (*n* = 6) mice fed with NCD sacrificed 3 and 8 days upon the last tamoxifen injection and immunostained with antibodies against TGN38 (red), E-cadherin (green) and PLIN2 (yellow). **c**, Representative fluorescence microscopy images from the proximal SI of 8-12-week-old *Dars2*^fl/fl (*n* = 6) and DARS2^tamIEC-KO (*n* = 6) mice under NCD and FFD sacrificed 7 days upon the last tamoxifen injection and immunostained with antibodies against TGN38 (red), E-cadherin (green) and PLIN2 (yellow). Nuclei stained with DAPI (blue). Scale bars, 50 μm. Confocal images shown are representative of the number of mice analysed as indicated in Supplementary Table 4.

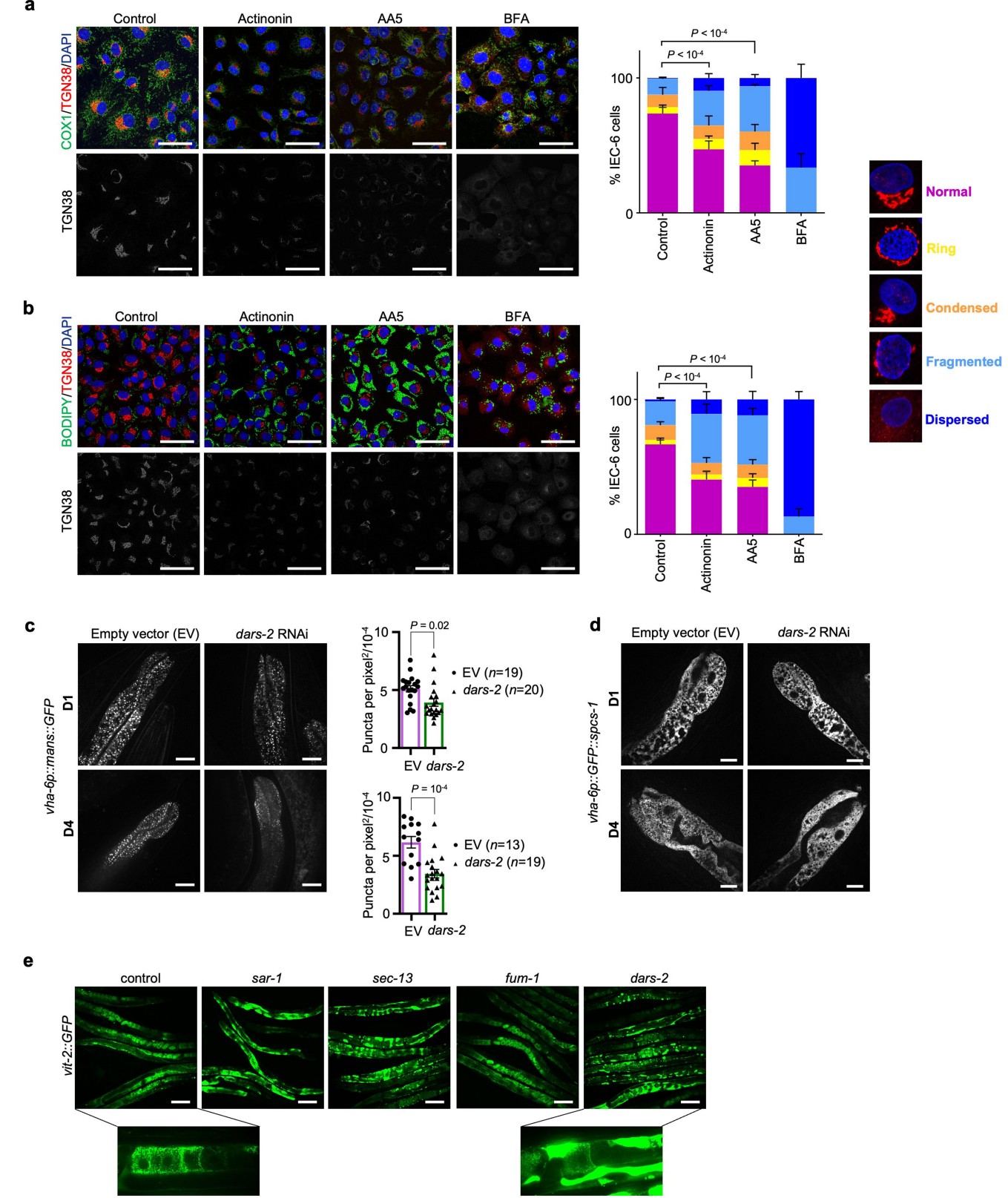

**Extended Data Fig. 11** | See next page for caption.

**Extended Data Fig. 11 | Mitochondrial dysfunction causes impairment of Golgi organisation and lipid processing in IEC-6 cells and in *C. elegans*.**
**a**, Representative fluorescence microscopy images depicting IEC-6 cells treated for 48 h with actinonin (100 μM), Atpenin *A5 (AA5, 1μM)* or 1% dimethyl sulfoxide; DMSO (control). Short treatment (6 h) with Brefeldin A (BFA, 5 μg/ml) was used as a positive control for Golgi dispersal. Scale bars, 50 μm **b**, Representative fluorescence microscopy images of IEC-6 cells grown under the same conditions as described in **a** were incubated with oleic acid (OA, 600 μM) for the last 24 h prior imaging. In this case, BFA was applied in the last 6 h of OA treatment to avoid cytotoxicity. Anti-TGN38 (red) antibody was used to visualize Golgi, Anti-COX1(green) to stain mitochondrial networks **(a)**, BODIPY (green) to stain lipid droplets **(b)**, and DAPI (blue) for nuclei (top). TGN38 staining is additionally depicted in white (bottom). Scale bars, 50 μm. Quantification of the observed Golgi morphology of the IEC-6 cells based on five distinct categories, as illustrated at the right ($n$ = 100–300 inspected IEC-6 cells from three independent biological experiments). **c**, Representative confocal images and graphs depicting quantification of GFP signal in *C. elegans* expressing α-mannosidase II fused to GFP under the control of the gut-specific vha-6 promoter grown either on a control empty vector (EV) or RNAi against *dars-2* at the first (D1) and fourth day of adulthood (D4) (EV, $n$ = 19 (**D1**), $n$ = 13 (**D4**), *dars-2*, $n$ = 20 (**D1**), $n$ = 19 (**D4**)). Scale bars, 1 μm. **d**, Representative confocal images of GFP signal in *C. elegans* expressing SPCS-1 fused to GFP under the control of the gut-specific vha-6 promoter grown either on a control empty vector (EV) or RNAi against *dars-2* at the first (D1) and fourth day of adulthood (D4) EV, $n$ = 19 (**D1**), $n$ = 19 (**D4**), *dars-2*, $n$ = 20 (**D1**), $n$ = 19 (**D4**)). Scale bars, 1 μm. **e**, Immunofluorescence micrographs of *C. elegans* carrying *vit-2::GFP* reporter on a control empty vector (EV) ($n$ = 10) and *RNAi* against *sar-1*, *sec-13*, *fum-1* and *dars-2* at D1 ($n$ = 10). Insets show magnification of a selected area of the worm. Scale bars, 100 μm. In bar graphs data are represented as mean ± s.e.m. and $P$ values were calculated by two-sided Chi-squared test (**a**, **b**) and two-sided Student's *t*-test with assumption of equal variance (**c**).

# Reporting Summary

Nature Research wishes to improve the reproducibility of the work that we publish. This form provides structure for consistency and transparency in reporting. For further information on Nature Research policies, see Authors & Referees and the Editorial Policy Checklist.

## Statistics

For all statistical analyses, confirm that the following items are present in the figure legend, table legend, main text, or Methods section.

| n/a | Confirmed | |
|---|---|---|
| ☐ | ☒ | The exact sample size (*n*) for each experimental group/condition, given as a discrete number and unit of measurement |
| ☐ | ☒ | A statement on whether measurements were taken from distinct samples or whether the same sample was measured repeatedly |
| ☐ | ☒ | The statistical test(s) used AND whether they are one- or two-sided<br>*Only common tests should be described solely by name; describe more complex techniques in the Methods section.* |
| ☒ | ☐ | A description of all covariates tested |
| ☐ | ☒ | A description of any assumptions or corrections, such as tests of normality and adjustment for multiple comparisons |
| ☐ | ☒ | A full description of the statistical parameters including central tendency (e.g. means) or other basic estimates (e.g. regression coefficient) AND variation (e.g. standard deviation) or associated estimates of uncertainty (e.g. confidence intervals) |
| ☐ | ☒ | For null hypothesis testing, the test statistic (e.g. $F$, $t$, $r$) with confidence intervals, effect sizes, degrees of freedom and $P$ value noted<br>*Give P values as exact values whenever suitable.* |
| ☒ | ☐ | For Bayesian analysis, information on the choice of priors and Markov chain Monte Carlo settings |
| ☒ | ☐ | For hierarchical and complex designs, identification of the appropriate level for tests and full reporting of outcomes |
| ☒ | ☐ | Estimates of effect sizes (e.g. Cohen's *d*, Pearson's *r*), indicating how they were calculated |

*Our web collection on statistics for biologists contains articles on many of the points above.*

## Software and code

Policy information about availability of computer code

| | |
|---|---|
| Data collection | Histological images were collected by using a light microscope equipped with a KY-F75U digital camera (JVC) (DM4000B, Leica Microsystems, Diskus 4.50 software).<br>Immunofluorescence images were acquired using a TCS SP8 confocal laser scanning microscope (Inverse, DMi 8 CS, Leica Microsystems LAS X, Lightning software version 5.1.0), a LSM Meta 710 confocal laser scanning microscope (Carl Zeiss Technology, ZEN 2009 software), a Spinning Disc Confocal Microscope (Inverse, Nikon TiE, UltraView VoX, Perkin Elmer, Volocity software) and an optical Zeiss Axio Imager Z1 microscope ( ZEN 2009 software).<br>Electron microscopy images were captured by a transmission electron microscope (JOEL JEM2100 Plus) at an acceleration voltage of 80 kV, using a 4K-CCD camera, OneView (GATAN).<br>The rt-PCR data was collected by QuantStudio 12K Flex Software v1.6 (Applied Biosystems).<br>For RNA sequencing analysis, gene expression was determined using QuantSeq 3' mRNA-Seq Library Prep Kit FWD for Illumina (Lexogen). QuantSeq libraries were sequenced on an Illumina NovaSeq 6000 sequencer using Illumina RTA v3.4.4 base-calling software.<br>Western blot data was collected by Curix 60 Processor and western blot imager (FUSION Solo X, Vilber).<br>Serum data was collected by Biochemical Cobas C111 Analyzer (Roche Diagnostics).<br>Plasma data was collected by fast performance liquid chromatography (FPLC) on a Superose 6 10/300 GL column (GE Healthcare). For the isolation of the TRL fractions, ultracentrifugation was performed using a Beckman Optima MAX-XP in a Beckman TL100 rotor.<br>Metabolic tracing data was collected by Perkin Elmer Tricarb Scintillation Counter.<br>Lipidomics data was collected by using QTRAP 6500 system coupled to a 1260 Infinity Binary LC (Agilent) or to a Nexera X2 UHPLC (Shimadzu) and TriVersa NanoMate (Advion).<br>Metabolomics data was collected by Thermo Scientific Q Exactive Hybrid Quadrupole-Orbitrap Mass spectrometer (HRMS) coupled to a Dionex Ultimate 3000 UHPLC, Orbitrap Eclipse™ Tribrid™ Mass Spectrometer (Thermo Fisher Scientific).<br>Proteomics data was collected by liquid chromatography tandem mass spectrometry on an Orbitrap Eclipse™ Tribrid™ Mass Spectrometer (Thermo Fisher) with FAIMS Pro device. |
| Data analysis | Image processing and analysis was performed using Photoshop (version 22.4.2 and 23.5.1) and the open-source software FIJI/ImageJ (version 1.53c and 2.0.0.-rc-46/1.50g). |

For analysis of the RNA sequencing data, reads were mapped to a mouse reference genome Mus_musculus.GRCm38.100.gtf which was downloaded from (ftp://ftp.ensembl.org/pub/release-100/gtf/mus_musculus/), augmented by entries for the ERCC92 Spike Ins downloaded from (https://assets.thermofisher.com/TFS-Assets/LSG/manuals/ERCC92.zip) and using subread-align (v2.0.1). Illumina adapters were clipped off the raw reads using Cutadapt. All analyses were done in R-4.0.0, using functionality of Bioconductor (v.3.11). Western blot data produced by FUSION Solo X (Vilber) were visualized using GIMP (v.2.10.20).

Proteomics data analysis was performed by using MaxQuant analysis software and the implemented Andromeda software (1.6.14). Peptides and proteins were identified using the canonical mouse UniProt database (downloaded 08/2019) and statistical analysis was performed using Perseus (1.5.8.5) software. Identified proteins were annotated with the following Gene Ontology terms: Biological Process, Molecular Function, and Cellular Compartment, and the Reactome Pathway database.

Analysis of the metabolomics data was performed by using the Thermo Fisher software Tracefinder (v.5.0), the muma package (v.1.4), the R package „gtools"(v.3.8.2) (https://cran.r-project.org/web/packages/gtools/index.html), and the R base package stats (v.4.0.5) (https://www.r-project.org/). Lipidomics data was analyzed by TraceFinder, Compound Discoverer (Thermo Scientific), MultiQuant, LipidView, MarkerView, PeakView, MasterView (SCIEX).

Gene set enrichment analysis (GSEA) was performed by using gene sets published on the MsigDB (Reactome, KEGG, Biocarta and Hallmarks) (Subramanian, A. et al., 2005) and by (Han, J. et al., 2013) using the packages fgsea (v.1.16.0) (Sergushichev, A. A. et al., 2016) and GSEABase (v.1.52.1) (Morgan, S. et al., 2021). The over representation analysis (ORA) was performed using the clusterProfiler package 22 (v.3.16.1). Volcano plots were generated using the EnhancedVolcano package (v.1.8.0). Graphical visualization was additionally achieved using Instant Clue software (v.0.5.3) (Nolte, H., MacVicar, T. D., et al., 2018).

The output data are plotted using the "emapplot" function of the enrichplot package (v.1.8.1) (https://www.bioconductor.org/packages/release/bioc/html/enrichplot.html).

Detailed code, data tables and figures are available under:
https://github.com/ChristinaSchmidt1/Mitochondria_in_Intestinal_Lipid-transport.

Data compiling and processing was performed using Microsoft Excel 2021 (v.2108) and Adobe Illustrator (v.26.5).

Statistical analysis was performed with GraphPad Prism 6 (v.6.01) and 9 (v.9.4.1).

Schematics were created with BioRender.com.

For manuscripts utilizing custom algorithms or software that are central to the research but not yet described in published literature, software must be made available to editors/reviewers. We strongly encourage code deposition in a community repository (e.g. GitHub). See the Nature Research guidelines for submitting code & software for further information.

# Data

Policy information about availability of data

All manuscripts must include a data availability statement. This statement should provide the following information, where applicable:
- Accession codes, unique identifiers, or web links for publicly available datasets
- A list of figures that have associated raw data
- A description of any restrictions on data availability

The original RNA sequencing data are uploaded and available in NCBI's Gene Expression Omnibus and are accessible through GEO Series accession number GSE207803 (https://www.ncbi.nlm.nih.gov/geo/query/acc.cgi?acc=GSE207803).
The original proteomic data are uploaded and available at the DATABASE under accession PXD026934. The mass spectrometry proteomics data have been deposited to the ProteomeXchange Consortium via the PRIDE partner repository with the dataset identifier PXD026934.
The original metabolomics data haven been deposited at Metabolomics Workbench (https://www.metabolomicsworkbench.org) under the study_id ST002184 (datatrack_id:3289, http://dx.doi.org/10.21228/M8X99S).
Numerical source data giving rise to graphical representations and statistical descriptions presented in the Figs. 1–4 and Extended Data Figs. 1, 2, 3, 5, 6, 7, 8, 9 and 11 are provided as Source Data files with the paper.
Uncropped images of immunoblots presented in the figures are included in Supplementary Figure 1.

# Field-specific reporting

Please select the one below that is the best fit for your research. If you are not sure, read the appropriate sections before making your selection.

☒ Life sciences  ☐ Behavioural & social sciences  ☐ Ecological, evolutionary & environmental sciences

For a reference copy of the document with all sections, see nature.com/documents/nr-reporting-summary-flat.pdf

# Life sciences study design

All studies must disclose on these points even when the disclosure is negative.

| | |
|---|---|
| Sample size | Sample size was determined empirically and was based on our previous mouse work. We aimed for a number of at least 3 animals per group to allow basic statistical analysis while using a justifiable number of mutant mice. All sample sizes were annotated within the respective Figure legends. Based on previous experience from similar studies, in vitro experiments with cultured cells were performed at least 3 times (3 biological replicates) to confirm reproducibility. Each biological replicate is defined as an independent culture of cells. |
| Data exclusions | No animal data was excluded from the analyses.<br>Sample exclusion criteria (OD260/280 <1.8, OD260/230 <1.5 and RIN<7) were applied for QuantSeq 3' mRNA sequencing analysis. Genes were excluded from a DESeq2 run if they had a zero count.<br>For metabolomic analysis, the peak area for each detected metabolite was subjected to the "Filtering 80% Rule", half minimum missing value imputation, and normalized against the total ion count (TIC) of that sample to correct any variations introduced from sample handling through instrument analysis. Samples were excluded after performing testing for outliers based on geometric distances of each point in the PCA score |

plot. For proteomic analysis, potential contaminants and reverse peptides were excluded.

| | |
|---|---|
| Replication | For in vivo studies we analyzed at least 6 mice per genotype to ensure the reproducibility of the results. For in vitro studies we independently replicated all experiments at least 3 times.<br>For C. elegans studies, at least 2 independent technical replicates were used. All attempts of replication were successful and results were reliably reproduced with the same trend. |
| Randomization | No specific method of randomization had been used to select animals. We compared groups of mice with different genotypes to assess the effect of specific genetic mutations in the phenotype. Group allocation was thus determined by the genotype of the mice.<br>For C. elegans experiments, worms were randomly selected and were not allocated into groups. |
| Blinding | No blinding was done during the group generation as the group allocation was determined by the genotype of the mice.<br>For 1-week-old mice, sample collection was not performed in a blinded manner as they exhibited an obvious macroscopic phenotype that revealed the sample identity.<br>Histological evaluation of intestinal sections was performed blindly.<br>Golgi quantification was performed by manually classifying the TGN38 pattern in each cell in one of the five Golgi phenotypes by the same observer, who was blind to the experimental conditions.<br>Metabolomic samples were analysed with LC–MS in a blinded manner. |

# Reporting for specific materials, systems and methods

We require information from authors about some types of materials, experimental systems and methods used in many studies. Here, indicate whether each material, system or method listed is relevant to your study. If you are not sure if a list item applies to your research, read the appropriate section before selecting a response.

## Materials & experimental systems

| n/a | Involved in the study |
|---|---|
| ☐ | ☒ Antibodies |
| ☐ | ☒ Eukaryotic cell lines |
| ☒ | ☐ Palaeontology |
| ☐ | ☒ Animals and other organisms |
| ☒ | ☐ Human research participants |
| ☒ | ☐ Clinical data |

## Methods

| n/a | Involved in the study |
|---|---|
| ☒ | ☐ ChIP-seq |
| ☒ | ☐ Flow cytometry |
| ☒ | ☐ MRI-based neuroimaging |

## Antibodies

Antibodies used

Primary antibodies:

1) Monoclonal rabbit anti-Olfm4 (D6Y5A) XP®, Cell Signaling, Cat. No. 39141, Clone name: D6X5A, Lot. No. 1
2) Polyclonal guinea-pig anti-adipophilin / perilipin 2 (N-terminus aa 1-16), Progen, Cat. No. GP46, Lot No. 05/09(0270)NT-02
3) Monoclonal rat anti-Ki67, DAKO, Cat. No. M724901, Clone name: 1O15, Lot No. TEC-3
4) Rabbit polyclonal anti-cleaved Caspase 3, Cell Signaling Technology, Cat. No. 9661, Lot. No. 433
5) Rabbit monoclonal anti-cleaved Caspase 8 (Asp387), Cell Signaling Technology, Cat. No. 8592, Clone name: D5B2, Lot. No. 3
6) Monoclonal rat anti-F4/80, AbD Serotec, Cat. No. MCA497, Clone name: A3-1
7) Monoclonal rat anti-CD45, BD Biosciences, Cat. No. 560510, Clone name: 30-F11, Lot. No. E03735-1631
8) Polyclonal sheep anti-TGN38, bio-techne, Cat. No. AF8059-SP, Lot. No CHWO012104A
9) Purified mouse anti-E-cadherin, BD Biosciences, Cat. No. 610182, Clone name: 36/ E-Cadherin (RUO), Lot. No 9315423
10) Polyclonal rabbit anti-DARS2, Proteintech, Cat. No. 13807-1-AP, Lot. No 00019410
11) Total OXPHOS Rodent WB Antibody Cocktail, Abcam, Cat. No. ab110413,  Lot. No K2342
12) Polyclonal goat anti-β-actin, Santa Cruz, Cat. No. sc-1616, Clone number: I-19, Lot. No B1116
13) Mouse monoclonal anti-MTCO1/COX1, Molecular Probes, Cat. No 459600, Clone name: 1D6E1A8, Part/Lot. No. 459600/ G1010
14) Mouse monoclonal anti-COX4L1, Molecular Probes, Cat. No A21348, Clone name: 20E8C12, Lot. No. 1801178
15) Mouse monoclonal anti-UQCRC1, Molecular Probes, Cat. No 459140, Clone name: 16D10AD9AH5, Part/Lot. No. 459140/ IC3878
16) Rabbit polyclonal anti-NDUFS1, Proteintech, Cat. No 12444-1-AP, Lot. No. 00007233
17) Rabbit polyclonal anti-NDUFS2, Abcam, Cat. No ab96160, Lot. No. GR3179687-4
18) Rabbit polyclonal anti-NDUFV2, Proteintech, Cat. No 15301-1-AP, Lot. No. 00006404
19) Mouse monoclonal anti-UQCRFS1/RISP, Abcam, Cat. No ab14746, Clone name: 5A5, Lot. No. GR304562-8
20) Mouse monoclonal anti-ATP5A, Abcam, Cat. No ab14748, Clone name: 15H4C4, Lot. No. GR3306993-24
21) Mouse monoclonal anti-SDHA, Molecular Probes, Cat. No 459200, Clone name: 2EGC12FB2AE2, Lot. No. UJ2872542A
22) Mouse monoclonal anti-NDUFA9, Molecular Probes, Cat. No 459100, Clone name: 20C11B11B11, Lot. No. GR3268847-10
23) Mouse monoclonal anti-tubulin, Sigma Aldrich, Cat. No T6074, Clone name: TUBA4A, Lot. No. 046M4763V
24) Rabbit polyclonal anti-TOMM70, Sigma Aldrich, Cat. No. HPA014589, Lot. No. 000010432
25) Rabbit polyclonal anti-FABP2, Proteintech, Cat. No. 21252-1-AP, Lot. No. 00014123
26) Rabbit polyclonal anti-FASN, Cell Signaling, Cat. No. 3189S, Lot. No. 2
27) Rabbit monoclonal anti-Vinculin XP, Cell Signaling Technology, Cat. No. 13901, Clone name: E1E9V, Lot. No. 6
28) Goat Antibody Monospecific for Human Apolipoprotein B, Beckmann Coulter, Cat. No. 467905

Secondary antibodies:

1) Donkey anti-rabbit IgG conjugated to HRP antibody, GE Healthcare, Cat. No. NA934V, Lot. No 17203153
2) Sheep anti-mouse IgG conjugated to HPR antibody, GE Heathcare, Cat. No. NA931, Lot. No 17317435
3) Donkey anti-goat IgG, conjugated to HPR antibody, Jackson Laboratories, Cat. No.705-035-003, Lot. No 17320421
4) Goat anti-guinea Pig IgG conjugated to HPR Antibody, Progen, Cat. No 90001, Lot. No.908211
5) Goat anti-mouse IgG antibody (H+L) Biotinylated, Vector Laboratories, Cat. No. BA-9200, Lot. No. X0623
6) Goat anti-rabbit IgG antibody (H+L) Biotinylated, Vector laboratories, Cat. No. BA-1000, Lot. No. 10180764
7) Goat anti-rat IgG antibody (H+L) Biotinylated, Vector Laboratories, Cat. No. BA-9400, Lot. No. 112632
8) Polyclonal donkey anti-sheep IgG NorthernLights™ NL557-conjugated Antibody, bio-techne, Cat. No. NL010, Lot. No. AADR0713071
9) Polyclonal goat anti-mouse Alexa 488 - fluorescence-conjugated secondary Antibody, Molecular Probes, Cat. No. A1101
10) Polyclonal goat anti-guinea pig Alexa 633 - fluorescence-conjugated secondary Antibody, Molecular Probes, Cat. No. A21105

**Validation**

The antibodies used in this study were tested by the manufacturer.

Primary antibodies:

1) anti-Olfm4 (D6Y5A) XP® (Cell Signaling, 39141, clone D6X5A) can be found in 83 citations. The manufacturer also provides antibody testing data: https://www.cellsignal.com/products/primary-antibodies/olfm4-d6y5a-xp-rabbit-mab/39141

2) anti- adipophilin/perilipin 2 (N-terminus aa 1-16)(Progen, GP46) can be found in 3 citations. The manufacturer also provides antibody testing data : https://www.progen.com/anti-Perilipin-2-N-terminus-aa-1-16-guinea-pig-polyclonal-serum/GP46

3) anti-Ki67 (DAKO, M724901,1O15) can be found in 2368 citations (https://www.citeab.com/antibodies/2390690-m7240-ki-67-antigen-concentrate)

4) anti-cleaved-caspase3 (Cell Signaling, 9661, Asp175) can be found in 9193 citations. The manufacturer provides antibody testing data and validation: https://www.cellsignal.com/products/primary-antibodies/cleaved-caspase-3-asp175-antibody/9661

5) anti-cleaved-caspase-8 (Cell signaling, 8592, D2B5) can be found in 294 citations. The manufacturer provides antibody testing data and validation: https://www.cellsignal.com/products/primary-antibodies/cleaved-caspase-8-asp387-d5b2-xp-rabbit-mab/8592

6) anti F4/80 (AbD Serotec, MCA497, A3-1) can be found in 230 citations. The manufacturer provides antibody testing data and validation: https://www.bio-rad-antibodies.com/monoclonal/mouse-f4-80-antibody-cl-a3-1-mca497.html?f=purified

7) anti-CD45 (BD Biosciences, 560510,30F11) can be found in 7 citations. The manufacturer provides antibody testing data and validations: https://www.bdbiosciences.com/en-de/products/reagents/flow-cytometry-reagents/research-reagents/single-color-antibodies-ruo/alexa-fluor-700-rat-anti-mouse-cd45.560510

8) anti-TGN38 (bio-techne, AF8059-SP). The manufacturer also provides antibody testing data: https://www.bio-techne.com/p/antibodies/rat-tgn38-antibody_af8059

9) anti-E-cadherin (BD Biosciences, 610182) can be found in 5 citations. The manufacturer also provides antibody testing data: https://www.bdbiosciences.com/en-de/products/reagents/microscopy-imaging-reagents/immunofluorescence-reagents/purified-mouse-anti-e-cadherin.610182

10) anti-DARS-2 (Proteintech, 13807-1-AP) can be found in 4 citations. The manufacturer also provides antibody testing Data and knockout/knockdown validation: https://www.ptglab.com/products/DARS2-Antibody-13807-1-AP.htm#publications

11) Total OXPHOS Antibody Cocktail (Abcam, ab110413) can be found in 1023 citations. The manufacturer also provides antibody testing data: https://www.abcam.com/products/panels/total-oxphos-rodent-wb-antibody-cocktail-ab110413.html

12) anti-ß-actin (Santa Cruz, sc-1616, I-19) can be found in 1651 citations. The manufacturer discontinued the product.

13) anti-MTCO1/COX1 (Molecular probes, 459600, clone 1DE1A8) can be found in 117 citations. The manufacturer also provides antibody test data: https://www.thermofisher.com/antibody/product/MTCO1-Antibody-clone-1D6E1A8-Monoclonal/459600.

14) anti-COX4L1 (Molecular probes, A21348, 20E8C12) can be found in 82 citations. The manufacturer also provides antibody test data: https://www.thermofisher.com/antibody/product/OxPhos-Complex-IV-subunit-IV-Antibody-clone-20E8C12-Monoclonal/A21348

15) anti- UQCRC1 (Molecular probes, 459140, 16D10AD9AH5) can be found in 45 citations. The manufacturer also provides antibody test data: https://www.thermofisher.com/antibody/product/UQCRC1-Antibody-clone-16D10AD9AH5-Monoclonal/459140

16) anti NDUFS1 (Proteintech, 12444-1-AP) can be found in 60 citations. The manufacturer also provides antibody testing data and validation: https://www.ptglab.com/products/NDUFS1-Antibody-12444-1-AP.htm

17) anti-NDUFS2 (Abcam,ab96160) can be found in 7 citations. The manufacturer discontinued the product.

18) anti-NDUFV2 (Proteintech, 15301-1-AP) can be found in 36 citations. The manufacturer also provides antibody testing data and validation: https://www.ptglab.com/products/NDUFV2-Antibody-15301-1-AP.htm

19) anti-UQCRFS1/RISP (Abcam, ab14746) can be found in 77 citations. The manufacturer also provides antibody testing data: https://www.abcam.com/products/primary-antibodies/uqcrfs1risp-antibody-5a5-ab14746.html

20) anti-ATP5A (Abcam, ab14748,15H4C4) can be found in 482 citations. The manufacturer also provides antibody testing data: https://www.abcam.com/products/primary-antibodies/atp5a-antibody-15h4c4-mitochondrial-marker-ab14748.html

21) anti-SDHA (Molecular probes, 459200,2EGC12FB2AE2) can be found in 85 citations. The manufacturer also provides antibody testing data: https://www.thermofisher.com/antibody/product/SDHA-Antibody-clone-2E3GC12FB2AE2-Monoclonal/459200

22) anti-NDUFA9 (Molecular probes, 459100, 20C11B11B11) can be found in 58 citations. The manufacturer also provides antibody testing data: https://www.thermofisher.com/antibody/product/NDUFA9-Antibody-clone-20C11B11B11-Monoclonal/459100

23) anti-Tubulin (Sigma Aldrich, T6074) can be found in 1214 citations. The manufacturer also provides antibody testing data and validations: https://www.sigmaaldrich.com/DE/en/product/sigma/t6074

24) anti-TOMM70 (Sigma Aldrich, HPA014589) can be found in 9 citations. The manufacturer also provides antibody testing data and validation: https://www.sigmaaldrich.com/DE/en/product/sigma/hpa014589

25) anti-FABP2 (Proteintech, 21252-1-AP) can be found in 10 citations. The manufacturer also provides antibody testing data and validations: https://www.ptglab.com/products/FABP2-Antibody-21252-1-AP.htm

26) anti-FASN (Cell Signaling,3189S) can be found in 98 citations. The manufacturer also provides antibody testing data and validation:https://www.cellsignal.com/products/primary-antibodies/fatty-acid-synthase-antibody/3189

27) anti-Vinculin XP (Cell Signaling, 13901,E1E9V) can be found in 426 citations. The manufacturer provides antibody testing data and validation: https://www.cellsignal.com/products/primary-antibodies/vinculin-e1e9v-xp-rabbit-mab/13901

28) Goat Antibody Monospecific for Human Apolipoprotein (B, Beckmann Coulter,467905)

Secondary antibodies:

1) Donkey anti-rabbit IgG, conjugated to HRP antibody (GE Healthcare, NA934V). The manufacturer also provides antibody testing data: https://www.cytivalifesciences.com/en/de/shop/protein-analysis/blotting-and-detection/blotting-standards-and-reagents/amersham-ecl-hrp-conjugated-antibodies-p-06260

2) Sheep anti-mouse IgG, conjugated to HPR antibody (GE Healthcare, NA931). The manufacturer also provides antibody testing data: https://www.cytivalifesciences.com/en/us/shop/protein-analysis/blotting-and-detection/blotting-standards-and-reagents/amersham-ecl-hrp-conjugated-antibodies-p-06260

3) Donkey anti-goat IgG, conjugated to HPR antibody (Jackson Laboratories, 705-035-003) can be found in 211 citations. The manufacturer also provides antibody testing data: https://www.jacksonimmuno.com/catalog/products/705-035-003

4) Goat anti-guinea pig (Progen, 90001). The manufacturer also provides antibody testing data: https://www.progen.com/anti-guinea-pig-IgG-goat-polyclonal-HRP-conjugate/90001

5) Goat anti-mouse IgG (Vector Laboratories, BA9200) can be found in 993 citations. The manufacturer provides antibody testing data and validations : https://www.2bscientific.com/Products/Vector-Laboratories/BA-9200/Goat-Anti-Mouse-IgG-Antibody-HL-Biotinylated

6) Goat anti-rabbit IgG (Vector Laboratories, BA-1000) can be found in 3625 citations. The manufacturer provides antibody testing data and validation: https://www.2bscientific.com/Products/Vector-Laboratories/BA-1000/Goat-Anti-Rabbit-IgG-Antibody-HL-Biotinylated

7) Goat anti-rat IgG ( Vector laboratories, BA-9400) can be found in 250 citations. The manufacturer provides antibody testing data and validation: https://www.2bscientific.com/Products/Vector-Laboratories/BA-9400/Goat-Anti-Rat-IgG-Antibody-HL-Biotinylated

8) Donkey Anti-Sheep IgG NorthernLights™ NL557- conjugated Antibody( bio-techne, NL010) can be found in 13 citations. The manufacturer provides antibody testing data and validation: https://www.bio-techne.com/p/secondary-antibodies/donkey-anti-sheep-igg-northernlights-nl557-conjugated-antibody_nl010

9) Goat anti-mouse Alexa 488 (Molecular Probes, A1101) can be found in 2708 citations (https://www.biocompare.com/9776-Antibodies/7011934-Goat-anti-Mouse-IgG-H-L-Secondary-Antibody-Alexa-Fluor-488-conjugate/#citations).

10) Goat anti-guinea pig Alexa 633 (Molecular Probes, A21105) can be found in 84 citations (https://www.citeab.com/antibodies/2401222-a-21105-goat-anti-guinea-pig-igg-h-l-highly-cross)

## Eukaryotic cell lines

Policy information about [cell lines]

| | |
|---|---|
| Cell line source(s) | IEC-6 cells (ACC 111) were purchased from the Leibniz Institute DSMZ – German collection of microorganisms and Cell Cultures GmbH (lot 5, 18.06.2015). |
| Authentication | The cell lines were not authenticated. |
| Mycoplasma contamination | The cell lines were routinely tested and confirmed negative for mycoplasma contamination. |
| Commonly misidentified lines (See [ICLAC] register) | No ISLAC cell lines were used in this study |

## Animals and other organisms

Policy information about [studies involving animals]; [ARRIVE guidelines] recommended for reporting animal research

| | |
|---|---|
| Laboratory animals | The study used the following mouse lines: Dars2fl/fl (Dogan, S. A. et al., 2014), Cox10fl/fl (Diaz, F. et al. 2005), Vil1-Cre (Madison, B. B. et al., 2002) and Villin-CreERT2 (el Marjou, F. et al., 2004). Sdhatm2a mice were obtained from the Knock Out Mouse Project (KOMP) repository (Project ID: CSD48939) and bred to FLP deleter mice (Rodríguez, C. I. et al.,2000) to delete the FRT-flanked region to generate Sdhafl/fl mice.<br><br>Both female and male mice between 1 – 12 weeks of age were used in all in vivo experiments which is specified in figure legends for each experiment, whereas metabolic tracing studies were performed exclusively on male mice. Littermates not carrying the Vil1-Cre or Villin-CreERT2 transgenes were used as controls in all experiments. All mice were maintained in C57BL/6N background. Mice were housed at the specific-pathogen-free (SPF) animal facilities of the CECAD Research Center of the University of Cologne under a 12 h dark/12 h light cycle in individually ventilated cages (Greenline GM500; Tecniplast) at 22 (±2)°C and a relative humidity of 55 (±5) %. All mice had unlimited access to water and fed with a standard chow diet (Harlan diet no. 2918 or Prolab Isopro RMH3000 5P76) ad libitum. For the experiments assessing the role of dietary fat, mice were fed with fat-free diet (E15104-3474, ssniff-Spezialdiäten GmbH, Soest, Germany).<br><br>The study used the following Caenorhabditis elegans strains: Bristol N2, RT1315 unc-119(ed3); pwIs503[pvha-6::mans::gfp;cbr-unc-119], VS25 hjIs14 [vha-6p::GFP::C34B2.10(SP12) + unc-119(+)] and RT130 pwIs23 [vit-2::GFP]. All the experiments were performed with hermaphrodites worms at day 1 and 4 of adulthood that is indicated in the corresponding figures and/or figure legends |
| Wild animals | The study did not involve wild animals. |
| Field-collected samples | The study did not involve samples collected from the field. |
| Ethics oversight | All animal procedures included in the study were conducted in accordance with European, national and institutional guidelines and protocols were approved by local government authorities (Landesamt für Natur, Umwelt und Verbraucherschutz Nordrhein-Westfalen, Germany) and Animal Welfare Officers of University Medical Center Hamburg-Eppendorf and Behörde für Gesundheit und Verbraucherschutz Hamburg, Germany.<br><br>In this research, invertebrate C. elegans was used as an organismal model and no ethical approval was required. According to the "Zentrale Kommission für die Biologische Sicherheit" (ZKBS), the responsible entity inside the Bundesamt für Verbraucherschutz und Lebensmittelsicherheit to assess the risk of Genetically Modified Organisms (GMO), genetic work with C. elegans is classified as risk group 1 (biological safety level 1: S1). Accordingly, C. elegans work was performed in a S1-laboratory. The use of GMO in Germany is regulated by the "Gentechnik-Gesetz", and the guidelines applying to S1 work with GMO (i.e., documentation of the project and of the, exact description of the creation and maintenance of the genetic modification or correct waste treatment) were followed. |

Note that full information on the approval of the study protocol must also be provided in the manuscript.

