## [Peer Review File · Nature]

Manuscript Title: Mitochondrial dysfunction abrogates dietary lipid processing in enterocytes

Reviewer Comments & Author Rebuttals

Reviewer Reports on the Initial Version:

Referees' comments:

Referee #1 (Remarks to the Author):

In this manuscript by Moschandrea et al., the authors explore how mitochondrial dysfunction in the proximal intestine compromises lipid absorption. Through a series of genetic and molecular experiments using *Dars*, *Sdha* and *Cox10* conditional knockout alleles, the authors discover that: (1) intestinal specific knockouts of *DARS* and *SDHA* result in premature organismal death as well as massive lipid accumulation in the proximal intestine, mainly triacylglyceride filled lipid droplet (LD) accumulation; (2) The accumulation of LDs in the enterocytes are likely due to impaired transport of dietary lipids, as evidenced by the absence of LDs in fat-free diet fed animals and by the decreased ³H-Triolein tracing in plasma; (3) Dispersion of Golgi apparatus is seen in *Dars* deficient enterocytes, and similar dispersal is seen in the *C. elegans*' gut, suggesting that *DARS* regulation of ER-golgi transport is likely conserved and underlies the LD accumulation defects.

Overall, the authors present interesting findings that mitochondrial dysfunction contributes to lipid accumulation within proximal enterocytes. The data are clearly presented, and multiple cellular and molecular evidence support the notion of dietary lipid sources, cellular disorganization, and the accumulation of LDs. However, it is unclear how mitochondrial dysfunction leads to the disorganization of ER and Golgi transport – is the LD accumulation specifically and primarily caused by mitochondrial dysfunction, or are we observing general enterocyte malfunction secondary to organismal death? The lack of mechanistic insights into how disruption of mitochondria causes defects in the secretory pathway dampens my enthusiasm for Nature.

Major points

0. A major concern that I have of this manuscript is how mitochondrial dysfunction induces lipid accumulation and whether this finding results from specific regulation of mitochondria and lipid secretion, or whether the authors are observing secondary effects resulting from cellular and organismal death?

1. The authors nicely illustrate lipid accumulation in the proximal intestine, but it is unclear whether other nutrients (e.g., amino acids) are also influenced in the *Dars* deficient enterocytes. Are the defects seen in *Dars/Sdha* deficient intestine due to mitochondrial specific regulation of lipid secretion or general enterocyte malfunction?

2. In the acute models, does a fat-free diet delay early death caused by *Dars/Sdha* loss? Is lipid accumulation the primary cause of organismal death?

3. Are the secretory defects specific to LDs in enterocytes or any other secretory molecules in other cell types? Do secretory cells in the gut (e.g., Paneth cells) demonstrate similar secretion defects?

4. In *C. elegans* orthogonal studies:

a. What is the phenotype of *dars-1* deficiency in the gut? Do lipids/other nutrients also accumulate in *dars-2* RNAi experiments as a result of Golgi dispersal?

b. *C. elegans* gut has nascent autofluorescent gut granules readily seen in the GFP channels; How do authors specifically visualize mannose-GFP?

Minor points:

Line 261: fila should read phyla

Line 273-278: It's a nice parallel with human chylomicron retention disease patients, but it's not clear whether they also have mitochondrial defects, as seen by the authors

Extended Data 2, Figure C: What is the input material for qPCR? (whole epithelium?)

Referee #2 (Remarks to the Author):

The manuscript of Moschandrea et al. is very interesting and reports a novel role for mitochondrial function in intestinal epithelial cells for chylomicron formation. The authors report that mitochondrial dysfunction in intestinal epithelial cells caused by defects in mitochondrial protein synthesis, complex II or complex IV, leads to accumulation of large lipid droplets in the enterocytes. This phenotype is also observed in the proximal small intestine examined 7 days after inducible DARS2 knockout in adult mice. It is concluded that the accumulated lipids are mostly originating from dietary fat because a fat-free diet leads to disappearance of the large lipid droplets in the enterocytes. Also, the authors present data on disturbed intracellular transport and chylomicron formation. Metabolic tracing study reveal reduced amounts of circulating chylomicrons and it is argued the chylomicron transport from the intestine is compromised after DARS2 knockout. Overall, the study is well done, but I have a few comments that need to be addressed.

Major points:

1. The non-inducible knockouts of DARS2, SDHA and COX10 do not only generate a mitochondrial defect in the intestinal endothelial cells, but also seems to have a developmental component, e.g. the gut is shorter and it contain less stem cells. I think this developmental phenotype could be better described. I am also a bit puzzled about the connection to bioenergetics. The SDHA knockout, in principle, allows electron flow through Complex I, III and IV, whereas this flow should be disrupted in the DARS2 and COX10 knockouts. Are the developmental and physiological phenotypes identical. If so, why? Do the effects on the Krebs cycle caused by disruption of SDHA impact electron flow through Complex I by decreasing NADH generation?
2. The authors, perform an inducible knockout of DARS2 in intestinal epithelial cells in adult animals to exclude any impact on development and to directly study the impact on intestinal physiology. They attribute the accumulation of lipid droplets in intestinal epithelial cells to accumulation of dietary fats based on diet manipulation. However, several key factors regulate fatty acid beta-oxidation, e.g., PPAR α , Acat1, Acox1, Mcad, Ehhadh, are also significantly decreased (ED Fig. 3g). Can reduced fatty acid oxidation also contribute to the observed lipid accumulation in enterocytes, especially as fatty acid beta-oxidation occurs in mitochondria that are deficient in this model?
3. I do not fully understand 3H-triolein accumulation pattern as it is decreased in the small intestine. Is lipid absorption is also affected in the knockout mice? The authors argue that lipids can be taken up by intestinal epithelial cells but not excreted as chylomicrons. Would the impaired Golgi transport and formation of lipid droplets not also lead to accumulation of 3H-triolein in intestinal cells.
4. I was surprised to see that mitochondrial dysfunction seems to impact chylomicron formation. I understand that a specific mechanism may be very difficult to establish and this is maybe beyond the scope of the present paper. The authors present some data from *C. elegans* showing that DARS2 knockdown seems to affect Golgi trafficking in worms. Can this system or cell line work shed some more light on the involved processes? Does the inducible mouse knockout allow some temporal dissection of the process? I think some more mechanistic insights would strengthen the

paper.

Minor points:

1. Fig 1E, the mitochondrial abnormalities in IEC need to be quantified. Number of mitochondria, size of mitochondria, cristae density, etc.
2. For the ED Fig. 3g, I recommend stratifying the genes by their pathways, e.g., de novo lipogenesis, beta oxidation, TAG synthesis, and lipolysis, that will be much clearer to the general audience.
3. line 203 - 205, "reduced 3H-triolein uptake in plasma and in all metabolically active organs except the stomach (Fig. 4b)." Uptake in plasma sounds really weird. I suggest to delete the word "uptake".
4. The authors frequently use "inhibit", e.g. line 28-29: "Here we show that mitochondrial dysfunction in enterocytes inhibits chylomicron production and the transport of dietary lipids to peripheral organs." Line 34-35: "Feeding a fat-free diet inhibited the formation of LDs in DARS2 deficient enterocytes,..." I think a better wording would be "prevents" or "does not allow" or something like that.
5. Fig 5a. How do the authors distinguish a small size lipid droplet from a chylomicron? The data would be more robust if the authors quantify Golgi with chylomicrons, ER with pre-chylomicrons, and secreted/secreting chylomicrons.
6. Title: Mitochondria regulate dietary lipid processing in enterocytes. Showing regulation would require studies of intestinal epithelial cells with graded down- and up-regulation of mitochondrial function. I suggest to not refer to this as regulation.
7. I think the discussion on mitochondrial diseases is a bit questionable. Certainly, there are many mitochondrial diseases where malabsorption is a problem, e.g. MNGIE, some PolG mutations, Pearson's syndrome etc. However, as far as I know the role for mitochondrial dysfunction in intestinal epithelial cells is not well documented and other mechanisms may be more important, e.g. pancreatic insufficiency. I think this discussion point should be rewritten and perhaps it would be better to argue that the findings in the manuscript now warrants studies to determine whether a similar mechanism is also found in mitochondrial disease patients.

Referee #3 (Remarks to the Author):

This manuscript demonstrates that mice in which enterocyte mitochondrial function has been severely disrupted or ablated have accumulation of lipid droplets in enterocytes that is apparently due to impaired chylomicron secretion, apparently reflecting a broader defect in the ER-to-Golgi secretory pathway.

Specific comments:

1. It appears that mRNA analyses were not performed globally in the intestine using RNAseq but only selected genes were quantitate by qPCR. A more global approach to gene expression would complement the proteomic and lipidomic analyses.
2. Does heterozygous deletion of DARS2 in enterocytes produce a milder form of mitochondrial dysfunction and/or influence dietary lipid handling? More broadly, is there a continuous relationship between enterocyte mitochondrial function and ability to secrete chylomicrons or is this phenotype only revealed when mitochondrial function is ablated?
3. Do the enterocytes lacking DARS2 (or other genes resulting in loss of mitochondrial function) display evidence of ER stress?

4. The plasma lipid concentrations of triglycerides and cholesterol should be provided for each mouse model reported in this manuscript as additional evidence for systemic intestinal lipid malabsorption.
5. The methods state that ¹⁴C-triolein was used for the oral lipid tracer studies but the results refer to ³H-triolein. Was the tracer on the fatty acids or the glycerol in the labeled triolein? This is critically important to the interpretation of these results.
6. Intestinal chylomicrons are secreted into intestinal lymph prior to accessing the blood. A more direct approach to physiologically assessing intestinal triglyceride secretion would be to measure TG mass and the orally-fed ³H-fatty acid tracer in the thoracic duct lymph from the mice.
7. ApoB-48 is the major structural protein in chylomicrons produced by enterocytes and is required for chylomicron secretion but is not mentioned in the results. Was it detected in the proteomics studies? ApoB-48 should be assessed in the intestinal tissue by western blot and measured in the plasma (and lymph) by an immunoquantitative technique.
8. The entire ER-to-Golgi transport secretory pathway appears to be disrupted. Other proteins known to be secreted by enterocytes should be measured as a physiological support for this broader disruption that is not just specific to chylomicron secretion per se.
9. What is known about the energy requirements of the ER-to-Golgi transport secretory pathway? One would assume that this is the major issue in the disruption of this pathway.
10. What is known about the effects of impaired or ablated mitochondrial function on the protein secretory pathway in other cell types, such as hepatocytes? Using the DARS2 floxed mice to ablate mitochondrial function in hepatocytes in mice and assessing the impact on LD accumulation and hepatocyte VLDL secretion would substantially enhance this report by either showing specificity of this finding for the enterocyte or broader applicability of this finding to hepatic lipid secretion as well.
11. The authors suggest that their findings may help explain some of the gastrointestinal problems in some patients with mitochondrial disorders. If so, one might expect evidence for lipid malabsorption, including LD accumulation in the small intestine and reduced plasma lipids—is that the case in these patients?

Referee #4 (Remarks to the Author):

In this manuscript the authors reported that mitochondrial dysfunction induced by DARS2 knockout impaired chylomicron maturation and affected gut lipid absorption capacity. The authors confirmed the findings in two more murine models lacking key proteins for mitochondrial function (SDH and COX10A). They document massive accumulation of dietary fat (TAG) in the enterocytes from the knockout models.

Although the phenotypes are extensively described in the manuscript, a clear explanation of the molecular mechanism underlying these effects is absent. The KO mice look very compromised, with severe impairment of intestinal morphology (villi are disorganized). Also, both labelled triolein and glucose uptake is impaired, suggesting a generic dysfunction of nutrient absorption (do the mice have diarrhea?), and not a lipid-specific signature of the phenotype. The deep impairment of the mitochondrial function may be responsible for a status of energetic depletion, that prevents proliferation (as reported) and possibly maturation of functional enterocytes. Cells can be just “exhausted”, without sufficient ATP to sustain physiological processes, including the transfer of pre-chylomicrons from ER to Golgi for maturation. There are evidences suggesting this might be an ATP dependent process (<https://doi.org/10.1242/jcs.00215>).

Over the manuscript falls short of providing new conceptual insight or specific mechanistic links between mitochondria and dietary lipid absorption.

Comments:

- 1) Gene expression in DARS2 KO mice is inconsistent: some genes are upregulated while some

others are downregulated. Do they have an explanation for the reduced expression of PPAR α and some of the targets?

2) The nutritional status of the mice is not reported at the time of sacrifice. This is particularly relevant for EM experiment when looking at the presence of the chylomicrons. Such an experiment should be performed in response to a lipid load (feeding with high fat diet or oil gavage).

3) When mice are fed fat free diet, there is no lipid accumulation. However, these mice show impaired crypt cells proliferation and dramatic weight loss. This seems to indicate that lethality is NOT specifically related to dietary lipid accumulation but rather to an overall unhealthy status of the mice.

a. Another explanation for the phenotype may stem from enterocyte death from mitochondrial dysfunction, thereby resulting in blunted villi. This would lead to malabsorption in carbohydrates, protein, and fats. It would be helpful to know if the KO mice on a fat free diet had diarrhea. It is also unknown if the KO mice on a fat free diet survived and were completely normal or if they continued to demonstrate defects despite the absence of fat. If the KO mice on a fat free diet did not survive, it is likely the phenotype is just due to a sick mouse.

4) The 3H triolein absorption study highlighted lower counts in duodenum of KO mice: these suggest that enterocytes are taking up less lipid from the lumen, which might explain itself the reduced counts in the plasma.

5) A better characterization of chylomicrons is required: all the conclusions are taken based on EM. The authors should isolate chylomicrons from plasma and measure ApoB in both plasma and isolated chylomicrons.

6) It appears that the phenotype may be a secondary effect of ATP depletion from impaired mitochondria. It would be more convincing if the authors showed that intracellular trafficking in general was not affected and that chylomicron transport was specifically affected. Otherwise, if all trafficking is impaired, then this is likely a result of impaired energy production.

7) It is still unclear why triglycerides are higher in the intestines of these knock-out mice. The authors are unable to differentiate between whether this is an effect of decreased chylomicron secretion or whether this is due to lipid accumulation from impaired fatty acid oxidation. It seems to be the later based on the increase in acylcarnitine. This was noted by the authors, but not elaborated on.

8) If the proximal and distal small intestine have appropriate knockdown of DARS2, why was the distal small intestine not affected? The authors state "lipid accumulation in IEC's depend on anatomic location". I am not sure I understand this statement since the distal small intestine can also secrete chylomicrons.

Author Rebuttals to Initial Comments:

Point by point response to the reviewer comments

Referees' comments:

Referee #1 (Remarks to the Author): IECs

In this manuscript by Moschandrea et al., the authors explore how mitochondrial dysfunction in the proximal intestine compromises lipid absorption. Through a series of genetic and molecular experiments using Dars, Sdha and Cox10 conditional knockout alleles, the authors discover that: (1) intestinal specific knockouts of DARS and SDHA result in premature organismal death as well as massive lipid accumulation in the proximal intestine, mainly triacylglyceride filled lipid droplet (LD) accumulation; (2) The accumulation of LDs in the enterocytes are likely due to impaired transport of dietary lipids, as evidenced by the absence of LDs in fat-free diet fed animals and by the decreased 3H-Triolein tracing in plasma; (3) Dispersion of Golgi apparatus is seen in Dars deficient enterocytes, and similar dispersal is seen in the C. elegans' gut, suggesting that DARS regulation of ER-golgi transport is likely conserved and underlies the LD accumulation defects.

Overall, the authors present interesting findings that mitochondrial dysfunction contributes to lipid accumulation within proximal enterocytes. The data are clearly presented, and multiple cellular and molecular evidence support the notion of dietary lipid sources, cellular disorganization, and the accumulation of LDs.

However, it is unclear how mitochondrial dysfunction leads to the disorganization of ER and Golgi transport – is the LD accumulation specifically and primarily caused by mitochondrial dysfunction, or are we observing general enterocyte malfunction secondary to organismal death? The lack of mechanistic insights into how disruption of mitochondria causes defects in the secretory pathway dampens my enthusiasm for Nature.

Major points

0. A major concern that I have of this manuscript is how mitochondrial dysfunction induces lipid accumulation and whether this finding results from specific regulation of mitochondria and lipid secretion, or whether the authors are observing secondary effects resulting from cellular and organismal death?

The reviewer raises a legitimate concern about the possible role of epithelial cell death in causing the lipid accumulation phenotype in DARS2^{IEC-KO} mice. We have indeed extensively studied the possible involvement of cell death in causing the intestinal pathology of these animals. In fact, we started this project aiming to study the role of mitochondrial function in IEC death and intestinal inflammation, as cell death and its role in inflammation is the main focus of our laboratory. However, to our surprise, we found no evidence of increased IEC death in the intestines of either the constitutive DARS2^{IEC-KO} or tamoxifen-inducible DARS2^{tamIEC-KO} mice, showing that DARS2 deficiency does not cause increased death of enterocytes (Data now included in ED Fig. 1e and ED Fig. 5b, d of the revised manuscript). In addition, to assess the possible role of apoptosis and necroptosis in the development of the intestinal pathology we have generated DARS2^{IEC-KO} mice that simultaneously lack the apoptotic caspase-8 in IECs, as well as MLKL, the protein executing necroptosis, in all cells

(DARS2^{IEC-KO} CASP8^{IEC-KO} *Mlkl*^{-/-} mice) and found that even combined inhibition of apoptosis and necroptosis did not prevent the intestinal pathology and the death of the mice (see **Figure 1 for reviewers** below). Thus, DARS2 deficiency does not cause considerably increased cell death in the intestine arguing against cell death being a primary driver of the observed intestinal pathology and particularly the massive lipid accumulation in enterocytes of DARS2^{IEC-KO} mice. Furthermore, our new immunostainings presented in Fig. 5d of the revised manuscript clearly showed that lipid droplets start forming in DARS2-deficient enterocytes already 5 days after tamoxifen induction, a time point when the epithelium is not yet severely affected and the mice are fully viable. These immunostainings also clearly show that lipid droplets start forming in cells that do not show any sign of undergoing cell death, demonstrating that the accumulation of fat within lipid droplets is not an indirect effect caused by enterocyte death. We would also like to comment that during the last 15 years we have studied in our lab a large number of mouse models with IEC-specific knockouts of key pro-survival proteins, which sensitized IECs to different types of cell death including apoptosis, necroptosis and pyroptosis and caused intestinal inflammation (Dannappel et al., 2014; Eftychi et al., 2019; Nenci et al., 2007; Schwarzer et al., 2020; Vlantis et al., 2016a; Vlantis et al., 2016b; Welz et al., 2011). Whereas all these models showed strongly increased numbers of dying IECs causing severe inflammatory pathologies in the gut, we, but also other investigators performing similar studies, did not detect lipid accumulation in enterocytes in any of these mouse models with increased IEC death, arguing that death of epithelial cells does not cause lipid accumulation. Therefore, we can exclude that the massive accumulation of lipids in large lipid droplets in enterocytes from DARS2^{IEC-KO} mice is a secondary consequence of cell death.

In addition to cell death, we were also curious whether the strong inhibition of epithelial proliferation caused by DARS2 ablation could contribute to the lipid accumulation. However, extensive data in the literature demonstrate that inhibition of IEC proliferation does not result in lipid accumulation. As an example, we refer to a manuscript published by the group of Hans Clevers (van Es et al., 2012), showing that tamoxifen-inducible ablation of *Tcf4* in IECs, using the same *Villin-Cre^{ERT2}* mouse we used in our study, caused complete inhibition of epithelial cell proliferation and resulted in the death of the mice about 9 days after tamoxifen induction, which is approximately the time that DARS2^{tamIEC-KO} mice need to be sacrificed due to weight loss. As is clearly shown in the manuscript by van Es et al, complete inhibition of epithelial proliferation in mice with IEC-specific *Tcf4* knockout did not cause lipid accumulation in enterocytes, arguing that inhibition of proliferation could not be the cause of lipid accumulation. Therefore, our additional results included here but also published literature provide compelling evidence that the massive accumulation of lipids within large lipid droplets in enterocytes lacking DARS2, but also SDHA and COX10, is not a secondary effect caused by epithelial or organismal death, or by inhibition of epithelial proliferation.

Figure 1 for reviewers. Combined inhibition of caspase-8-dependent apoptosis and MLKL-dependent necroptosis did not ameliorate the intestinal pathology of DARS2^{IEC-KO} mice. 1-week-old DARS2^{IEC-KO} CASP8^{IEC-KO}Mlkl^{-/-} mice showed reduced body weight and serum glucose levels (a) as well as reduced epithelial cell proliferation concomitant with massive accumulation of lipids in large lipid droplets in enterocytes (b), similarly to DARS2^{IEC-KO} mice.

1. The authors nicely illustrate lipid accumulation in the proximal intestine, but it is unclear whether other nutrients (e.g., amino acids) are also influenced in the Dars deficient enterocytes. Are the defects seen in Dars/Sdha deficient intestine due to mitochondrial specific regulation of lipid secretion or general enterocyte malfunction? We have not specifically assessed the transport of amino acids, but we measured glucose absorption and transport in DARS2^{tamIEC-KO} mice. As shown in Figure 4d of the revised manuscript, transport of orally administered ¹⁴C-labelled DOG to the plasma was not affected by DARS2 deficiency in enterocytes, in contrast to the transport of ³H-triolein to the plasma that was strongly suppressed in the same animals (Figure 4c). These findings argue that DARS2-deficient enterocytes do not show a general malfunction resulting in impairment of all nutrient transport but a rather specific defect in transporting dietary fat.

2. In the acute models, does a fat-free diet delay early death caused by Dars/Sdha loss? Is lipid accumulation the primary cause of organismal death?

Our studies showed that feeding a fat-free diet strongly prevented lipid accumulation in enterocytes of DARS2^{tamIEC-KO} mice, with these animals showing only a few small lipid droplets in proximal enterocytes 7 days after tamoxifen induction, in contrast to mice on chow diet that exhibited massive lipid accumulation in large lipid droplets at this stage (Figure 4a). However, as shown in ED Fig. 7b of the revised manuscript, DARS2^{tamIEC-KO} mice on fat-free diet also showed progressive weight loss after tamoxifen inducible DARS2 ablation, which necessitated their culling according to the animal experiment protocol. In most of our experiments using mice on normal chow diet we sacrificed the mice at days 7 or 8 after the last tamoxifen injection as this is the time point when most mice reached about 20% weight loss. We used this time point for the analysis of the mice on fat-free diet for consistency and in order to be able to compare the results with the normal diet. However, in one pilot experiment we run a small group of mice on fat-free diet when we followed the mice and culled them when they reached 20% loss of body weight. In this experiment, we had to sacrifice the mice at day 8 (1 mouse),

day 9 (2 mice) and day 10 (3 mice). Therefore, fat-free diet feeding delayed by about two days but ultimately could not prevent the severe loss of body weight necessitating the culling of the mice upon tamoxifen-inducible DARS2 ablation in IECs. These results show that lipid accumulation exacerbates but does not appear to be the primary cause of weight loss and subsequent death of the mice. Moreover, mice with inducible enterocyte-specific ablation of *Mttp* displayed impaired chylomicron production and accumulation of fat in large lipid droplets in enterocytes but did not show the dramatic weight loss and death observed in DARS2^{tamIEC-KO} mice (Iqbal et al., 2013), further supporting that the impairment of dietary lipid transport is not the primary cause of death in DARS2^{tamIEC-KO} mice. While the exact reason causing weight loss and ultimately the death of the mice is not clear at present, we would like to refer again to the study by Clevers and colleagues showing that tamoxifen-inducible knockout of *Tcf4* in IECs also caused the death of the mice about 9 days after tamoxifen induction (van Es et al., 2012). These mice did not show lipid accumulation, but displayed strong inhibition of epithelial cell proliferation similarly to DARS2^{tamIEC-KO} mice, arguing that suppression of IEC proliferation is a possible cause of death likely due to impairment of epithelial cell renewal. Therefore, mitochondrial dysfunction clearly has important pathological consequences in intestinal epithelial cells **in addition** to the lipid processing defect, as for example the complete impairment of IEC proliferation. In fact, we did not mean to imply that the lipid accumulation phenotype was the only and most critical consequence of mitochondrial dysfunction in enterocytes causing the death of the mice. We have now edited the text to clarify this point and avoid misleading the readers that lipid accumulation is the primary cause of organismal death.

3. Are the secretory defects specific to LDs in enterocytes or any other secretory molecules in other cell types? Do secretory cells in the gut (e.g., Paneth cells) demonstrate similar secretion defects?

We have examined the presence of Paneth cells in the intestines of DARS2^{tamIEC-KO} mice by immunostaining sections with antibodies against Lysozyme. As shown in Figure 2a for reviewers below, the number of Paneth cells was strongly reduced in the proximal small intestine of DARS2^{tamIEC-KO} mice 8 days after the last tamoxifen injection when enterocytes show massive accumulation of lipids within large lipid droplets. However, the number of Paneth cells in the distal small intestine of these mice was not strongly affected, suggesting that the disappearance of Paneth cells in the proximal small intestine is likely caused by indirect effects related to the lipid accumulation in this part of the tissue. Moreover, proximal small intestinal sections from DARS2^{tamIEC-KO} mice showed similar numbers of Paneth cells compared to the control mice 3 and 5 days after the last tamoxifen injection. Therefore, loss of Paneth cells occurs progressively in the proximal but not the distal small intestine of DARS2^{tamIEC-KO} mice, arguing that DARS2 ablation does not cause an immediate and direct effect in Paneth cells. We have also attempted to assess the integrity of the Golgi network in Paneth cells using immunofluorescence similarly to our approach in enterocytes. Specifically, we immunostained proximal small intestinal sections from DARS2^{tamIEC-KO} and control mice for lysozyme, TGN38 and E-cadherin. Because of the massive loss of lysozyme staining at 8 days, we focused on intestinal sections from mice 5 days after tamoxifen induction. However, as shown in panel b of the Figure 2 for reviewers below, we were not able to observe convincing staining for TGN38 in Paneth cells. We are not sure at this stage what is the reason for this, but it is likely that because of their unique physiology and the massive accumulation of secretory vesicles containing lysozyme and other anti-microbial peptides Paneth cells have a different Golgi structure compared to absorptive enterocytes. Because of this, unfortunately

we were not able to obtain a convincing answer as to whether Paneth cells lacking DARS2 also show defects in the secretory pathway. While we agree that this is an interesting question that is worth investigating in the future, we suggest that this is outside the scope of the current manuscript which focuses on the role of mitochondria in regulating dietary lipid processing and transport in enterocytes.

Figure 2 for Reviewers. Loss of Paneth cells in the proximal but not the distal small intestine of *DARS2^{tamIEC-KO}* and *Dars2^{fl/fl}* littermate control mice 8 days after tamoxifen induction.

a) Sections from the proximal and distal small intestine of *DARS2^{tamIEC-KO}* mice 3, 5 and 8 days after tamoxifen induction were immunostaining for lysozyme (brown colour) and counterstained with haematoxylin. Paneth cells appear as cells stained strongly for lysozyme at the bottom of the crypts. b) Sections from proximal small intestinal tissue of *DARS2^{tamIEC-KO}* and *Dars2^{fl/fl}* littermate control mice 5 days after the last tamoxifen injection were immunostaining for TGN38, E-cadherin and lysozyme. DAPI stains nuclei.

4. In *C. elegans* orthogonal studies:

a. What is the phenotype of *dars-1* deficiency in the gut? Do lipids/other nutrients also accumulate in *dars-2* RNAi experiments as a result of Golgi dispersal?

Dars-2 knockdown in *C. elegans* does not lead to accumulation of cytoplasmic lipid droplets, or gut granules. However, one needs to consider that in *C. elegans* the gut does not perform only the nutrient absorption and transport functions of the mammalian gut, but also acts as liver, adipose tissue and a collection of endocrine glands. Therefore, the handling of lipids in the *C. elegans* gut is expected to be different than that of the mammalian intestine. For this reason, we believe that we will need to perform a much more thorough investigation of how mitochondrial dysfunction affects intestinal cells in *C. elegans* before we can draw firm conclusions. We therefore decided not to include the *C. elegans* data in the revised manuscript in order to focus on our results in mice and particularly the new findings showing that Golgi dispersal occurs only in enterocytes handling dietary fat and precedes the formation of lipid droplets.

b. C. elegans gut has nascent autofluorescent gut granules readily seen in the GFP channels; How do authors specifically visualize mannose-GFP?

The autofluorescence signal coming from gut granules/LROs is very low and was easily distinguished from the mannose-GFP used in our studies.

Minor points:

Line 261: fila should read phyla

We thank the reviewer for pointing out this typo.

Line 273-278: It's a nice parallel with human chylomicron retention disease patients, but it's not clear whether they also have mitochondrial defects, as seen by the authors

As stated in the manuscript, chylomicron retention disease is caused by mutations in the GTPase Sar1b, which controls the transport of PCTVs from the ER to the Golgi. Mitochondrial defects have not been reported in these patients and according to published electron microscopy images (Okada et al., 2011), mitochondria appear normal. We did not mean to imply that chylomicron retention disease provides evidence for a role of mitochondria in this process. Our aim in mentioning this disease was to draw the reader's attention to the fact that inhibition of PCTV transport from the ER to the Golgi causes lipid accumulation in human enterocytes resembling the phenotype of the mice. Together with our results showing impaired chylomicron production and Golgi disorganisation in DARS2-deficient enterocytes, this finding suggests that mitochondrial dysfunction could cause the pathology by interfering with PCTV generation and/or transport. We have now edited the text to clarify this point.

Extended Data 2, Figure C: What is the input material for qPCR? (whole epithelium?)

The qPCR has been performed using RNA isolated from a small piece of the small intestine from 7-day-old pups.

Referee #2 (Remarks to the Author): mitochondrial biology

The manuscript of Moschandrea et al. is very interesting and reports a novel role for mitochondrial function in intestinal epithelial cells for chylomicron formation. The authors report that mitochondrial dysfunction in intestinal epithelial cells caused by defects in mitochondrial protein synthesis, complex II or complex IV, leads to accumulation of large lipid droplets in the enterocytes. This phenotype is also observed in the proximal small intestine examined 7 days after inducible DARS2 knockout in adult mice. It is concluded that the accumulated lipids are mostly originating from dietary fat because a fat-free diet leads to disappearance of the large lipid droplets in the enterocytes. Also, the authors present data on disturbed intracellular transport and chylomicron formation. Metabolic tracing study reveal reduced amounts of circulating chylomicrons and it is argued the chylomicron transport from the intestine is compromised after DARS2 knockout. Overall, the study is well done, but I have a few comments that need to be addressed.

Major points:

1. The non-inducible knockouts of DARS2, SDHA and COX10 do not only generate a mitochondrial defect in the intestinal endothelial cells, but also seems to have a developmental component, e.g. the gut is shorter and it contain less stem cells. I think this developmental phenotype could be better described.

IEC-specific knockout of DARS2, SDHA and COX10 caused essentially identical phenotypes. All these mice showed shortening of the intestine, reduced epithelial proliferation and diminished stem cell numbers. Our interpretation of these findings is that intestinal shortening as well as reduction of the crypt stem cells is directly related to the strong suppression of epithelial cell proliferation. Intestinal crypt stem cells divide regularly to give rise to the cells populating the transient amplifying zone as well as to self-renew to maintain the stem cell pool. Therefore, suppression of proliferation is expected to directly affect stem cell numbers. Similarly, particularly in the early postnatal stage, epithelial cell proliferation is critical for the growth of the intestine, therefore suppression of IEC proliferation will, as a consequence, limit the growth of the tissue resulting in shorter intestines in these mice compared to healthy littermates. We would like to point out that tamoxifen-inducible DARS2 knockout also caused suppression of IEC proliferation resulting in reduced stem cells in adult mice, showing that the defect in the stem cell compartment is not developmental in nature but rather linked to the proliferation impairment. We thank the reviewer for raising this issue, which was indeed not clearly discussed in our manuscript. We have edited the text in the revised manuscript to increase clarity.

I am also a bit puzzled about the connection to bioenergetics. The SDHA knockout, in principle, allows electron flow through Complex I, III and IV, whereas this flow should be disrupted in the DARS2 and COX10 knockouts. Are the developmental and physiological phenotypes identical. If so, why? Do the effects on the Krebs cycle caused by disruption of SDHA impact electron flow through Complex I by decreasing NADH generation?

As explained above, IEC-specific ablation of DARS2, COX10 or SDHA caused identical phenotypes, despite the fact that these knockouts result in mitochondrial dysfunction by different mechanisms. Complex II is the second entry point of electrons to the electron transport chain and as such contributes to the electron flow and overall ATP production in the cell. Through its second role in the TCA cycle it also indirectly impacts the production of NADH

in the cell, thus also affecting the electron flow through Complex I. Therefore, it is not surprising that the loss of Complex II leads to a similar effect as DARS2 depletion. Reciprocally, Complex I deficiency will also, very quickly lead to TCA defect due to inability to recycle NADH. Respiratory chain deficiency as a primary effect of Complex II deficiency has been previously described for SDHCdeficient mice (Al Khazal et al., 2019), which develop Leigh-like syndrome, commonly caused by mt-tRNA mutations, CIV assembly factor mutations (including COX10) and CI mutations. Similar LS associations have been described in patients (Bourgeron et al., 1995; Horvath et al., 2006; Parfait et al., 2000). Moreover, our own preliminary data revealed strong respiratory chain deficiency caused by SDHA depletion specifically in heart or skeletal muscle. Our interpretation of these results is that the impaired transport and massive accumulation of lipids in enterocytes is a direct consequence of mitochondrial dysfunction, irrespectively of whether this is induced by defects in different respiratory complexes.

2. The authors, perform an inducible knockout of DARS2 in intestinal epithelial cells in adult animals to exclude any impact on development and to directly study the impact on intestinal physiology. They attribute the accumulation of lipid droplets in intestinal epithelial cells to accumulation of dietary fats based on diet manipulation. However, several key factors regulate fatty acid beta-oxidation, e.g., PPAR α , Acat1, Acox1, Mcad, Ehhadh, are also significantly decreased (ED Fig. 3g). Can reduced fatty acid oxidation also contribute to the observed lipid accumulation in enterocytes, especially as fatty acid beta-oxidation occurs in mitochondria that are deficient in this model?

We thank the reviewer for this comment, which highlights one aspect of our study that we did not discuss and explain sufficiently in our manuscript. The reviewer correctly points out that fatty acid beta-oxidation is expected to be suppressed in cells with severe respiratory chain deficiency and wonders whether this may contribute to the lipid accumulation observed in enterocytes lacking DARS2, SDHA or COX10. Mutation of all these proteins is expected to lead to severe OXPHOS defect and reduced fatty acid beta-oxidation. Indeed, our metabolomic analysis revealed drastic accumulation of almost all species of long chain and very long chain acylcarnitines in DARS2 knockout IECs (ED Fig. 4e.), supporting that DARS2 deficiency did cause severe impairment of fatty acid beta-oxidation. However, although impaired fatty acid beta-oxidation could also contribute to the accumulation of lipids DARS2-deficient enterocytes, our experimental studies provided evidence that this cannot be the main cause of the massive lipid accumulation in our mice. Specifically, if defective lipid utilisation due to impaired fatty acid beta-oxidation were the primary cause for the lipid accumulation, then we would expect this to happen in both the proximal and distal small intestine that both exhibited robust OXPHOS deficiency upon DARS2 ablation (Fig.3, ED Fig.5). The fact that lipid accumulation occurs specifically in proximal enterocytes, which are primarily responsible for lipid absorption and transport as opposed to distal enterocytes that do not play an important role in lipid transport (please see also our response to comment #8 of reviewer 4 on this issue), strongly argues that the accumulation is not caused by impaired lipid catabolism due to reduced fatty acid beta-oxidation, but is rather the consequence of impaired transport. In addition, our findings that feeding a fat-free diet suppressed the accumulation of lipids, provided experimental evidence that the lipids stored in lipid droplets originate mainly from dietary fat. Together, these data provide evidence supporting that, whereas inhibition of fatty acid beta-oxidation might also contribute to the accumulation of lipids to a small extent, impaired transport of dietary fat is the primary reason for the accumulation of lipids in proximal enterocytes of mice with mitochondrial dysfunction.

3. I do not fully understand 3H-triolein accumulation pattern as it is decreased in the small intestine. Is lipid absorption also affected in the knockout mice? The authors argue that lipids can be taken up by intestinal epithelial cells but not excreted as chylomicrons. Would the impaired Golgi transport and formation of lipid droplets not also lead to accumulation of 3H-triolein in intestinal cells.

Our data showing that feeding the mice with a fat-free diet strongly prevented the formation of large lipid droplets in DARS2-deficient enterocytes demonstrated that the lipids accumulating in these cells are primarily derived from dietary fat, which argues that food-derived lipids are taken up by the enterocytes and stored in lipid droplets instead of being secreted in the form of chylomicrons. The reviewer correctly points out that one would expect accumulation of ³H-triolein in DARS2-deficient enterocytes, whereas in our experiments we observed a small reduction of ³H--triolein counts in the proximal small intestine of these mice compared to controls (Fig. 4c). We cannot fully explain this finding at this stage, but we believe this could be related to compensatory mechanisms activated by the cells, which upon sensing that lipids start to accumulate may try to slow down further lipid uptake to prevent a lipid overload. A small reduction in intestinal uptake of ³H--triolein was also observed in our new metabolic tracing experiments that were performed in mice 5 days after tamoxifen induction (ED. Fig. 9c), suggesting that cells may attempt to suppress lipid uptake from the lumen early on in an effort to prevent lipid overload. It is also possible that the two-hour time window of the metabolic tracing experiment is too short to observe the ultimate accumulation of the lipid tracer, particularly if the net balance between a throttled-down uptake and suppressed secretion is small. This effect might be exaggerated by the bolus administration of a large amount of lipids via oral gavage in the metabolic tracing experiments. However, as shown by the massive accumulation of dietary lipids in lipid droplets at 7-8 days after tamoxifen induction, on balance more lipids are taken up than are secreted. This is also supported by comparing the relatively modest reduction in uptake (about 60% of that in control mice) by the intestine to the strongly diminished secretion to the circulation (about 8-fold less ³H--triolein in plasma, Fig. 4b). Therefore, together with our findings that DARS2-deficient enterocytes lack mature chylomicrons and show severe disorganisation of the Golgi network, these findings support that the primary defect lies in the processing and secretion of dietary lipids and not in their uptake.

4. I was surprised to see that mitochondrial dysfunction seems to impact chylomicron formation. I understand that a specific mechanism may be very difficult to establish and this is maybe beyond the scope of the present paper. The authors present some data from C. elegans showing that DARS2 knockdown seems to affect Golgi trafficking in worms. Can this system or cell line work shed some more light on the involved processes? Does the inducible mouse knockout allow some temporal dissection of the process? I think some more mechanistic insights would strengthen the paper.

We thank the reviewer for this comment. We are particularly glad that the reviewer recognises that fully elucidating the specific mechanism by which mitochondria control chylomicron production might indeed be very difficult and will require an extended series of new experiments going far beyond the scope of the present paper, which aims to provide the first report that mitochondria critically contribute to dietary lipid transport in enterocytes. As the reviewer correctly points out, cell line work could indeed be a valuable tool to test and dissect the potential mechanisms. We have made extensive efforts to establish a cellular system

recapitulating the defects seen in proximal enterocytes *in vivo*, both using enterocyte cell lines and organoids, but unfortunately without success so far. We reason that optimal lipid transport in proximal enterocytes is controlled by a number of factors that likely involve hormones and mediators that are present *in vivo* but are very difficult to recapitulate in a cellular culture system. While we will continue investing in developing such a system, unfortunately this is not available at this stage. Considering that the *in vivo* model is all we can use currently to elucidate this process, as the reviewer also suggested, we focused our efforts on temporally dissecting the events leading to the lipid transport defect and lipid droplet formation in DARS2-deficient enterocytes. Our previous experiments revealed the formation of large cytoplasmic lipid droplets in enterocytes 7 to 8 days after tamoxifen-inducible DARS2 ablation, which correlated with lack of chylomicrons and impaired transport of dietary lipids to the circulation. Analysis of intestinal tissues from mice 3 days after tamoxifen induction did not show any signs of lipid accumulation, despite the fact that enterocytes from these animals displayed loss of mitochondria-encoded respiratory chain subunits and a complete inhibition of proliferation indicative of severe mitochondrial dysfunction already at this stage (ED Fig, 6b, d). We then analysed intestinal tissue samples from mice sacrificed two days later, namely 5 days after tamoxifen induction, and found that these animals displayed an intermediate phenotype with only a small number of proximal enterocytes containing small lipid droplets identified by immunostaining for perilipin 2 (PLIN2) and Oil Red O staining (ED Fig. 6h). Together, these results showed that lipid accumulation does not occur immediately after loss of mitochondrial function, but occurs progressively starting around 5 days after tamoxifen induction. With this knowledge, we then assessed whether the impairment of Golgi organisation occurred before or after lipid droplet formation by immunostaining intestinal tissue sections from mice 5 days after tamoxifen induction against PLIN2 (lipid droplet marker), TGN38 (Golgi marker) and E-cadherin. The results of these experiments clearly showed that Golgi organisation was already significantly impaired at this stage, with most enterocytes showing weak and dispersed TGN38 staining (Fig 5d, ED Fig. 10a). Most importantly, we found that Golgi dispersion was evident also in enterocytes that did not contain lipid droplets yet, arguing that Golgi disorganisation is not a secondary consequence of the accumulation of lipid droplets but it rather precedes their formation (see Figure 5d).

In addition, we then asked whether Golgi dispersion was also observed in distal enterocytes from DARS2^{tamIEC-KO} mice, which do not show lipid accumulation in contrast to proximal enterocytes that contain large lipid droplets 8 days after tamoxifen induction. Immunostaining for PLIN2/TGN38/E-cadherin revealed that distal enterocytes did not show the dramatic impairment of Golgi organisation observed in proximal enterocytes from the same DARS2^{tamIEC-KO} mice, with the TGN38 staining being largely unaffected (ED Fig. 10b). We consider this a very important finding, as it shows that although the entire intestinal epithelium in DARS2^{tamIEC-KO} mice shows severe mitochondrial dysfunction, severe Golgi dispersion is specifically observed in proximal enterocytes that are primarily responsible for absorbing, processing, packaging and exporting dietary lipids and not in distal enterocytes that are involved only to a minor extent in dietary lipid processing. These findings argue that Golgi disorganisation is not a primary and direct consequence of severe mitochondrial dysfunction generally induced in all enterocytes, but is rather linked to the specific physiology of proximal enterocytes that are primarily endowed with the task of transporting dietary lipids in the form of chylomicrons and are therefore under increased demand of extensive and specialized intracellular membrane trafficking. In conclusion, although the precise molecular mechanism by which mitochondrial dysfunction causes impaired transport and massive accumulation of

lipids in large lipid droplets in proximal enterocytes remains unclear at this stage, our results argue that an impairment in the processing and transport of pre-chylomicrons from the ER to the Golgi is a likely cause.

Minor points:

Fig 1E, the mitochondrial abnormalities in IEC need to be quantified. Number of mitochondria, size of mitochondria, cristae density, etc.

We have now quantified the mitochondrial abnormalities by assessing the cristae density in control and $DARS2^{tamIEC-KO}$ mice. We chose to perform the quantification in $DARS2^{tamIEC-KO}$ mice 7 days after the last tamoxifen injection instead of the $DARS2^{IEC-KO}$ pups, as we believe that the tamoxifen-inducible model gives a more elegant and accurate picture of the phenotype and it is easier to identify the distinct anatomic locations of the gut in adult mice compared to the pups. As expected, we found that most mitochondria in $DARS2$ -deficient enterocytes appear severely damaged with completely disorganised cristae, with a small fraction showing milder changes and an even smaller number appearing normal. In contrast, control mice showed mainly normal mitochondria in their enterocytes. These data are now presented in new Fig. 3c in the revised manuscript.

2. For the ED Fig. 3g, I recommend stratifying the genes by their pathways, e.g., de novo lipogenesis, beta oxidation, TAG synthesis, and lipolysis, that will be much clearer to the general audience.

Following the advice of the reviewer, we have grouped the genes according to their pathway functions in the respective figure.

3. line 203 - 205, “reduced 3H-triolein uptake in plasma and in all metabolically active organs except the stomach (Fig. 4b).” Uptake in plasma sounds really weird. I suggest to delete the word “uptake”.

Indeed, uptake is not the correct term to use for plasma. We have edited the text to correct this description.

4. The authors frequently use “inhibit”, e.g. line 28-29: “Here we show that mitochondrial dysfunction in enterocytes inhibits chylomicron production and the transport of dietary lipids to peripheral organs.” Line 34-35: “Feeding a fat-free diet inhibited the formation of LDs in $DARS2$ deficient enterocytes,...” I think a better wording would be “prevents” or “does not allow” or something like that.

We thank the reviewer for this suggestion. We have now edited and polished the text to avoid repetition.

5. Fig 5a. How do the authors distinguish a small size lipid droplet from a chylomicron? The data would be more robust if the authors quantify Golgi with chylomicrons, ER with pre-chylomicrons, and secreted/secretory chylomicrons.

By definition, CMs are found in the membranes of the secretory pathway (ER, Golgi, post-Golgi membrane carriers and secreted in the extracellular space), hence they are always contained within a lipid bilayer membrane carrier. In contrast, LDs are free in the cytoplasm and surrounded only by a phospholipid monolayer (D'Aquila et al., 2016; Demignot et al., 2014). Therefore, the identification or not of a membrane lipid bilayer around the neutral lipids would be the only way to distinguish CMs from LDs by conventional EM (see Figure 3 for

reviewers below). Although our EM samples were of good quality, the membrane preservation was not always possible, in particular in the case of $DARS2^{tamIEC-KO}$ mice where the cellular and membrane integrity is severely affected at 7 days after the end of tamoxifen treatment. For this reason, we consider that quantifying small sized LDs and CMs in particular would be inaccurate and practically impossible. Ideally, membrane preservation would likely be possible if one had performed High-Pressure Freezing and freeze substitution, but this technique unfortunately is not established in our EM facility. Although we cannot perform a reliable quantification of the different LD and CM profiles at the EM level, we provide several other lines of evidence that describe both qualitatively and quantitatively the dramatic conversion of CMs into LDs in proximal intestine IECs of $DARS2^{tamIEC-KO}$ mice. These include:

1. Time-dependent accumulation of LDs of increasing size.
2. Strongly increased expression of perilipins and particularly PLIN2 both at mRNA and protein level.
3. Significant reduction of CM secretion in portal plasma in metabolic tracing experiments in fasted $DARS2^{tamIEC-KO}$ mice, as compared to control mice.
4. Strongly decreased expression of ApoB48 compared to ApoB100 in plasma lipoproteins from $DARS2^{tamIEC-KO}$ mice compared to $Dars2^{fl/fl}$ littermates.

Figure 3 for reviewers. Electron microscopy image showing lipid bilayers around CMs (arrowheads) and LDs that are not surrounded by lipid bilayers (arrows). The mitochondria are indicated with m. Bars: 1 μ m.

6. Title: Mitochondria regulate dietary lipid processing in enterocytes. Showing regulation would require studies of intestinal epithelial cells with graded down- and up-regulation of mitochondrial function. I suggest to not refer to this as regulation.

We thank the reviewer for this suggestion. By using the term regulates we did not mean to imply this is a graded response, but we agree that this could be interpreted as such. We therefore changed the title to “Mitochondrial dysfunction abrogates dietary fat transport by enterocytes”.

7. I think the discussion on mitochondrial diseases is a bit questionable. Certainly, there are many mitochondrial diseases where malabsorption is a problem, e.g. MNGIE, some PolG mutations, Pearson's syndrome etc. However, as far as I know the role for mitochondrial dysfunction in intestinal epithelial cells is not well documented and other mechanisms may be more important, e.g. pancreatic insufficiency. I think this discussion point should be rewritten and perhaps it would be better to argue that the findings in the manuscript now warrants studies to determine whether a similar mechanism is also found in mitochondrial disease patients.

We thank the reviewer for this suggestion. We agree that it remains to be investigated whether the function of mitochondria identified in our study contributes to the intestinal complications and particularly malabsorption and failure to thrive observed in some patients with mitochondrial disease. We have edited the text to clarify this point and clearly state that our results warrant investigating whether lipid processing and transport defects as seen in our mice could be implicated in the intestinal manifestations observed in patients with mitochondrial diseases.

Referee #3 (Remarks to the Author): lipoproteins

This manuscript demonstrates that mice in which enterocyte mitochondrial function has been severely disrupted or ablated have accumulation of lipid droplets in enterocytes that is apparently due to impaired chylomicron secretion, apparently reflecting a broader defect in the ER-to-Golgi secretory pathway.

Specific comments:

1. It appears that mRNA analyses were not performed globally in the intestine using RNAseq but only selected genes were quantitate by qPCR. A more global approach to gene expression would complement the proteomic and lipidomic analyses.

We have now performed global gene expression analysis by RNA sequencing of proximal intestinal tissue of tamoxifen-inducible DARS2 knockout mice at 3 and 7 days after the last tamoxifen injection, which are now included in the revised manuscript (ED Fig. 3f, g). Analysis of intestinal RNAseq data 7 days after tamoxifen induction revealed a broad de-regulation of several pathways, including those regulating lipid metabolism. Indeed, GSEA of the RNAseq data identified fatty acid metabolism as one of the most significantly downregulated pathway in DARS2^{tamIEC-KO} intestines (ED Fig. 3f). In addition, over-representation analysis (ORA) of these RNAseq data revealed downregulation of genes under GO term categories such as "long-chain fatty acid metabolic process", "fatty acid metabolic process", "lipid modification", "lipid catabolic process", "fatty acid catabolic process" etc in the intestine of DARS2^{tamIEC-KO} mice (Figure 4 for reviewers). This was consistent with the results from the ORA analysis of the proteomic data that revealed a downregulation of the expression of proteins described by similar GO terms (ED Fig. 4d). However, these changes were not observed in RNAseq data from the intestines of mice 3 days after tamoxifen, a time point when severe mitochondrial dysfunction is already present. Instead, the major signature altered in the DARS2^{tamIEC-KO} mice at this early timepoint was an upregulation of the ATF4 gene signature indicative of activation of integrated stress responses, which was also strongly induced at 7 days (ED Fig. 3g). The induction of the ATF4 pathway indicative of the activation of the integrated stress response is consistent with previously published results in DARS2-deficient hearts (Kaspar et al., 2021)

and multiple other models of mitochondrial dysfunction (Bao et al., 2016; Kuhl et al., 2017; Quiros et al., 2017). Our interpretation of these results is that the observed deregulation of genes controlling lipid metabolism are not primary effects of DARS2 ablation but rather secondary consequences of the impaired dietary lipid processing and secretion resulting in massive fat accumulation in lipid droplets within enterocytes.

D7_TGvWT: Log2FC<-0.5 & p.adj<0.05

Figure 4 for reviewers. Over-representation analysis of RNAseq data from proximal small intestines of *DARS2^{tam1IEC-KO}* mice 7 days after tamoxifen induction revealed lipid metabolic processes among the most significantly downregulated pathways.

2. Does heterozygous deletion of DARS2 in enterocytes produce a milder form of mitochondrial dysfunction and/or influence dietary lipid handling? More broadly, is there a continuous relationship between enterocyte mitochondrial function and ability to secrete chylomicrons or is this phenotype only revealed when mitochondrial function is ablated?

The lipid accumulation phenotype develops only when DARS2, SDHA or COX10 are homozygously ablated in IECs. We have not observed intestinal pathology in mice with heterozygous knockout of these proteins in the intestinal epithelium, suggesting that haploinsufficiency is not enough to cause lipid accumulation in enterocytes under steady state conditions. This is not surprising as it has been shown in other cellular systems that heterozygous deficiency of these proteins did not cause measurable mitochondrial

dysfunction, probably because the haploinsufficiency often does not cause halving of protein levels. When it comes to essential mitochondrial proteins, haploinsufficiency is commonly compensated on the level of translation and/or increased protein activity, hence often leads only to minor decrease at the level of protein function. For example, this has been shown for animals carrying only one copy of TFAM or POLG (mitochondrial transcription factor A or mtDNA polymerase, respectively) that showed only minor effects on the mtDNA levels (Hance et al., 2005; Larsson et al., 1998). We have also analysed hearts of heterozygous Dars2 knockout (Dars2^{+/-}) mice and could not detect any effect on the protein synthesis rates or OXPHOS complexes (unpublished results).

3. Do the enterocytes lacking DARS2 (or other genes resulting in loss of mitochondrial function) display evidence of ER stress?

Mitochondrial OXPHOS dysfunction has been shown to induce the mitochondrial branch of the Integrated Stress Response (mitoISR), which is activated through DELE1-OMA1-HRI axis (Fessler et al., 2020; Guo et al., 2020). Therefore, there is no evidence that OXPHOS deficiency directly causes ER stress, although secondary and late effects on ER homeostasis cannot be completely excluded, especially in the terminal stages of the dysfunction. Consistent with the activation of stress responses previously described in heart-, neuron- and macrophage-specific DARS2 deficient animals (Kaspar *et al.*, 2021; Rumyantseva et al., 2020; Willenborg et al., 2021), we have observed a strong signature of the mitoISR in RNAseq data from the intestines of DARS2^{tamIEC-KO} mice, illustrated by the ATF4 signature being the most significantly induced pathway both in the proteomics and RNAseq data (ED Fig. 3d-g). Importantly, ATF4 pathway activation is already observed at 3 days after tamoxifen induction, before the appearance of lipid droplets and Golgi defects, suggesting that induction of mitoISR is an early event. However, although enterocytes lacking DARS2 show induction of the ISR signature, a large amount of literature on ISR and ER stress responses suggests this cannot be the cause of the lipid accumulation observed in DARS2-deficient enterocytes. Specifically, the role of ISR from the perspective of ER stress in the intestinal epithelium has been extensively studied in mice lacking critical components regulating this and other branches of ER stress responses in IECs (e.g. ATF4, p-eIF2a, CHOP, IRE1b, IRE1a, GRP78, XBP1), however, none of these mouse models showed evidence of lipid accumulation in enterocytes and/or lipid transport defects (Bertolotti et al., 2001; Cao et al., 2014; Hu et al., 2019; Kaser et al., 2008; Liu et al., 2018; van Lidth de Jeude et al., 2017; Zhang et al., 2015). Therefore, studies in genetic mouse models demonstrated that induction of ER stress does not lead to lipid accumulation arguing that increased ER stress or (mito)ISR cannot be the primary cause of the impaired lipid processing and transport observed in DARS2-deficient enterocytes.

4. The plasma lipid concentrations of triglycerides and cholesterol should be provided for each mouse model reported in this manuscript as additional evidence for systemic intestinal lipid malabsorption.

The plasma lipid concentrations of triglycerides and cholesterol for both the conditional and the inducible mouse models are now included in the revised manuscript in the following figures (Fig. 1i, 3g, 2c, 2h, ED Fig. 7f).

5. The methods state that 14C-triolein was used for the oral lipid tracer studies but the results refer to 3H-triolein. Was the tracer on the fatty acids or the glycerol in the labeled triolein? This is critically important to the interpretation of these results.

We thank the reviewer for this comment, which made us realise that we had made a mistake in the Methods section of the paper. For the postprandial oral glucose and fat tolerance test we used ³H-triolein (1.7 MBq/kg body weight) and ¹⁴C-deoxyglucose (1.4 MBq/kg body weight). For the experiment assessing chylomicron production by measuring accumulation of labelled lipids in the plasma, we used ³H-triolein (1.7 MBq/kg body weight) in combination with ¹⁴C-cholesterol (1.4 MBq/kg body weight). In both experiments, the tracer was on the fatty acids, not on the glycerol backbone. We have now corrected the Methods section in the revised manuscript.

6. Intestinal chylomicrons are secreted into intestinal lymph prior to accessing the blood. A more direct approach to physiologically assessing intestinal triglyceride secretion would be to measure TG mass and the orally-fed 3H-fatty acid tracer in the thoracic duct lymph from the mice.

We thank the reviewer for this suggestion. Our data obtained by measuring ³H-triolein in both the serum and peripheral tissues clearly showed that intestinal lipid transport to the periphery, which mainly occurs in the form of chylomicrons, is impaired in DARS2^{tamIEC-KO} mice. While we agree that measuring the lipid tracer in the thoracic duct lymph would provide a more elegant approach to directly assess intestinal lipid secretion, in our opinion, the potential gain in knowledge that could be achieved by this experiment will be incremental and will not provide a fundamentally different result from the studies we already performed. Particularly considering that this experiment is technically challenging as it involves complex surgical procedures that also cause extreme stress on the animals, we respectfully suggest that this is not an essential experiment to perform.

7. ApoB-48 is the major structural protein in chylomicrons produced by enterocytes and is required for chylomicron secretion but is not mentioned in the results. Was it detected in the proteomics studies? ApoB-48 should be assessed in the intestinal tissue by western blot and measured in the plasma (and lymph) by an immunoquantitative technique.

We did not detect APOB in the proteomic data. Following the advice of the reviewer, we have assessed APOB48 expression in primary enterocytes and in plasma from mice with tamoxifen-inducible ablation of DARS2 in IECs. As shown in the new data presented in Fig. 5a, b, immunoblot analysis of proximal enterocyte protein extracts revealed that ApoB48 expression was not reduced in DARS2-deficient IECs isolated from DARS2^{tamIEC-KO} mice 7 days after tamoxifen induction compared to control enterocytes (Fig, 5a). To measure APOB in VLDL and/or chylomicrons, we first isolated triglyceride-rich lipoproteins from plasma by ultracentrifugation, followed by immunoblot analysis. As shown in the new data presented in Fig. 5b, APOB48 levels were strongly reduced in DARS2^{tamIEC-KO} mice 7 days and to a lesser extent 5 days after tamoxifen induction compared to control mice. We should note that the anti-APOB antibodies recognize both APOB48, which is produced by enterocytes and APOB100 that is produced by the liver. APOB100 appears as the predominant form in plasma because liver-derived APOB100-containing VLDL have a longer plasma half-life compared to intestinal-derived ApoB48 particles and also because all available antibodies have higher affinity for APOB100 than APOB48. We have also attempted to detect APOB48 by ELISA, however, this was not possible as all available ELISAs for mouse APOB detect both APOB100

and APOB48, and therefore it is not feasible to specifically measure APOB48. Nevertheless, our immunoblot experiments clearly showed that while APOB48 expression is not reduced in enterocytes, the levels of APOB48 in plasma are strongly reduced in *DARS2*^{tamIEC-KO} mice, consistent with a reduced release of chylomicrons from the intestines of these animals.

8. The entire ER-to-Golgi transport secretory pathway appears to be disrupted. Other proteins known to be secreted by enterocytes should be measured as a physiological support for this broader disruption that is not just specific to chylomicron secretion per se.

Indeed, a major disruption of the Golgi is observed in proximal *DARS2*-deficient enterocytes 7-8 days after tamoxifen induction, suggesting that membrane trafficking along the entire secretory pathway should be compromised in these cells. In addition to the impaired chylomicron secretion, this is also suggested by the gradual fragmentation (starting already at 5 days after tamoxifen injection) and eventual dispersal of TGN38, which is itself a transmembrane protein that is constantly synthesized in the rough ER and undertakes the anterograde route to the trans-Golgi. Moreover, our immunostaining experiments revealed that plasma membrane localisation of E-Cadherin is strongly diminished in *DARS2*-deficient enterocytes, while its diffused cytoplasmic levels increase, providing experimental evidence that anterograde protein transport is impaired in these cells, leading to intracellular retention of a plasma membrane marker. Unfortunately, we cannot assess overall protein secretion looking at a secreted marker in *DARS2*-deficient enterocytes *in vivo* due to technical limitations. As we have not succeeded in establishing a cellular system recapitulating the phenotype, currently we have no means to examine broader protein secretion by *DARS2*-deficient enterocytes. Nevertheless, our results from the TGN38 and E-cadherin immunostaining experiments support that *DARS2*-deficient proximal enterocytes exhibit a broad disruption of the ER-to-Golgi secretory pathway and not a specific impairment of chylomicron transport. However, as discussed in our response to comment #4 from referee #2, mitochondrial dysfunction does not directly cause a general disruption of ER-to-Golgi transport but this defect is linked to the abrogation of dietary lipid processing and transport, as it only occurs in proximal enterocytes primarily responsible for dietary lipid transport within chylomicrons and requires the presence of fat in the diet. Collectively, these findings argue that an impairment of chylomicron processing and transport is likely the primary defect, which then causes a more general secretory pathway disruption as a secondary effect by interfering with intracellular membrane transport that is required for sustaining the Golgi network organisation.

9. What is known about the energy requirements of the ER-to-Golgi transport secretory pathway? One would assume that this is the major issue in the disruption of this pathway.

The secretory pathway does require energy at different steps, including the assembly, transport and fusion of COPII vesicles but also trafficking of vesicles from the Golgi and fusion with the plasma membrane. However, our metabolomic analysis showed that ATP levels were not substantially reduced in *DARS2*-deficient enterocytes as indicated by the fact that enterocytes from *DARS2*^{tamIEC-KO} mice showed similar ATP/ADP ratio with those from control mice, most likely because these cells switch to glycolysis for ATP production as supported by increased levels of intermediates in glycolysis (ED. Fig 4f, also see Figure 5 for reviewers below). Based on these results, it is unlikely that the impaired ER-to-Golgi transport is directly

caused by an energy defect. This is further supported by our findings that the Golgi system is not severely affected in the distal small intestine of mice with inducible DARS2 deficiency 8 days after tamoxifen induction, in contrast to the strong Golgi disorganisation observed in proximal enterocytes of the same mice. Since both proximal and distal enterocytes lack DARS2 and show severe respiratory chain deficiency, if energy deficiency would be the reason for Golgi dispersal this should happen in both tissues. Moreover, our new results obtained by the analysis of mice 5 days after tamoxifen injection revealed that the first signs of Golgi dispersal were observed just before proximal enterocytes started showing the formation of small lipid droplets (Figure 5d). Together, these results showed that the Golgi defect is only induced in enterocytes specializing in the processing and secretion of dietary lipids, suggesting that it is coupled to an impairment in dietary lipid processing and transport in the form of chylomicrons.

Figure 5 for reviewers: Levels of glycolysis intermediates and ATP/ADP ratio measured by metabolomic analysis of isolated primary enterocytes from proximal small intestines of *DARS2^{tam1EC-KO}* and control mice 8 days after tamoxifen induction.

10. What is known about the effects of impaired or ablated mitochondrial function on the protein secretory pathway in other cell types, such as hepatocytes? Using the DARS2 floxed mice to ablate mitochondrial function in hepatocytes in mice and assessing the impact on LD accumulation and hepatocyte VLDL secretion would substantially enhance this report by either showing specificity of this finding for the enterocyte or broader applicability of this finding to hepatic lipid secretion as well.

Unfortunately, we have not studied the function of DARS2 in hepatocytes. However, the role of COX10 in the liver has been studied in mice with hepatocyte-specific COX10 knockout (Diaz *et al.*, 2008). Indeed, this study found that lipids and primarily triglycerides accumulated in the COX10-deficient hepatocytes (Diaz *et al.*, 2008). Although in that study the role of COX10 for VLDL secretion was not studied, these results certainly reinforce the concept that mitochondria play an important role in lipid metabolism. However, we are not sure how strong the correlation of the findings in hepatocytes is with our own study in enterocytes. Although it would be very interesting to perform a detailed side by side comparison of the lipid accumulation phenotypes in the liver and proximal small intestine, we respectfully suggest that these are outside the scope of the current manuscript, also due to the extensive amount of time it would take for us to generate and study the hepatocyte-specific knockouts.

11. The authors suggest that their findings may help explain some of the gastrointestinal problems in some patients with mitochondrial disorders. If so, one might expect evidence for lipid malabsorption, including LD accumulation in the small intestine and reduced plasma lipids—is that the case in these patients?

As discussed also above in our response to comment #7 of reviewer 2, our results in mouse models showing that mitochondrial dysfunction causes severe impairment of dietary lipid transport by enterocytes warrant further studies to address whether human patients with mitochondrial mutations show similar defects that could contribute to the pathology and particularly the gastrointestinal complications and failure to thrive observed in a subset of the patients. We do not have access to human patients and we could not find in the literature data specifically addressing dietary lipid processing and transport in such patients. Therefore, at this stage we cannot provide an answer to this question. While we agree that such studies need to be performed, we respectfully suggest that these are outside the scope of the current manuscript and should be pursued in follow up studies.

Referee #4 (Remarks to the Author): cholesterol metabolism

In this manuscript the authors reported that mitochondrial dysfunction induced by DARS2 knockout impaired chylomicron maturation and affected gut lipid absorption capacity. The authors confirmed the findings in two more murine models lacking key proteins for mitochondrial function (SDH and COX10A). They document massive accumulation of dietary fat (TAG) in the enterocytes from the knockout models.

Although the phenotypes are extensively described in the manuscript, a clear explanation of the molecular mechanism underlying these effects is absent. The KO mice look very compromised, with severe impairment of intestinal morphology (villi are disorganized). Also, both labelled triolein and glucose uptake is impaired, suggesting a generic dysfunction of nutrient absorption (do the mice have diarrhea?), and not a lipid-specific signature of the phenotype.

The deep impairment of the mitochondrial function may be responsible for a status of energetic depletion, that prevents proliferation (as reported) and possibly maturation of functional enterocytes. Cells can be just “exhausted”, without sufficient ATP to sustain physiological processes, including the transfer of pre-chylomicrons from ER to Golgi for maturation. There are evidences suggesting this might be an ATP dependent process (<https://doi.org/10.1242/jcs.00215>).

Over the manuscript falls short of providing new conceptual insight or specific mechanistic links between mitochondria and dietary lipid absorption.

Comments:

1) Gene expression in DARS2 KO mice is inconsistent: some genes are upregulated while some others are downregulated. Do they have an explanation for the reduced expression of PPAR α and some of the targets?

Indeed, although we observed an overall downregulation of genes regulating lipid metabolism there are some genes that are upregulated. This is true for the qPCR data obtained from analysis of RNA from intestines of 7-day old pups, but also for the RNAseq data we obtained

from the proximal intestine of tamoxifen-inducible DARS2 knockout mice. ORA analysis of the RNAseq data revealed significant downregulation ($\text{Log}_2\text{FC} < -0.5$ and $p.\text{adj} < 0.05$) of GO terms such as 'long-chain fatty acid metabolic process', 'fatty acid metabolic process', 'lipid modification', 'lipid catabolic process', 'fatty acid catabolic process' etc. in the intestine of $\text{DARS2}^{\text{tamIEC-KO}}$ compared to $\text{Dars2}^{\text{fl/fl}}$ mice 7 days after tamoxifen (see Figure 4 for reviewers above). This was consistent with the results from the ORA analysis of the proteomic data that revealed a downregulation of the expression of proteins described by similar GO terms (ED Fig 4d). However, these changes were not observed in RNAseq data from the intestines of mice 3 days after tamoxifen, a time point when the major signature altered in the $\text{DARS2}^{\text{tamIEC-KO}}$ mice was an upregulation of ATF4 target genes indicative of an activated integrated stress response (ED Fig. 3g and Supplementary Data Table 2). Our interpretation of these results is that the observed changes in gene expression are not primary effects of DARS2 ablation but rather secondary consequences of the severe phenotype developing in the mice because of the abrogation of dietary lipid transport and the massive accumulation of lipids in lipid droplets. Given the complex metabolic alterations taking place in these cells, it is very difficult to provide a clear explanation why some genes are reduced and others are not, however, as discussed above we interpret these changes as secondary effects and not causal to the observed phenotype.

2) The nutritional status of the mice is not reported at the time of sacrifice. This is particularly relevant for EM experiment when looking at the presence of the chylomicrons. Such an experiment should be performed in response to a lipid load (feeding with high fat diet or oil gavage).

We thank the reviewer for raising this issue, which gives us the opportunity to clarify one aspect that was not sufficiently stressed in the manuscript. All our studies were performed in mice fed *ad libitum* a normal chow diet (except the studies with fat-free diet). The $\text{DARS2}^{\text{tamIEC-KO}}$ mice were always sacrificed side-by-side and at the same time with their co-housed littermate $\text{Dars2}^{\text{fl/fl}}$ controls (not expressing the villinCreERT2 transgene) that also received tamoxifen. Under these conditions, 100% of the $\text{DARS2}^{\text{tamIEC-KO}}$ mice sacrificed 7 or 8 days after tamoxifen induction ($n=33$) showed massive accumulation of lipid droplets in proximal enterocytes, while none of the controls ($n=33$) showed this phenotype. We believe that this aspect needs to be highlighted, as the massive lipid accumulation within large lipid droplets in DARS2-deficient enterocytes occurred under a normal chow diet that contains only a low amount of fat (3.4%), and not under a lipid load induced by feeding a high fat diet, which is often used to induce lipid overloading in mice with milder phenotypes. Indeed, we agree that a high-fat diet would massively exacerbate the phenotype, but we do not see how this experiment would add to what our experiments already show. In fact, high-fat diet feeding is often used to trigger a phenotype in mice that under normal chow diet show no or only mild metabolic abnormalities and in these cases this is a legitimate approach. However, the main aim of our manuscript is to demonstrate that mitochondrial dysfunction causes a massive impairment of dietary fat processing in mice **fed a normal chow diet under normal conditions** without an excessive lipid load. Our data clearly demonstrate that DARS2-deficient enterocytes contain massive lipid droplets and lack ultrastructural signs of chylomicron production, while enterocytes in their littermate control mice do not contain lipid droplets and show clear evidence of chylomicron presence and release ($n=4$). This is in line with a previous study showing that mice with tamoxifen-inducible IEC-specific knockout of *Mttp* in enterocytes (Iqbal *et al.*, 2013) caused defective CM secretion and marked lipid accumulation in proximal enterocytes in mice fed with normal chow diet. Therefore, while we

agree with the reviewer that feeding a high-fat diet will exacerbate the phenotype, we respectfully suggest that this experiment would not provide additional conceptual or mechanistic insight in addition to what we already know from mice fed a normal chow diet to justify performing such studies.

3) When mice are fed fat free diet, there is no lipid accumulation. However, these mice show impaired crypt cells proliferation and dramatic weight loss. This seems to indicate that lethality is NOT specifically related to dietary lipid accumulation but rather to an overall unhealthy status of the mice.

a. Another explanation for the phenotype may stem from enterocyte death from mitochondrial dysfunction, thereby resulting in blunted villi. This would lead to malabsorption in carbohydrates, protein, and fats. It would be helpful to know if the KO mice on a fat free diet had diarrhea. It is also unknown if the KO mice on a fat free diet survived and were completely normal or if they continued to demonstrate defects despite the absence of fat. If the KO mice on a fat free diet did not survive, it is likely the phenotype is just due to a sick mouse.

As discussed above in our response to a similar comment from reviewer 1, our studies showed that feeding a fat-free diet strongly prevented lipid accumulation in enterocytes of $DARS2^{tamIEC-KO}$ mice, with these animals showing only a few very small lipid droplets in proximal enterocytes 7 days after tamoxifen induction, in contrast to mice on chow diet that exhibited massive lipid accumulation in large lipid droplets at this stage (Figure 4a). However, $DARS2^{tamIEC-KO}$ mice on fat-free diet also showed progressive weight loss after tamoxifen inducible $DARS2$ ablation, which necessitated their culling according to the animal experiment protocol (ED. Fig. 7b). In most of our experiments using mice on normal chow diet we sacrificed the mice at days 7 or 8 after the last tamoxifen injection as this is the time point when most mice reached about 20% weight loss. We used this time point for the analysis of the mice on fat-free diet for consistency and in order to be able to compare the results with the normal chow diet. However, in one pilot experiment we induced $DARS2$ ablation in a small group of mice on fat-free diet, then we followed the mice and culled them when they reached 20% loss of body weight. In this experiment, we had to sacrifice the mice at day 8 (1 mouse), day 9 (2 mice) and day 10 (3 mice). Therefore, fat-free diet feeding delayed by about two days but ultimately could not prevent the severe loss of body weight necessitating the culling of the mice upon tamoxifen-inducible $DARS2$ ablation in IECs. These results showed that lipid accumulation exacerbates but does not appear to be the primary cause of weight loss and subsequent death of the mice.

As also discussed in response to a similar comment from reviewer 1, we have indeed extensively studied the possible involvement of cell death in causing the intestinal pathology of these animals. In fact, we started this project aiming to study the role of mitochondrial function in IEC death and intestinal inflammation, as cell death and its role in inflammation is the main focus of our laboratory. However, to our surprise, we found no evidence of increased IEC death in the intestines of either the constitutive $DARS2^{IEC-KO}$ or tamoxifen-inducible $DARS2^{tamIEC-KO}$ mice, showing that $DARS2$ deficiency does not cause increased death of enterocytes (Data now included in ED Fig. 1e and ED Fig. 5b, d of the revised manuscript). In addition, to assess the possible role of apoptosis and necroptosis in the development of the intestinal pathology we have generated $DARS2^{IEC-KO}$ mice that simultaneously lack the apoptotic caspase-8 in IECs, as well as MLKL, the protein executing necroptosis, in all cells ($DARS2^{IEC-KO} CASP8^{IEC-KO} Mkl1^{-/-}$ mice) and found that even combined inhibition of apoptosis and necroptosis did not prevent the intestinal pathology and the death of the mice (see **Figure**

1 for reviewers). Thus, DARS2 deficiency does not cause considerably increased cell death in the intestine arguing against cell death being a primary driver of the observed intestinal pathology and particularly the massive lipid accumulation in enterocytes of DARS2^{IEC-KO} mice. Furthermore, during the last 15 years we have studied in our lab a large number of mouse models with IEC-specific knockouts of key pro-survival proteins, which sensitized IECs to different types of cell death including apoptosis, necroptosis and pyroptosis and caused intestinal inflammation (Dannappel *et al.*, 2014; Eftychi *et al.*, 2019; Nenci *et al.*, 2007; Schwarzer *et al.*, 2020; Vlantis *et al.*, 2016a; Vlantis *et al.*, 2016b; Welz *et al.*, 2011). Whereas all these models showed strongly increased numbers of dying IECs causing severe inflammatory pathologies in the gut, we, but also other investigators performing similar studies, did not detect lipid accumulation in enterocytes in any of these mouse models with increased IEC death, arguing that death of epithelial cells does not cause lipid accumulation. Therefore, we can exclude that the massive accumulation of lipids in large lipid droplets in enterocytes from DARS2^{IEC-KO} mice is a secondary consequence of cell death.

In addition to cell death, we were also curious whether the strong inhibition of epithelial proliferation caused by DARS2 ablation could contribute to the lipid accumulation and the lethal phenotype. However, extensive data in the literature demonstrate that inhibition of IEC proliferation does not result in lipid accumulation. As an example, we refer to a manuscript published by Clevers and colleagues (van Es *et al.*, 2012), where they reported that tamoxifen-inducible ablation of Tcf4 in IECs (using the same VillinCreERT2 mouse we used in our study) caused complete inhibition of epithelial cell proliferation and resulted in the death of the mice 9 days after tamoxifen induction, which is approximately the time that DARS2^{tamIEC-KO} mice need to be sacrificed due to weight loss. Importantly, despite the complete inhibition of epithelial proliferation and the severe lethal phenotype, mice with IEC-specific Tcf4 knockout did not show any lipid accumulation in enterocytes (van Es *et al.*, 2012). These findings argue that the complete inhibition of epithelial cell proliferation and not the lipid accumulation is the most likely cause of the lethal phenotype of DARS2^{tamIEC-KO} mice. Therefore, mitochondrial dysfunction induced by ablation of DARS2, SDHA or COX10 clearly has important pathological consequences in intestinal epithelial cells **in addition** to the lipid processing defect, as for example the complete impairment of IEC proliferation. In fact, we did not mean to imply that lipid accumulation was the only consequence of mitochondrial dysfunction in enterocytes causing the death of the mice. We have now edited the text to clarify this point.

The reviewer also asks if the mice under fat-free diet had diarrhoea. While the DARS2^{IEC-KO} pups did show watery stool and diarrhoea, we did not observe diarrhoea in mice after tamoxifen-inducible knockout of DARS2 even when they were fed a normal chow diet. Therefore, we would not expect the fat-free diet to change that. It is indeed interesting that DARS2^{tamIEC-KO} mice do not show diarrhoea, but we believe this is likely related to the low content of fat in the normal chow diet (3.7%), which is not sufficient to lead to massive lipid accumulation in the stool even if its absorption is reduced. This is different in the pups that feed on milk with very high fat content. Another factor that potentially contributes to the lack of diarrhoea is the complete suppression of epithelial cell proliferation in these animals, which necessarily leads to a slower turnover of enterocytes and thus prevents the sloughing of lipid-laden enterocytes into the lumen.

Collectively, whereas lipid accumulation is not the primary cause of the severe weight loss and ultimate death of the mice, our results together with numerous studies in the literature

provide compelling evidence that the dietary lipid processing defect leading to massive accumulation of lipids in enterocytes lacking DARS2, but also SDHA and COX10, is not a secondary consequence caused by epithelial or organismal death, or by inhibition of epithelial proliferation, but rather a primary and direct effect of mitochondrial dysfunction. We consider this to be the most interesting and novel aspect of our work as it reveals an important but previously unrecognised role of mitochondria in the processing and transport of dietary lipids by proximal enterocytes.

4) The ³H triolein absorption study highlighted lower counts in duodenum of KO mice: these suggest that enterocytes are taking up less lipid from the lumen, which might explain itself the reduced counts in the plasma.

The reviewer correctly points out that our experiments showed a moderate reduction of ³H-triolein counts in the proximal small intestine of DARS2^{tamIEC-KO} mice compared to controls, which could also contribute to the reduced counts in plasma (Fig. 4c; please note that in the revised manuscript now present the duodenum and jejunum data together combined under “proximal SI”, because we realised that it is difficult to clearly separate these two segments of the intestine during dissection, which may lead to differences between individual mice). However, as discussed also above in our response to a similar comment of reviewer 2, we do not consider the reduced ³H-triolein counts in the intestine as the primary cause of the impaired transport of dietary lipids to the circulation. We believe that the moderately reduced uptake of ³H-triolein in the intestine could be related to compensatory mechanisms activated by the cells, which upon sensing that lipids start to accumulate may try to slow down further lipid uptake to prevent a lipid overload. A small reduction in intestinal uptake of ³H-triolein was also observed in our new metabolic tracing experiments that were performed in mice 5 days after tamoxifen induction (ED. Fig. 9c), suggesting that cells may attempt to suppress lipid uptake from the lumen early on in an effort to prevent lipid overload. We believe that there are strong arguments supporting that the primary defect in DARS2-deficient enterocytes is an impairment of processing and secretion of dietary lipids and not their uptake. First, our data showing that feeding the mice with a fat-free diet strongly prevented the formation of large lipid droplets in DARS2-deficient enterocytes demonstrated that the lipids accumulating in these cells are primarily derived from dietary fat, which argues that food-derived lipids are taken up by the enterocytes and stored in lipid droplets instead of being secreted in the form of chylomicrons. Second, as shown by the massive accumulation of dietary lipids in lipid droplets at 7-8 days after tamoxifen induction, on balance more lipids are taken up than are secreted. Third, comparing the relatively modest reduction in uptake (about 60% of that in control mice) by the intestine to the strongly diminished secretion to the circulation (about 8-fold less ³H-triolein in plasma, Fig. 4b) clearly supports a severe defect in secretion with only a mild impairment in uptake. Therefore, together with our findings that DARS2-deficient enterocytes lack mature chylomicrons and show severe disorganisation of the Golgi network, these findings support that the primary defect lies in the processing and secretion of dietary lipids and not in their uptake.

5) A better characterization of chylomicrons is required: all the conclusions are taken based on EM. The authors should isolate chylomicrons from plasma and measure ApoB in both plasma and isolated chylomicrons.

Following the advice of the reviewer, we have assessed APOB48 expression in isolated triglyceride-rich lipoproteins from plasma by ultracentrifugation (it is impossible to distinguish chylomicrons from VLDL particles). To distinguish between APOB100, which is produced by

the liver and is found in VLDL particles, from APOB48, which is produced by enterocytes and is found in chylomicrons, we used immunoblotting that can separate the two isoforms due to their different molecular weight. As shown in the new data presented in Fig. 5b, APOB48 levels were strongly reduced in triglyceride-rich lipoprotein particles isolated from DARS2^{tamIEC-KO} mice 7 days after tamoxifen induction compared to control mice. APOB48 levels were also moderately lessened at 5 days after tamoxifen indicative of the progressive defect in chylomicron production. APOB100 appears as the predominant form in plasma because liver-derived APOB100-containing VLDL have a longer plasma half-life compared to intestinal-derived ApoB48 particles and also because all available antibodies have higher affinity for APOB100 than APOB48. We have also attempted to detect APOB48 by ELISA, however, this was not possible as all available ELISAs detect both APOB100 and APOB48, and therefore it was not possible to specifically measure APOB48.

6) It appears that the phenotype may be a secondary effect of ATP depletion from impaired mitochondria. It would be more convincing if the authors showed that intracellular trafficking in general was not affected and that chylomicron transport was specifically affected. Otherwise, if all trafficking is impaired, then this is likely a result of impaired energy production.

As discussed also above in response to the general assessment of the reviewer, as well as to a similar comment from reviewer #3 on energy depletion as a potential cause of cellular exhaustion causing the lipid accumulation, our metabolomic analysis showed that ATP levels were not substantially reduced in DARS2-deficient enterocytes as indicated by the fact that enterocytes from DARS2^{tamIEC-KO} mice showed similar ATP/ADP ratio with those from control mice, most likely because these cells switch to glycolysis for ATP production as supported by increased levels of intermediates in glycolysis (ED. Fig 4f, also see Figure 5 for reviewers above). Moreover, as discussed in our response to comment #8 from reviewer #3, a major disruption of the Golgi is observed in proximal DARS2-deficient enterocytes 8 days after tamoxifen induction, suggesting that membrane trafficking along the entire secretory pathway should be compromised in these cells. In addition to the impaired chylomicron secretion, this is also supported by the gradual fragmentation (starting already at 5 days after tamoxifen injection) and eventual dispersal of the Golgi, as indicated by immunostaining for TGN38, which is itself a transmembrane protein that is constantly synthesized in the rough ER and undertakes the anterograde route to the trans-Golgi. Moreover, our immunostaining experiments revealed that 8 days after tamoxifen induction the plasma membrane localisation of E-Cadherin is strongly diminished in DARS2-deficient enterocytes, while its diffused cytoplasmic levels increase, providing experimental evidence that anterograde protein transport is impaired in these cells leading to intracellular retention of a plasma membrane marker. Therefore, our results from the TGN38 and E-cadherin immunostaining experiments support that DARS2-deficient proximal enterocytes exhibit a general disruption of the ER-to-Golgi secretory pathway and not a specific impairment of chylomicron transport at 8 days after tamoxifen. However, as discussed in our response to comment #4 from referee #2, mitochondrial dysfunction does not directly cause a general disruption of ER-to-Golgi transport but this defect is linked to the abrogation of dietary lipid processing and transport. This is shown by our findings that the Golgi system is not considerably affected in the distal small intestine of mice with inducible DARS2 deficiency 8 days after tamoxifen induction, in contrast to the strong Golgi disorganisation observed in proximal enterocytes of the same mice. Since both proximal and distal small intestinal epithelial cells lack DARS2 and show severe respiratory chain deficiency, if energy deficiency would be the reason for Golgi dispersal this

should happen in both tissues. Moreover, our new results obtained by the analysis of mice 5 days after tamoxifen induction revealed that the first signs of Golgi dispersal were observed just before proximal enterocytes started showing the formation of small lipid droplets (Figure 5d). Therefore, the Golgi defect is primarily induced in enterocytes specializing in lipid transport, required the presence of fat in the diet and precedes lipid droplet formation and enlargement. Together, these findings argue that mitochondrial dysfunction does not directly induce a general ER-to-Golgi transport defect in all cells, but that this is specifically coupled to an impairment in dietary lipid processing and transport in the form of chylomicrons.

7) It is still unclear why triglycerides are higher in the intestines of these knock-out mice. The authors are unable to differentiate between whether this is an effect of decreased chylomicron secretion or whether this is due to lipid accumulation from impaired fatty acid oxidation. It seems to be the later based on the increase in acylcarnitine. This was noted by the authors, but not elaborated on.

The reviewer raises an issue that was also raised by reviewer 2, specifically whether impaired fatty acid beta-oxidation could be the cause for the accumulation of lipids in enterocytes with severe mitochondrial dysfunction. Indeed, our metabolomic analysis revealed drastic accumulation of almost all species of long chain and very long chain acylcarnitines in DARS2 knockout IECs (ED Fig. 4e.), supporting that DARS2 deficiency cause severed impairment of fatty acid beta-oxidation. However, as discussed above in our response to reviewer #2, our experimental studies provide evidence showing that impaired fatty acid beta-oxidation cannot be the main cause of the massive lipid accumulation in lipid droplets in enterocytes from our mice. Specifically, if impaired lipid utilisation due to diminished fatty acid beta-oxidation were the primary cause for the lipid accumulation, then we would expect this to happen in both the proximal and distal small intestine and to be independent of the content of fat in the diet. The fact that lipid accumulation occurs specifically in proximal enterocytes, which are primarily responsible for lipid absorption and transport as opposed to distal enterocytes (please see also our response to the following comment below), argues that the accumulation is not caused by impaired lipid catabolism, but is rather the consequence of impaired transport. Moreover, our findings that feeding a fat-free diet suppressed the accumulation of lipids, provided experimental evidence that the lipids stored in lipid droplets originate mainly from dietary fat. We would also like to refer to a recent study showing that fatty acid oxidation in enterocytes utilises primarily lipids that are obtained from blood VLDL via the basolateral side and not dietary lipids absorbed by the lumen (Korbelius et al., 2019). In this study, mice lacking ATGL and CGI-58 in IECs were shown to accumulate lipids in lipid droplets within enterocytes, however, in contrast to our mice, these were not prevented by feeding a fat-free diet. Moreover, these mice did not show a defect in transporting dietary lipids to the circulation in the form of chylomicrons, again in contrast to our mice. These findings argue that the handling of dietary lipids is different from that of lipids reaching enterocytes from the blood, with the former primarily being transported to the circulation and the latter predominantly used for fatty acid beta-oxidation. Therefore, whereas inhibition of fatty acid beta-oxidation might also contribute to the accumulation of lipids to a small extent, our results show that impaired transport of dietary fat is the primary reason for the accumulation of lipids in DARS2-deficient proximal enterocytes.

8) If the proximal and distal small intestine have appropriate knockdown of DARS2, why was the distal small intestine not affected? The authors state “lipid accumulation in

IEC's depend on anatomic location". I am not sure I understand this statement since the distal small intestine can also secrete chylomicrons.

Indeed, DARS2 is efficiently deleted in epithelial cells in both the proximal and distal small intestine in our mice, resulting in severe OXPHOS deficiency (see COX/SDH staining in Fig. 3d and ED Fig. 5c). This is further supported by our findings that epithelial proliferation is completely suppressed in both the proximal and distal small intestine (see Fig. 3d and ED Fig. 5c). However, as shown in Fig 3e, massive lipid accumulation in large lipid droplets is observed exclusively in proximal enterocytes, whereas epithelial cells in the distal small intestine do not show any signs of lipid droplet formation. In fact, we believe this is one of the most interesting and informative results in our studies, as it demonstrates that lipids accumulate exclusively in those enterocytes that are primarily endowed with the task to absorb and transport dietary fat. We are somewhat puzzled by the comment of the reviewer that proximal and distal small intestine are both equally involved in mediating dietary lipid uptake and transport in the form of chylomicrons. It is our understanding that a large amount of literature clearly supports that dietary lipids are absorbed primarily by enterocytes in the proximal part of the small intestine (duodenum and jejunum), whereas the distal part of the small intestine (ileum) has a negligible contribution in fat absorption and transport (Jersild and Clayton, 1971; Nassir et al., 2007; Nauli et al., 2006). Moreover, our own data from the metabolic tracing experiments clearly showed that ³H-triolein was mainly absorbed by the proximal SI with the distal part playing a minor role (Figure 6 for reviewer below). Therefore, our findings that mitochondrial dysfunction causes massive lipid accumulation in proximal but not in distal enterocytes, together with our results in mice fed a fat-free diet, provide strong experimental evidence that lipid accumulation is directly related to a defect in the processing and transport of dietary fat.

Figure 6 for reviewers. Graph showing ³H-triolein uptake in different tissues of *DARS2^{tamIEC-KO}* and littermate *Dars2^{fl/fl}* mice 7 days after tamoxifen induction, presented as counts / mg of tissue.

References

Al Khazal, F., Holte, M.N., Bolon, B., White, T.A., LeBrasseur, N., and Maher, L.J., 3rd (2019). A conditional mouse model of complex II deficiency manifesting as Leigh-like syndrome. *FASEB journal : official publication of the Federation of American Societies for Experimental Biology* 33, 13189-13201. 10.1096/fj.201802655RR.

Bao, X.R., Ong, S.E., Goldberger, O., Peng, J., Sharma, R., Thompson, D.A., Vafai, S.B., Cox, A.G., Marutani, E., Ichinose, F., et al. (2016). Mitochondrial dysfunction remodels one-carbon metabolism in human cells. *Elife* 5. 10.7554/eLife.10575.

Bourgeron, T., Rustin, P., Chretien, D., Birch-Machin, M., Bourgeois, M., Viegas-Pequignot, E., Munnich, A., and Rotig, A. (1995). Mutation of a nuclear succinate dehydrogenase gene results in mitochondrial respiratory chain deficiency. *Nature Genetics* 11, 144-149. 10.1038/ng1095-144.

D'Aquila, T., Hung, Y.H., Carreiro, A., and Buhman, K.K. (2016). Recent discoveries on absorption of dietary fat: Presence, synthesis, and metabolism of cytoplasmic lipid droplets within enterocytes. *Biochim Biophys Acta* 1861, 730-747. 10.1016/j.bbali.2016.04.012.

Dannappel, M., Vlantis, K., Kumari, S., Polykratis, A., Kim, C., Wachsmuth, L., Eftychi, C., Lin, J., Corona, T., Hermance, N., et al. (2014). RIPK1 maintains epithelial homeostasis by inhibiting apoptosis and necroptosis. *Nature* 513, 90-94. 10.1038/nature13608.

Demignot, S., Beilstein, F., and Morel, E. (2014). Triglyceride-rich lipoproteins and cytosolic lipid droplets in enterocytes: key players in intestinal physiology and metabolic disorders. *Biochimie* 96, 48-55. 10.1016/j.biochi.2013.07.009.

Diaz, F., Garcia, S., Hernandez, D., Regev, A., Rebelo, A., Oca-Cossio, J., and Moraes, C.T. (2008). Pathophysiology and fate of hepatocytes in a mouse model of mitochondrial hepatopathies. *Gut* 57, 232-242. 10.1136/gut.2006.119180.

Eftychi, C., Schwarzer, R., Vlantis, K., Wachsmuth, L., Basic, M., Wagle, P., Neurath, M.F., Becker, C., Bleich, A., and Pasparakis, M. (2019). Temporally Distinct Functions of the Cytokines IL-12 and IL-23 Drive Chronic Colon Inflammation in Response to Intestinal Barrier Impairment. *Immunity* 51, 367-380 e364. 10.1016/j.immuni.2019.06.008.

Fessler, E., Eckl, E.M., Schmitt, S., Mancilla, I.A., Meyer-Bender, M.F., Hanf, M., Philippou-Massier, J., Krebs, S., Zischka, H., and Jae, L.T. (2020). A pathway coordinated by DELE1 relays mitochondrial stress to the cytosol. *Nature* 579, 433-437. 10.1038/s41586-020-2076-4.

Guo, X., Aviles, G., Liu, Y., Tian, R., Unger, B.A., Lin, Y.T., Wiita, A.P., Xu, K., Correia, M.A., and Kampmann, M. (2020). Mitochondrial stress is relayed to the cytosol by an OMA1-DELE1-HRI pathway. *Nature* 579, 427-432. 10.1038/s41586-020-2078-2.

Hance, N., Ekstrand, M.I., and Trifunovic, A. (2005). Mitochondrial DNA polymerase gamma is essential for mammalian embryogenesis. *Hum Mol Genet* 14, 1775-1783. 10.1093/hmg/ddi184.

Horvath, R., Abicht, A., Holinski-Feder, E., Laner, A., Gempel, K., Prokisch, H., Lochmuller, H., Klopstock, T., and Jaksch, M. (2006). Leigh syndrome caused by mutations in the flavoprotein (Fp) subunit of succinate dehydrogenase (SDHA). *J Neurol Neurosurg Psychiatry* 77, 74-76. 10.1136/jnnp.2005.067041.

Iqbal, J., Parks, J.S., and Hussain, M.M. (2013). Lipid absorption defects in intestine-specific microsomal triglyceride transfer protein and ATP-binding cassette transporter A1-deficient mice. *J Biol Chem* 288, 30432-30444. 10.1074/jbc.M113.501247.

Jersild, R.A., Jr., and Clayton, R.T. (1971). A comparison of the morphology of lipid absorption in the jejunum and ileum of the adult rat. *Am J Anat* 131, 481-503. 10.1002/aja.1001310408.

Kaspar, S., Oertlin, C., Szczepanowska, K., Kukat, A., Senft, K., Lucas, C., Brodesser, S., Hatzoglou, M., Larsson, O., Topisirovic, I., and Trifunovic, A. (2021). Adaptation to mitochondrial stress requires CHOP-directed tuning of ISR. *Sci Adv* 7. 10.1126/sciadv.abf0971.

Korbelius, M., Vujic, N., Sachdev, V., Obrowsky, S., Rainer, S., Gottschalk, B., Graier, W.F., and Kratky, D. (2019). ATGL/CGI-58-Dependent Hydrolysis of a Lipid Storage Pool in Murine Enterocytes. *Cell reports* 28, 1923-1934 e1924. 10.1016/j.celrep.2019.07.030.

Kuhl, I., Miranda, M., Atanassov, I., Kuznetsova, I., Hinze, Y., Mourier, A., Filipovska, A., and Larsson, N.G. (2017). Transcriptomic and proteomic landscape of mitochondrial dysfunction reveals secondary coenzyme Q deficiency in mammals. *Elife* 6. 10.7554/eLife.30952.

Larsson, N.G., Wang, J., Wilhelmsson, H., Oldfors, A., Rustin, P., Lewandoski, M., Barsh, G.S., and Clayton, D.A. (1998). Mitochondrial transcription factor A is necessary for mtDNA maintenance and embryogenesis in mice. *Nature Genetics* 18, 231-236. 10.1038/ng0398-231.

Nassir, F., Wilson, B., Han, X., Gross, R.W., and Abumrad, N.A. (2007). CD36 is important for fatty acid and cholesterol uptake by the proximal but not distal intestine. *J Biol Chem* 282, 19493-19501. 10.1074/jbc.M703330200.

Nauli, A.M., Nassir, F., Zheng, S., Yang, Q., Lo, C.M., Vonlehmden, S.B., Lee, D., Jandacek, R.J., Abumrad, N.A., and Tso, P. (2006). CD36 is important for chylomicron formation and secretion and may mediate cholesterol uptake in the proximal intestine. *Gastroenterology* 131, 1197-1207. 10.1053/j.gastro.2006.08.012.

Nenci, A., Becker, C., Wullaert, A., Gareus, R., van Loo, G., Danese, S., Huth, M., Nikolaev, A., Neufert, C., Madison, B., et al. (2007). Epithelial NEMO links innate immunity to chronic intestinal inflammation. *Nature* 446, 557-561. 10.1038/nature05698.

Okada, T., Miyashita, M., Fukuhara, J., Sugitani, M., Ueno, T., Samson-Bouma, M.E., and Aggerbeck, L.P. (2011). Anderson's disease/chylomicron retention disease in a Japanese patient with uniparental disomy 7 and a normal SAR1B gene protein coding sequence. *Orphanet J Rare Dis* 6, 78. 10.1186/1750-1172-6-78.

Parfait, B., Chretien, D., Rotig, A., Marsac, C., Munnich, A., and Rustin, P. (2000). Compound heterozygous mutations in the flavoprotein gene of the respiratory chain complex II in a patient with Leigh syndrome. *Hum Genet* 106, 236-243. 10.1007/s004390051033.

Quiros, P.M., Prado, M.A., Zamboni, N., D'Amico, D., Williams, R.W., Finley, D., Gygi, S.P., and Auwerx, J. (2017). Multi-omics analysis identifies ATF4 as a key regulator of the mitochondrial stress response in mammals. *J Cell Biol* 216, 2027-2045. 10.1083/jcb.201702058.

Rumyantseva, A., Motori, E., and Trifunovic, A. (2020). DARS2 is indispensable for Purkinje cell survival and protects against cerebellar ataxia. *Hum Mol Genet* 29, 2845-2854. 10.1093/hmg/ddaa176.

Schwarzer, R., Jiao, H., Wachsmuth, L., Tresch, A., and Pasparakis, M. (2020). FADD and Caspase-8 Regulate Gut Homeostasis and Inflammation by Controlling MLKL- and GSDMD-Mediated Death of Intestinal Epithelial Cells. *Immunity* 52, 978-993 e976. 10.1016/j.immuni.2020.04.002.

van Es, J.H., Haegebarth, A., Kujala, P., Itzkovitz, S., Koo, B.K., Boj, S.F., Korving, J., van den Born, M., van Oudenaarden, A., Robine, S., and Clevers, H. (2012). A critical role for the Wnt effector Tcf4 in adult intestinal homeostatic self-renewal. *Mol Cell Biol* 32, 1918-1927. 10.1128/MCB.06288-11.

Vlantis, K., Polykratis, A., Welz, P.S., van Loo, G., Pasparakis, M., and Wullaert, A. (2016a). TLR-independent anti-inflammatory function of intestinal epithelial TRAF6 signalling prevents DSS-induced colitis in mice. *Gut* 65, 935-943. 10.1136/gutjnl-2014-308323.

Vlantis, K., Wullaert, A., Polykratis, A., Kondylis, V., Dannappel, M., Schwarzer, R., Welz, P., Corona, T., Walczak, H., Weih, F., et al. (2016b). NEMO Prevents RIP Kinase 1-Mediated Epithelial Cell Death and Chronic Intestinal Inflammation by NF-kappaB-Dependent and -Independent Functions. *Immunity* 44, 553-567. 10.1016/j.immuni.2016.02.020.

Welz, P.S., Wullaert, A., Vlantis, K., Kondylis, V., Fernandez-Majada, V., Ermolaeva, M., Kirsch, P., Sterner-Kock, A., van Loo, G., and Pasparakis, M. (2011). FADD prevents RIP3-mediated epithelial cell necrosis and chronic intestinal inflammation. *Nature* 477, 330-334. 10.1038/nature10273.

Willenborg, S., Sanin, D.E., Jais, A., Ding, X., Ulas, T., Nuchel, J., Popovic, M., MacVicar, T., Langer, T., Schultze, J.L., et al. (2021). Mitochondrial metabolism coordinates stage-specific repair processes in macrophages during wound healing. *Cell Metab* 33, 2398-2414 e2399. 10.1016/j.cmet.2021.10.004.

Reviewer Reports on the First Revision:

Referees' comments:

Referee #1 (Remarks to the Author):

In this revised manuscript from Moschandrea et al., the authors present additional data to support the idea that mitochondrial dysfunction leads to chylomicron (CM) accumulation in the proximal intestine, raising an interesting possibility that mitochondria can regulate ER-golgi network to control secretion. Improvements of this revision include (1) Demonstration that cellular death is not the cause of CM accumulation; (2) Some specificity showing that the observed phenotype is not general enterocyte malfunction; (3) Additional molecular data characterizing general defects. This manuscript is technically impressive, employing multiple genetic, molecular, cell biological, and biochemical techniques. Data are clearly presented, and after reading the authors' responses, I have no concern in believing that CM accumulation and ER-Golgi defects are seen in these mice.

The remaining concern is whether mitochondrial dysfunction is indeed causal to the observed CM accumulation. My previous points raised the possibility that CM accumulation is secondary to cellular, organ, or organismal level. Here the authors convincingly argue that "cellular" death does not occur as a result of DARS deletion in the intestine (very nice), but it is still unclear whether CM accumulation is still secondary to failure at the organ, or organismal level. Unfortunately, fat-free diet does not rescue the organismal death as the authors have examined. To better dissect this, the authors need to find a condition in which mitochondrial function is severely compromised without causing gross organ or organismal damage, be it by analyzing these mice at a shorter-time point, employing an ex vivo model, enterocyte-specific deletion of Dars using Alpi-CreERT2, etc. While I understand that some of these models may not be feasible at the moment, it is hard to evaluate the claims without some direct evidence.

In this regard, Dars tamIEC-KO data presented in Extended Figure 6 need some attention. This acute model, even after 3 days of Dars depletion, does not exhibit CM accumulation. This is somewhat surprising to me because, given that the VilCreERT2 driver is extremely robust, the authors' definition of 3 day deletion is actually 8 days after the initial dose (meaning cells that responded to the first dose of Tamoxifen has already turned over as the entire epithelium turn over 5-7 days). The actual CM accumulation happens much later than the time of epithelial depletion of Dars, and it coincides with the time point that animals start to collapse. These mice clearly have stem cell and proliferation defects (KI67 staining), as well as a decrease in overall secretory Goblet cells/ enterocyte number as the authors point out (Alpi and PAS staining); alternate interpretations of the data would include: (1) Dars depletion in the ISC pool may result in immature enterocyte specification and formation, leading to immature TGN network causing CM accumulation; (2) CM accumulation happens as a result of secondary effects at the organismal level.

Referee #2 (Remarks to the Author):

I was truly surprised when I read the revised version of the manuscript. I actually suspected that the journal had provided me with the wrong version of the manuscript. After having checked with the editor at Nature there seems to no mix-up of files at the journal.

Here are some examples of requested changes that have not been implemented in the manuscript despite promises in the response letter:

Major point 4. There is no Fig 5d and ED Fig 10a

Minor point 1, Fig 1E. I see no new panel Fig 3c.

Minor point 2. Where is the stratification of pathways in ED Fig 3g?

Minor point 3 "uptake in plasma" is still there.

Minor point 4. I advised the authors to reword "inhibit" but this word still persists twice in the abstract.

Minor point 6, the authors promised to change the title, which has not been done.

Clearly something is wrong here. I do not think it is meaningful to give this an in-depth review as the response letter clearly is incongruent with the submitted manuscript file. It has certainly wasted my time to deal with this confusion. If the authors really submitted this file as the revised manuscript file, I think the paper should be rejected so that the authors are more careful next time.

Referee #3 (Remarks to the Author):

The authors have addressed many of the reviewers' comments. The revised manuscript is improved.

Specific comments:

1. The authors did not actually address the question of whether enterocytes lacking DARS2 (or other genes resulting in loss of mitochondrial function) display evidence of ER stress.
2. It is very quick to delete a gene in liver of floxed mice by injecting AAV-cre, a well-established method.

Referee #4 (Remarks to the Author):

The authors show that loss of intestinal DARS2 causes severe mitochondrial dysfunction, resulting in severe weight loss, failure to thrive, and an inability to survive past 4 weeks. Electron microscopy images show swollen mitochondria. Intestinal-specific DARS2-KO mice display decreased numbers of proliferating cells, goblet cells, and absorptive enterocytes, as well as a decrease in the expression of stem-cell markers. Large lipid droplets within DARS2-KO enterocytes were visualized 7 days after tamoxifen injection, which the authors attribute to a disruption in the trans-golgi network and subsequent decrease in chylomicron formation and secretion. Fatty acid and cholesterol absorption is decreased in intestinal specific DARS2-KO mice.

Overall, the data suggest that impairment of mitochondrial health non-specifically compromises many aspects of enterocyte function. Specific mechanistic connections to dietary lipid transport are not established.

In response to comment 1 and 3:

1. The data would be more convincing if the authors showed that mitochondrial dysfunction specifically led to lipid droplet accumulation and abrogation of dietary lipid transport, by showing that other energy-dependent pathways were not affected. The results do suggest that mitochondrial dysfunction results in a decrease in IEC and crypt proliferation, as well as cell death, leading to a global decrease in overall enterocyte function, including chylomicron production and secretion. The physiological significance of this observation is unclear, other than mitochondria are important for cell function.

In response to comment 6:

2. It is unclear how mitochondrial dysfunction is related to a disrupted golgi network and chylomicron formation. The authors state that golgi disorganization is not a direct consequence of

severe mitochondrial dysfunction, which makes golgi disorganization a poor explanation for why chylomicron formation appears to be inhibited in the setting of mitochondrial dysfunction. Furthermore, the authors need to show that the disruption in the golgi network is restricted to lipid absorption alone, by excluding that other secretory effects modulated by the golgi network are affected. It still remains possible that mitochondrial dysfunction leads to a disruption in the TGN. The authors do observe that TGN disorganization only occurs in the proximal small intestine and not the distal small intestine, which suggests that mitochondrial dysfunction may not be related to problems within the TGN network at all.

In response to comment 3:

2. The authors say that there was "no evidence of increased IEC death in the intestine of either the constitutive DARS2-IEC-KO or tamoxifen-inducible DARS2-tam-IEC-KO mice".

-The figures only show Ki67 staining, a marker of proliferation, which is absent from crypt cells of DARS2-IEC-KO intestines. There are no markers of cell death. Therefore, it is unclear whether mitochondrial dysfunction leads to enterocyte death, which compromises lipid absorption.

In response to comment 3:

3. All experiments were performed 7-8 days after tamoxifen injection, when IEC and crypt cell proliferation were impaired with highly abnormal appearing enterocytes and a decrease in villi to crypt ratio, signifying enterocyte cell death.

-Crypt cells contribute little to lipid absorption and produce a minimal amount of chylomicrons if any. Therefore, the decrease in the proliferation of crypt cells is unlikely to be due to lipid accumulation, the formation of lipid droplets, and an abrogation of dietary lipid transport, which the authors attribute as the cause for the severe phenotype of DARS2-KO mice. If this were secondary to an inability to transport lipids, crypt cells should not be affected. This makes the phenotype of impaired proliferation and cell death, more likely to result from mitochondrial dysfunction.

-Furthermore, no signs of lipid droplet accumulation were seen 3 days after tamoxifen injection though IEC proliferation was already impaired. This was also shown on DARS2-KO mice on a fat free diet that displayed a minimal amount of lipid droplet accumulation, even when IEC proliferation was impaired. This indicates that lipid droplet accumulation is unlikely to be the cause of the phenotype.

-Perhaps, the studies would have been more convincing if performed 3 days after tamoxifen injection, when efficient ablation of DARS2 was achieved and while the enterocytes were healthier.

4. There is no apparent clinical relevance for decreased triglyceride absorption in the setting of mitochondrial dysfunction.

Author Rebuttals to First Revision:

Response to reviewers

We would like to take this opportunity to thank the reviewers for their comments. However, after reading their responses, we were puzzled as it seemed as if they did not read the revised manuscript. Indeed, when we checked the journal online system we realised that the revised manuscript was not uploaded to the site, although the rebuttal that was submitted at the same time was correctly uploaded. Instead, the reviewers only had access to the old manuscript and figures. This made us realise that the judgment of the reviewers was based on the results included in the originally submitted manuscript from last year. We are not sure how this error occurred, as we did provide the correct file with the revised manuscript. We are sorry about this and we of course understand that this puzzled and confused the reviewers, however, we kindly ask the reviewers to understand that this was not our fault. Below, we provide detailed responses to the new comments of the reviewers, indicating the new data in the revised manuscript that address their concerns. We hope that when the reviewers read our revised manuscript they will agree with us that we have addressed their major concerns.

Here, we would like to briefly comment on one particular issue that seems to persist throughout the review process. It seems that some of the reviewers doubt whether mitochondrial dysfunction directly impairs dietary lipid processing in enterocytes and instead suggest that this could be an indirect and unspecific consequence of more “general” defects at the cellular, organ or organismal level. As alternative explanations the reviewers suggested that mitochondrial dysfunction could cause cell death, stem cell and proliferation defects, immature enterocyte specification, energy depletion, and more broadly ‘failure’ at the organ or organismal level, which could indirectly cause impaired processing and transport of dietary lipids. We have taken these comments at heart and have performed a number of experiments to address these concerns that are outlined in detail in our rebuttal submitted with the revised manuscript but also in our responses below. We have also discussed why the observed stem cell and proliferation defects observed in our mice could not cause indirectly the lipid processing defect, citing specific examples in the literature of mice with severe stem cell and proliferation defects that do not show lipid accumulation in enterocytes. Importantly, in the revised manuscript we provide evidence that at a time point when the mice are still healthy (5 days after the last tamoxifen-induction), *DARS2*^{tamiEC-KO} mice already show impaired dietary lipid transport, dispersal of the Golgi network and the formation of lipid droplets in enterocytes. In conclusion, we believe that our results demonstrate that mitochondrial dysfunction directly causes an impairment in the processing and transport of dietary lipids by proximal enterocytes resulting in their storage in large cytoplasmic lipid droplets.

As a more general comment, and with all respect to the reviewer’s opinion, the argument that dietary lipid accumulation in proximal enterocytes could be an indirect and unspecific consequence of “general” malfunction at the cellular, organ or organismal level is not supported by any evidence. There are tens or even hundreds of studies in the literature with mouse models with intestinal epithelial specific knockout of genes that caused severe pathologies ranging from cell death, inflammation, stem and proliferation defects etc. We have extensively searched the literature to find examples where such ‘unspecific’ defects caused dietary lipid accumulation in proximal enterocytes, and found none. We may have of course missed such studies and would appreciate it if the reviewers could point such examples to us. Instead, we found only a handful of examples in the literature of mouse models that show lipid accumulation in lipid droplets in enterocytes, and all these relate to genes important for chylomicron production, trafficking and release such as *ApoB* (Young et al., 1995), *Mttp* (Iqbal et al., 2013), *Sar1b* (Auclair et al., 2021) and *DENND5B* (Gordon et al., 2019). Therefore,

this phenotype is highly specific and relates directly to defects in the processing and trafficking of dietary lipids and could not be caused by a general failure at the cellular, organ or organismal level. We hope that, after having read in detail our point by point response to their comments and, importantly, the revised manuscript, they will agree with us that our studies reveal a novel, important and hitherto unacknowledged function of mitochondria in enterocytes that is important for the processing and transport of dietary fat.

Point by point responses to the reviewer comments

Referee #1 (Remarks to the Author):

In this revised manuscript from Moschandrea et al., the authors present additional data to support the idea that mitochondrial dysfunction leads to chylomicron (CM) accumulation in the proximal intestine, raising an interesting possibility that mitochondria can regulate ER-golgi network to control secretion. Improvements of this revision include (1) Demonstration that cellular death is not the cause of CM accumulation; (2) Some specificity showing that the observed phenotype is not general enterocyte malfunction; (3) Additional molecular data characterizing general defects. This manuscript is technically impressive, employing multiple genetic, molecular, cell biological, and biochemical techniques. Data are clearly presented, and after reading the authors' responses, I have no concern in believing that CM accumulation and ER-Golgi defects are seen in these mice.

The remaining concern is whether mitochondrial dysfunction is indeed causal to the observed CM accumulation. My previous points raised the possibility that CM accumulation is secondary to cellular, organ, or organismal level. Here the authors convincingly argue that "cellular" death does not occur as a result of DARS deletion in the intestine (very nice), but it is still unclear whether CM accumulation is still secondary to failure at the organ, or organismal level. Unfortunately, fat-free diet does not rescue the organismal death as the authors have examined. To better dissect this, the authors need to find a condition in which mitochondrial function is severely compromised without causing gross organ or organismal damage, be it by analyzing these mice at a shorter-time point, employing an ex vivo model, enterocyte-specific deletion of Dars using Alpi-CreERT2, etc. While I understand that some of these models may not be feasible at the moment, it is hard to evaluate the claims without some direct evidence.

In this regard, Dars tamIEC-KO data presented in Extended Figure 6 need some attention. This acute model, even after 3 days of Dars depletion, does not exhibit CM accumulation. This is somewhat surprising to me because, given that the VilCreERT2 driver is extremely robust, the authors' definition of 3 day deletion is actually 8 days after the initial dose (meaning cells that responded to the first dose of Tamoxifen has already turned over as the entire epithelium turn over 5-7 days). The actual CM accumulation happens much later than the time of epithelial depletion of Dars, and it coincides with the time point that animals start to collapse. These mice clearly have stem cell and proliferation defects (KI67 staining), as well as a decrease in overall secretory Goblet cells/ enterocyte number as the authors point out (Alpi and PAS staining); alternate interpretations of the data would include: (1) Dars depletion in the ISC pool may result in immature enterocyte specification and formation, leading to immature TGN network causing CM accumulation; (2) CM accumulation happens as a result of secondary effects at the organismal level.

We are glad to read that the reviewer appreciated the comprehensive and 'technically impressive' according to the reviewer study we have undertaken to investigate the

consequences of mitochondrial dysfunction to the intestinal epithelium. The remaining concern of the reviewer is whether mitochondrial dysfunction is causal to the observed CM accumulation, or if the latter is secondary to failure at the organ or organismal level. Specifically, the reviewer was concerned that lipid accumulation in lipid droplets and impaired CM formation and transport was only observed at a late time point when the mice start to collapse (7 days after the last tamoxifen injection) but not at earlier time points, although at this stage DARS2 is efficiently deleted and a strong proliferation defect is already observed. As possible alternative explanations, the reviewer wonders whether the crypt stem cell and proliferation defects observed early on in DARS2^{tamIEC-KO} mice might result in immature enterocyte specification leading to immature TGN network causing CM accumulation or, alternatively, whether CM accumulation could happen as a result of secondary defects at the organismal level.

Initially, we were surprised to read the reviewer's comments as we have extensively addressed these issues in our manuscript. However, we then realised that the reviewers were not given access to the submitted revised version of our manuscript, therefore, they did not have the chance to appreciate our new data and their comments could only be based on reading our point by point response. Specifically, in our revised manuscript we presented data demonstrating that DARS2^{tamIEC-KO} mice display impaired lipid transport (ED Figure 9b), disorganisation of the Golgi network as well as accumulation of lipids in lipid droplets (Fig. 5d and ED Fig. 10a) already at 5 days after the last tamoxifen injection, when the mice are still healthy and show only moderate weight loss. These new results clearly showed that the lipid processing defect is already apparent at a time point when the mice are still healthy and therefore could not be a consequence of a collapse at the tissue or organismal level. In our view, these results demonstrate that the lipid transport defect and the accumulation of lipids in lipid droplets are direct consequences of mitochondrial dysfunction and not indirect secondary effects caused by severe systemic pathology.

The reviewer also wonders whether the stem cell and proliferation defects observed early on in DARS2^{tamIEC-KO} mice could indirectly cause the lipid accumulation by resulting in impaired enterocyte specification. As we have discussed in our point-by-point response to the reviewers, the existing scientific literature does not support that impaired stemness and proliferation in intestinal crypts could cause lipid accumulation in enterocytes. Specifically, we wrote in the point by point response submitted together with the revised manuscript: *"In addition to cell death, we were also curious whether the strong inhibition of epithelial proliferation caused by DARS2 ablation could contribute to the lipid accumulation. However, extensive data in the literature demonstrate that inhibition of IEC proliferation does not result in lipid accumulation. As an example, we refer to a manuscript published by the group of Hans Clevers (van Es et al., 2012), showing that tamoxifen-inducible ablation of Tcf4 in IECs, using the same Villin-Cre^{ERT2} mouse we used in our study, caused complete inhibition of epithelial cell proliferation and resulted in the death of the mice about 9 days after tamoxifen induction, which is approximately the time that DARS2^{tamIEC-KO} mice need to be sacrificed due to weight loss. As is clearly shown in the manuscript by van Es et al, complete inhibition of epithelial proliferation in mice with IEC-specific Tcf4 knockout did not cause lipid accumulation in enterocytes, arguing that inhibition of proliferation could not be the cause of lipid accumulation. Therefore, our additional results included here*

but also published literature provide compelling evidence that the massive accumulation of lipids within large lipid droplets in enterocytes lacking DARS2, but also SDHA and COX10, is not a secondary effect caused by epithelial or organismal death, or by inhibition of epithelial proliferation.” We have also looked for additional studies reporting on mice with IEC-specific knockout of genes resulting in severe crypt stem cell and proliferation defects that did not cause lipid accumulation. Another example supporting that crypt stem cell defects do not cause lipid accumulation in enterocytes is provided in the study by Zhao et al, who showed that IEC-specific knockout of Zinc finger HIT-type containing 1 (Znhit1), a subunit of the SRCAP chromosome remodelling complex, caused the depletion of Lgr5+ crypt stem cells and inhibited epithelial cell proliferation resulting in the death of the mice 7-8 days after the last tamoxifen injection, but did not cause lipid accumulation in enterocytes (Zhao et al., 2019). We would like to stress that Zhao et al used a nearly identical strategy with our study, by using the same Villin-CreERT2 transgenics and injecting tamoxifen on 4 consecutive days. Therefore, the studies by van Es et al and Zhao et al clearly demonstrated that inducing a severe impairment of crypt stem cells and epithelial proliferation causes severe intestinal pathology resulting in the death of the mice, but **does not cause lipid accumulation in enterocytes**. Furthermore, our results showing that fat-free diet feeding suppressed lipid accumulation but could not prevent the death of the mice, together with the findings of Iqbal et al that mice with inducible enterocyte-specific ablation of *Mttp* displayed impaired chylomicron production and accumulation of fat in large lipid droplets in enterocytes but did not show the dramatic weight loss and death observed in DARS2^{tamIEC-KO} mice (Iqbal *et al.*, 2013) (please see also our response to comment #2 in the rebuttal submitted with the revised manuscript), strongly support that the impairment of dietary lipid transport is not the primary cause of death in DARS2^{tamIEC-KO} mice. Therefore, the most rational interpretation of our results is that severe mitochondrial dysfunction causes defects in crypt stem cells resulting in impaired epithelial proliferation and renewal, as well as an **additional** defect in dietary lipid processing resulting in impaired CM production and secretion and the accumulation of lipids in lipid droplets. Although at present it is impossible to genetically separate these two effects of mitochondrial dysfunction in enterocytes to formally demonstrate that lipid accumulation also occurs independently of the stem cell defects, all available data in the published literature strongly support that these two defects are independent of each other.

The reviewer also wonders why the CM accumulation is not observed at the earliest time point of our analysis, which is 3 days after the last tamoxifen injection. The reviewer argues that this time point is already 8 days after the first tamoxifen injection therefore enterocytes responding initially to tamoxifen should be already replaced by new cells given that the entire epithelium in mice renews every 5-7 days. While this indeed happens in normal mice, we would also like to point out that the inhibition of IEC proliferation in DARS2^{tamIEC-KO} mice is expected to severely hinder enterocyte self-renewal, which is driven by the sloughing off of enterocytes at the tip of the villus and their replacement by cells generated by proliferation at the transit-amplifying zone. Therefore, the DARS2-deficient enterocytes cannot be replaced and should persist for the time frame of our analysis. However, the reviewer is right that at this stage (3 days after the last tamoxifen injection) we observed efficient deletion of DARS2, depletion of respiratory subunits and impaired crypt proliferation but no lipid droplet formation in

DARS2^{tamIEC-KO} mice. Yet, we would like to point out our new results presented in the revised manuscript (Fig. 5d and ED Fig. 9), which the reviewer did not have access to, showing that impaired lipid processing and accumulation are already apparent two days later, namely on day 5 after the last tamoxifen injection. The finding that the lipid processing impairment appears with a slightly delayed kinetic (2-3 days) compared to the proliferation defect does not mean that the former is not directly caused by the mitochondrial dysfunction. It is conceivable that the lipid processing defect requires the re-programming of cellular processes or metabolic changes that take time to manifest. For example, it is possible that critical metabolites need to be depleted or accumulate below or above, respectively, of a threshold before lipid processing is impaired. In fact, it is not unusual that deficiency in important mitochondrial proteins causes severe pathologies that manifest days, weeks or even months after the knockout of the respective genes. For example, conditional knockout of DARS2 in heart and skeletal muscle develop progressive cardiac and muscle pathologies during the first 6 weeks of age, although the Ckmm-cre transgene used to delete the floxed Dars2 allele in muscle cells is fully active during embryonic life (Dogan et al., 2014). In addition, neuron-specific knockout of DARS2 or TFAM using the CaMKIIa-Cre transgene, which deletes floxed alleles in cortical neurons during the first month of life, caused late-onset neuronal pathology manifesting at the age of several months (Aradjanski et al., 2017; Sorensen et al., 2001). Therefore, the slightly delayed appearance of the dietary lipid processing defects cannot be used as an argument that these were not directly caused by mitochondrial dysfunction but are secondary and unspecific consequences of collapse at the organ or organismal level. In fact, we are not aware of any examples in the literature that severe intestinal or systemic pathology indirectly caused accumulation of dietary lipids and CM production defects in enterocytes. In conclusion, with all respect to the reviewer's opinion, our work but also the available literature do not support the argument that the lipid processing defect observed in our mice could be an unspecific secondary consequence of organ or organismal collapse. On the contrary, although the precise underlying molecular mechanism remains to be fully elucidated, we believe that our results reveal a novel role of mitochondria that is critical for efficient processing and transport of dietary lipids by enterocytes.

Referee #2 (Remarks to the Author):

I was truly surprised when I read the revised version of the manuscript. I actually suspected that the journal had provided me with the wrong version of the manuscript. After having checked with the editor at Nature there seems to no mix-up of files at the journal.

Here are some examples of requested changes that have not been implemented in the manuscript despite promises in the response letter:

Major point 4. There is no Fig 5d and ED Fig 10a

Minor point 1, Fig 1E. I see no new panel Fig 3c.

Minor point 2. Where is the stratification of pathways in ED Fig 3g?

Minor point 3 "uptake in plasma" is still there.

Minor point 4. I advised the authors to reword "inhibit" but this word still persists twice in the abstract.

Minor point 6, the authors promised to change the title, which has not been done.

Clearly something is wrong here. I do not think it is meaningful to give this an in-depth review as the response letter clearly is incongruent with the submitted manuscript file. It has certainly wasted my time to deal with this confusion. If the authors really submitted this file as the revised manuscript file, I think the paper should be rejected so that the authors are more careful next time.

We fully understand the response of the reviewer, who realised that the manuscript provided did not contain any of the changes listed in the rebuttal. We were ourselves surprised to find out that the revised manuscript that we submitted was not made available to the reviewers, who instead had access only to the originally submitted first version of the manuscript. We do not know how this happened, but we ask the reviewer to understand that this mistake was not our fault. We hope that when the reviewer reads our revised manuscript, they will appreciate all the effort we have made to address their comments and concerns and improve our manuscript.

Referee #3 (Remarks to the Author):

The authors have addressed many of the reviewers' comments. The revised manuscript is improved.

We thank the reviewer for appreciating that we made a major effort to address their comments and have improved the revised manuscript. Considering that the reviewers did not have access to the revised manuscript file, they were not able to fully appreciate all the additional work we have performed. We hope that when they read the revised version, they will agree with us that we have convincingly addressed most of their critical concerns.

Specific comments:

1. The authors did not actually address the question of whether enterocytes lacking DARS2 (or other genes resulting in loss of mitochondrial function) display evidence of ER stress.

In our response to the respective comment of the reviewer (comment #3) during the first round of review, we explained that DARS2-deficient enterocytes exhibit a strong induction of the mitochondrial branch of the ISR clearly detected by a prominent induction of ATF4 target genes both in the RNAseq and proteomic data. We interpreted this comment of the reviewer as a concern that the lipid accumulation phenotype could be indirectly caused by ER stress. Therefore, we focused our response in discussing examples in the literature showing that mice with IEC-specific knockout of proteins regulating ER stress responses did not show lipid accumulation. We have also directly looked at ER stress in DARS2-deficient enterocytes isolated from mice 7 days after the last tamoxifen injection and indeed found evidence for activation of an ER stress response (see Figure below). However, as we discussed extensively in our point by point response to the original comment of the reviewer, studies performed in numerous mouse models showed that ER stress induction in IECs did not cause lipid accumulation in enterocytes.

Figure for reviewers. Immunoblot analysis of the indicated proteins in enterocytes from *DARS2^{tamIEC-KO}* and littermate control *Dars2^{fl/fl}* mice isolated 7 days after the last tamoxifen injection.

2. It is very quick to delete a gene in liver of floxed mice by injecting AAV-cre, a well-established method.

As discussed in our response to the respective comment of the reviewer (#10) in the rebuttal submitted with the revised manuscript, we respectfully suggested that these studies are outside the scope of the current manuscript, also due to the amount of time and work it will require to fully characterise the effect of DARS2 ablation in hepatocytes. However, if the reviewer considers these studies absolutely essential, we could of course try to perform these experiments as soon as possible, but this will take some time considering the regulatory issues that need to be addressed before generating these mice.

Referee #4 (Remarks to the Author):

The authors show that loss of intestinal DARS2 causes severe mitochondrial dysfunction, resulting in severe weight loss, failure to thrive, and an inability to survive past 4 weeks. Electron microscopy images show swollen mitochondria. Intestinal-specific DARS2-KO mice display decreased numbers of proliferating cells, goblet cells, and absorptive enterocytes, as well as a decrease in the expression of stem-cell markers. Large lipid droplets within DARS2-KO enterocytes were visualized 7 days after tamoxifen injection, which the authors attribute to a disruption in the trans-golgi network and subsequent decrease in chylomicron formation and secretion. Fatty acid and cholesterol absorption is decreased in intestinal specific DARS2-KO mice.

Overall, the data suggest that impairment of mitochondrial health non-specifically compromises many aspects of enterocyte function. Specific mechanistic connections to dietary lipid transport are not established.

We were somewhat puzzled by this response of the reviewer, reiterating some of their original concerns without taking into account all the new results and discussion we provided. However, as explained also above, unfortunately the reviewers did not get access to the revised manuscript that we submitted and they could only read the originally submitted paper, which obviously did not contain any of the new data and the changes described in the rebuttal. We hope many of these issues will be resolved when the reviewer has the chance to read our revised manuscript and fully appreciate all the new results we included.

In response to comment 1 and 3:

1. The data would be more convincing if the authors showed that mitochondrial dysfunction specifically led to lipid droplet accumulation and abrogation of dietary lipid transport, by showing that other energy-dependent pathways were not affected. The results do suggest that mitochondrial dysfunction results in a decrease in IEC and crypt proliferation, as well as cell death, leading to a global decrease in overall enterocyte function, including chylomicron production and secretion. The physiological significance of this observation is unclear, other than mitochondria are important for cell function.

We provided detailed answers to comments 1 and 3 of the reviewer in the point-by-point response submitted with the revised manuscript, where we responded to the specific issues raised by providing specific arguments citing the respective figures of the revised manuscript as well as the relevant literature. We were therefore surprised that the reviewer here reiterates general concerns without taking into account our responses and detailed arguments, which however could be caused by the fact the reviewer did not have access to the revised manuscript. Without wanting to paste here again several pages of text from the rebuttal submitted with the revised manuscript, we would like to briefly outline some aspects that we find particularly important:

First, we responded to the argument about energy depletion as a possible unspecific cause of the lipid accumulation phenotype by providing data showing that the ATP/ADP ratio was not diminished in DARS2-deficient enterocytes, most likely because these cells switched to glycolysis to compensate for the impaired OXPHOS (Figure 5 for reviewers in rebuttal for revision 1, ED. Fig. 4f in revised manuscript). Therefore, these enterocytes do not display energy deficiency.

Second, as clearly discussed in the rebuttal in response to comment #3 of the reviewer, we extensively assessed the possible role of enterocyte death and found no evidence of increased cell death in the intestines of our mice (Data included in ED Fig. 1e and ED Fig. 5b, d of the revised manuscript). Moreover, we could not rescue the phenotype by inhibiting apoptosis and necroptosis (Figure 1 for reviewers in the rebuttal). These results demonstrate that the lipid accumulation defect could not be caused by increased death of enterocytes, which is further supported by the fact that no lipid accumulation was observed in a large number of mouse models with genetic knockouts causing increased enterocyte death (please also read our detailed response to this in the rebuttal submitted with the revised manuscript).

Third, we specifically discussed that the impaired proliferation and crypt stem cell defects could not be the cause of the lipid phenotype by providing specific examples

in the literature. For this please also see our detailed response to reviewer 1 above, where we provide justified arguments why the chylomicron production defect and lipid accumulation observed in our mice cannot be an indirect and unspecific consequence of the proliferation and stem cell defects. As discussed in this response, the most rational interpretation of our data is that these two defects, namely the proliferation impairment and the lipid accumulation, occur independently of each other and are both directly caused by mitochondrial dysfunction.

In response to comment 6:

2. It is unclear how mitochondrial dysfunction is related to a disrupted golgi network and chylomicron formation. The authors state that golgi disorganization is not a direct consequence of severe mitochondrial dysfunction, which makes golgi disorganization a poor explanation for why chylomicron formation appears to be inhibited in the setting of mitochondrial dysfunction. Furthermore, the authors need to show that the disruption in the golgi network is restricted to lipid absorption alone, by excluding that other secretory effects modulated by the golgi network are affected. It still remains possible that mitochondrial dysfunction leads to a disruption in the TGN. The authors do observe that TGN disorganization only occurs in the proximal small intestine and not the distal small intestine, which suggests that mitochondrial dysfunction may not be related to problems within the TGN network at all.

As discussed above, the reviewer did not have access to all the new data and the revised manuscript, which likely explains why they repeat here their original concerns. As we extensively discussed in our detailed response to this comment of the reviewer, the disorganisation of the Golgi is coupled to the impairment of lipid processing. Without wanting to paste again here our extensive response that can be read in the rebuttal submitted with the revised manuscript, we would like to briefly outline the key points:

First, extensive Golgi disorganisation is observed in proximal enterocytes from $DARS2^{\text{tamIEC-KO}}$ mice 7-8 days after the last tamoxifen injection, concomitant with the massive lipid accumulation in large lipid droplets observed at this stage.

Second, our new results presented in the revised manuscript clearly demonstrated that the disorganisation of the Golgi depends on the processing of dietary lipids. This is supported by two specific pieces of evidence: a) Golgi dispersal was only observed in proximal and not in distal enterocytes from the same $DARS2^{\text{tamIEC-KO}}$ mice (ED Fig. 10b), therefore this defect only occurred in enterocytes endowed with the task of processing and transporting dietary lipids, and b) Golgi dispersal was not observed in proximal enterocytes of $DARS2^{\text{tamIEC-KO}}$ mice fed a fat-free diet (ED Fig. 10c), showing that this is induced by the presence of lipids.

Third, our new analysis of mice 5 days after the last tamoxifen injection clearly demonstrated that a mild dispersal of the Golgi is already observed in proximal enterocytes from $DARS2^{\text{tamIEC-KO}}$ mice at this stage, when the formation of lipid droplets is also observed in a small number of enterocytes (Fig. 5d). Importantly, the Golgi dispersal is observed in most proximal enterocytes of $DARS2^{\text{tamIEC-KO}}$ mice at this stage, also in enterocytes that do not contain lipid droplets (Fig. 5d), which

demonstrated that the Golgi dispersal does not happen as secondary consequence of lipid droplet formation but rather precedes the formation of the lipid droplets.

Collectively, the most rational interpretation of these results is that mitochondrial dysfunction induces an impairment in dietary lipid processing in proximal enterocytes, which results in dispersal of the Golgi network most likely due to impaired trafficking of CM transport vesicles from the ER (maintenance of the Golgi depends on membrane trafficking from the ER), ultimately causing the accumulation of dietary fat within large cytoplasmic lipid droplets in enterocytes.

In response to comment 3:

2. The authors say that there was “no evidence of increased IEC death in the intestine of either the constitutive DARS2-IEC-KO or tamoxifen-inducible DARS2-tam-IEC-KO mice”.

-The figures only show Ki67 staining, a marker of proliferation, which is absent from crypt cells of DAR2-IEC-KO intestines. There are no markers of cell death.

Therefore, it is unclear whether mitochondrial dysfunction leads to enterocyte death, which compromises lipid absorption.

Obviously, the reviewer could not appreciate the new data we presented on the analysis of cell death as they did not have access to the revised manuscript and figures. As mentioned also above, we extensively assessed the possible role of enterocyte death and found no evidence of increased cell death in the intestines of our mice (Data included in ED Fig. 1e and ED Fig. 5b, d of the revised manuscript). Moreover, we could not rescue the phenotype by inhibiting apoptosis and necroptosis (Figure 1 for reviewers in the rebuttal). These results demonstrate that the lipid accumulation defect could not be caused by increased death of enterocytes, which is further supported by the fact that no lipid accumulation was observed in a large number of mouse models with genetic knockouts causing increased enterocyte death. We hope that the reviewer will agree with us when they have the chance to read the revised manuscript and the additional data provided in the respective figures.

In response to comment 3:

3. All experiments were performed 7-8 days after tamoxifen injection, when IEC and crypt cell proliferation were impaired with highly abnormal appearing enterocytes and a decrease in villi to crypt ratio, signifying enterocyte cell death.

-Crypt cells contribute little to lipid absorption and produce a minimal amount of chylomicrons if any. Therefore, the decrease in the proliferation of crypt cells is unlikely to be due to lipid accumulation, the formation of lipid droplets, and an abrogation of dietary lipid transport, which the authors attribute as the cause for the severe phenotype of DARS2-KO mice. If this were secondary to an inability to transport lipids, crypt cells should not be affected. This makes the phenotype of impaired proliferation and cell death, more likely to result from mitochondrial dysfunction.

-Furthermore, no signs of lipid droplet accumulation were seen 3 days after tamoxifen injection though IEC proliferation was already impaired. This was also shown on DARS2-KO mice on a fat free diet that displayed a minimal amount of lipid droplet accumulation, even when IEC proliferation was impaired. This indicates that lipid droplet accumulation is unlikely to be the cause of the phenotype.

Also here, the reviewer could not appreciate all our new results in the revised manuscript supporting that the lipid accumulation is not secondary to the inhibition of proliferation, death of enterocytes or a general collapse of at the organ or organismal level. We have responded in detail to the original comment #3 of the reviewer in the rebuttal, but would like to briefly mention here again the key points:

First, as discussed above, we demonstrated that cell death cannot be the cause of impaired dietary fat processing and lipid droplet formation in enterocytes from DARS2^{tamIEC-KO} mice (Data included in ED Fig. 1e and ED Fig. 5b, d of the revised manuscript).

Second, as discussed extensively also in response to reviewer 1 above (please read our response above as it does not make sense to paste the same text again here), the defects in proliferation and crypt stem cells cannot be the cause of impaired lipid processing and lipid droplet formation in DARS2-deficient enterocytes.

Third, as discussed explicitly both in the original rebuttal and also in response to reviewer 1 above, but also clearly stated in the revised manuscript (lines 254-262), we do not claim that the lipid processing defect caused the death of the mice. Instead, our interpretation of our results is that mitochondrial dysfunction caused two independent pathologies in the intestinal epithelium. On the one hand, it induced a proliferation and stem cell defect, which based on evidence from other mouse models as for example described in the cited studies by van Es et al and Zhao et al (van Es *et al.*, 2012; Zhao *et al.*, 2019) is most likely the reason causing the death of the mice but cannot be the cause of lipid accumulation. In parallel, mitochondrial dysfunction causes an impairment in processing and transport of dietary fat resulting in its storage within large cytoplasmic lipid droplets in enterocytes. We focused our manuscript primarily on the latter as this is the most novel aspect of our work.

-Perhaps, the studies would have been more convincing if performed 3 days after tamoxifen injection, when efficient ablation of DARS2 was achieved and while the enterocytes were healthier.

Also here, unfortunately the reviewer did not have the chance to read the revised manuscript and appreciate the new data we included. Specifically, we performed extensive analysis of mice 5 days after tamoxifen injection, which are still healthy and only display a moderate loss of body weight. We found that already at this stage the mice displayed moderately impaired transport of dietary fat to the circulation (ED Fig. 9), early signs of dispersal of the Golgi network and formation of lipid droplets in proximal enterocytes (Fig. 5d and ED Fig. 10). Therefore, the impairment in dietary lipid processing and transport is clearly observed at a stage when the mice are still healthy. We hope that these new results convincingly address the concerns of the reviewer.

4. There is no apparent clinical relevance for decreased triglyceride absorption in the setting of mitochondrial dysfunction.

As we have also discussed in our response to a similar comment from reviewer 2 in the original rebuttal, it is unclear if similar defects in dietary lipid processing are present in patients with mitochondrial diseases. As mitochondria have not been implicated in this process previously, this aspect has not been studied in human patients. However, our work warrants studies assessing if dietary lipid processing defects contribute to the pathologies induced by mutations affecting mitochondrial function in human patients. We have adapted accordingly the discussion in the revised manuscript (lines 388-393).

References

- Aradjanski, M., Dogan, S.A., Lotter, S., Wang, S., Hermans, S., Wibom, R., Rugarli, E., and Trifunovic, A. (2017). DARS2 protects against neuroinflammation and apoptotic neuronal loss, but is dispensable for myelin producing cells. *Hum Mol Genet* 26, 4181-4189. 10.1093/hmg/ddx307.
- Auclair, N., Sane, A.T., Ahmarani, L., Patey, N., Beaulieu, J.F., Peretti, N., Spahis, S., and Levy, E. (2021). Sar1b mutant mice recapitulate gastrointestinal abnormalities associated with chylomicron retention disease. *J Lipid Res* 62, 100085. 10.1016/j.jlr.2021.100085.
- Dogan, S.A., Pujol, C., Maiti, P., Kukat, A., Wang, S., Hermans, S., Senft, K., Wibom, R., Rugarli, E.I., and Trifunovic, A. (2014). Tissue-specific loss of DARS2 activates stress responses independently of respiratory chain deficiency in the heart. *Cell Metab* 19, 458-469. 10.1016/j.cmet.2014.02.004.
- Gordon, S.M., Neufeld, E.B., Yang, Z., Pryor, M., Freeman, L.A., Fan, X., Kullo, I.J., Biesecker, L.G., and Remaley, A.T. (2019). DENND5B Regulates Intestinal Triglyceride Absorption and Body Mass. *Sci Rep* 9, 3597. 10.1038/s41598-019-40296-0.
- Iqbal, J., Parks, J.S., and Hussain, M.M. (2013). Lipid absorption defects in intestine-specific microsomal triglyceride transfer protein and ATP-binding cassette transporter A1-deficient mice. *J Biol Chem* 288, 30432-30444. 10.1074/jbc.M113.501247.
- Sorensen, L., Ekstrand, M., Silva, J.P., Lindqvist, E., Xu, B., Rustin, P., Olson, L., and Larsson, N.G. (2001). Late-onset corticohippocampal neurodepletion attributable to catastrophic failure of oxidative phosphorylation in MILON mice. *J Neurosci* 21, 8082-8090.
- van Es, J.H., Haegebarth, A., Kujala, P., Itzkovitz, S., Koo, B.K., Boj, S.F., Korving, J., van den Born, M., van Oudenaarden, A., Robine, S., and Clevers, H. (2012). A critical role for the Wnt effector Tcf4 in adult intestinal homeostatic self-renewal. *Mol Cell Biol* 32, 1918-1927. 10.1128/MCB.06288-11.
- Young, S.G., Cham, C.M., Pitas, R.E., Burri, B.J., Connolly, A., Flynn, L., Pappu, A.S., Wong, J.S., Hamilton, R.L., and Farese, R.V., Jr. (1995). A genetic model for absent chylomicron formation: mice producing apolipoprotein B in the liver, but not in the intestine. *J Clin Invest* 96, 2932-2946. 10.1172/JCI118365.
- Zhao, B., Chen, Y., Jiang, N., Yang, L., Sun, S., Zhang, Y., Wen, Z., Ray, L., Liu, H., Hou, G., and Lin, X. (2019). Znhit1 controls intestinal stem cell maintenance by regulating H2A.Z incorporation. *Nature communications* 10, 1071. 10.1038/s41467-019-09060-w.

Reviewer Reports on the Second Revision:

Referees' comments:

Referee #1 (Remarks to the Author):

The revised manuscript from the authors demonstrates that: 1) Mitochondrial impairment causes LD accumulation in the proximal enterocytes, as evidenced by cell biological and biochemical/molecular methods; (2) Mitochondrial impairment also causes multiple failures at the cellular and organismal level. Notably, LD accumulation is not the cause of organismal death (which is clarified from additional evidence) but whether the converse is true is the point of concern.

I understand that LD accumulation is generally not observed in other genetic mutants that impact intestine proliferation, in favor of the argument that mitochondrial defects cause LD specific. These are circumstantial evidence, though— first, it is hard to argue for something that has not been found in previous literature since they were not specifically looking for LD defects. Moreover, the causes of crypt anomalies are different in every case. For example, Tcf4 knockout ablates the intestinal crypt formation at the stem cell level, independent of enterocyte specification or maturation, so unless you find the same cell type death for defects, it's not a fair comparison. A direct way to show LD accumulation is "independent" from organismal failure is to (1) Ablate DARS/SDHA/Cox4 in an enterocytes-specific manner (ex. Alpi-Cre) to decouple proliferation, and other organismal defects; (2) Show similar effects in cell-culture or organoids model. These are not "impossible" but I understand that these experiments will take time to develop and analyze. As an alternative, the authors examine Dars acute deletion models as suggested (3d-5d after the last tamoxifen dose) to show that LD accumulation is not apparent at this stage, and there is a delay in the kinetics to manifest the phenotype. Unfortunately, this acute deletion result is open to interpretation (I could see the argument go in both ways).

My previous points encouraged the authors to delve into mechanistic insights on LD accumulation. Which cell types are responsible? What aspects of mitochondria are needed? I think all reviewers are asking for the same question in their background, providing potential ways to probe this question. My thought was that (specific) mitochondrial function is required for mature enterocyte specification (which is not tested in this manuscript, and this is fine). The manuscript does an excellent job of demonstrating accumulating LDs – but what I don't see from this manuscript is how the authors think about this lipid processing problem at the mechanistic level, what aspects of mitochondria as a whole organelle, and in which cell types are connected to LD biology. Does this stem from defects in mitochondrial energetics? Specific metabolites?

Referee #2 (Remarks to the Author):

I feel that the concerns expressed in my review have been fully addressed and I have no further comments. I congratulate the authors to a well executed and interesting study.

Referee #3 (Remarks to the Author):

None

Referee #4 (Remarks to the Author):

Although some of the comments from the original review have been addressed, and the manuscript is more complete in the current form, I continue to have major reservations about the physiological conclusions that can be drawn from the data.

The phenotype is profound, as the authors show dramatic lipid accumulation characterized by big cytosolic lipid droplets. The phenotype has a dramatic effect on multiple physiological processes, including cell proliferation and nutrient absorption.

In the rebuttal letters the authors stress the idea that "an indirect and unspecific consequence of general malfunction at the cellular organ or organismal level is not supported by any evidence". However, they do not provide any convincing mechanistic explanation of why they observe this phenotype. The mechanism proposed is that mitochondrial dysfunction determines Golgi disorganization and that, in turns, this prevents maturation of pre-chylomicrons to mature chylomicrons. These conclusions were made based on electron micrographs, which are very hard to interpret because even single enterocytes in the same villus can look very different, and these differences might become more dramatic when looking into different villi or different segment of intestine.

This work would be potentially interesting if the authors elucidated why and how mitochondrial disfunction affects Golgi disorganization and chylomicrons release, but the current data do not establish such clear connections

Author Rebuttals to Second Revision:

Point by point response to the reviewers' comments

Referee #1 (Remarks to the Author):

The revised manuscript from the authors demonstrates that: 1) Mitochondrial impairment causes LD accumulation in the proximal enterocytes, as evidenced by cell biological and biochemical/molecular methods; (2) Mitochondrial impairment also causes multiple failures at the cellular and organismal level. Notably, LD accumulation is not the cause of organismal death (which is clarified from additional evidence), but whether the converse is true is the point of concern.

The reviewer argues that LD accumulation in enterocytes with mitochondrial dysfunction is not a direct effect but could instead be indirectly caused by organismal death. This is surprising to us as in our last rebuttal and revision we provided clear evidence and convincing arguments why the lipid accumulation could not be an unspecific and indirect outcome of organismal death. Please allow us to briefly summarise here the main arguments that in our view demonstrate that LD accumulation is not an indirect effect caused by enterocyte or organismal death:

First, we showed that mitochondrial dysfunction does not cause enterocyte death (ED Fig. 1e and ED Fig. 5b, d), and that blocking cell death (apoptosis and necroptosis in $DARS2^{IEC-KO}$ $CASP8^{IEC-KO}$ $Mik1^{-/-}$ mice) did not rescue the lipid accumulation or the death of the mice. Therefore, LD formation in enterocytes is not caused by cell death.

Second, in the revised manuscript we provided new experimental evidence showing that LD formation is already observed at 5 days after the last tamoxifen injection (Fig. 5d and ED Fig. 6h). We would like to stress that at this time point the mice are still healthy and only show a mild loss of body weight. In fact, organismal death (meaning when the mice reach 20% loss of body weight and need to be sacrificed according to the ethical protocol) will occur 3-4 days **after** this time point. Therefore, LD formation occurs 3-4 days before the mice develop severe intestinal pathology and need to be sacrificed. In our view, this is clear and convincing evidence that LD formation is not caused by organismal death but is a direct consequence of mitochondrial dysfunction.

I understand that LD accumulation is generally not observed in other genetic mutants that impact intestine proliferation, in favor of the argument that mitochondrial defects cause LD specific. These are circumstantial evidence, though— first, it is hard to argue for something that has not been found in previous literature since they were not specifically looking for LD defects.

With due respect, we strongly disagree with the reviewer that LD accumulation in enterocytes could have been missed in previous studies. All studies dealing with intestinal function in mouse models with systemic or epithelial-specific knockouts, no matter what the focus is, include a basic histological analysis of intestinal tissues in the figures. The lipid accumulation phenotype, as is clearly shown in the figures of our paper (for example in Figures 1f, 2d&i, 3d&e, etc.), is impossible to miss even in basic H&E-stained sections because of the unusual and characteristic presence of large lipid-laden vacuoles in enterocytes. Moreover, lipid accumulation in enterocytes would immediately be noticed macroscopically by the white coloured and thickened proximal SI (ED Fig. 5a). In fact, we ourselves did not intend to study lipid accumulation originally, however, we immediately observed the formation of large LDs in enterocytes of $DARS2^{IEC-KO}$ mice, which was the initial finding that prompted us to investigate in depth the underlying mechanism. Therefore, the lipid accumulation in LD in enterocytes is very obvious both microscopically and macroscopically and cannot be missed or ignored by any investigator.

Moreover, the causes of crypt anomalies are different in every case. For example, Tcf4 knockout ablates the intestinal crypt formation at the stem cell level, independent of enterocyte specification or maturation, so unless you find the same cell type death for defects, it's not a fair comparison.

As discussed above, we provided clear evidence that enterocyte death does not play a role in our mouse models. Therefore, we do not understand the argument of the reviewer that "... unless you find the same cell type death for defects, it's not a fair comparison". We used the Tcf4 knockouts to make the point that suppression of epithelial proliferation, which is observed in our mice with DARS2 but also COX10 and SDHA deletion, could not be the cause of lipid accumulation through an indirect and unspecific effect. To support this argument, we pointed out that intestinal epithelial cell (IEC)-specific knockouts in other genes, specifically *Tcf4* (van Es *et al*, 2012) and *Znht1* (Zhao *et al*, 2019), which caused a complete inhibition of epithelial cell proliferation resulting in the death of the mice, **did not cause lipid accumulation** in enterocytes. We also argued that, despite the fact that there are tens if not hundreds of studies in the literature reporting on mice with IEC-specific knockouts of different genes that caused severe intestinal pathologies, to the best of our knowledge **there is not even one example** where "unspecific" stress in enterocytes induced the massive lipid accumulation. In fact, the only mice showing lipid accumulation in proximal enterocytes under normal chow diet are those with mutations in genes affecting chylomicron production, trafficking and release such as ApoB (Young *et al*, 1995), Mttp (Iqbal *et al*, 2013), Sar1b (Auclair *et al*, 2021) and DENND5B (Gordon *et al*, 2019). These results demonstrate that the impairment of the transport of dietary lipids resulting in their accumulation in cytosolic LDs in enterocytes is a highly specific response caused only by mutation of proteins directly involved in chylomicron production and/or release. Simply put, there is no example in the literature that 'unspecific' effects related to intestinal pathology/malfunction can cause dietary lipid accumulation in proximal enterocytes. Therefore, our data provide strong evidence that accumulation of dietary lipids in large LD in enterocytes is directly caused by mitochondrial dysfunction.

A direct way to show LD accumulation is "independent" from organismal failure is to (1) Ablate DARS/SDHA/Cox4 in an enterocytes-specific manner (ex. Alpi-Cre) to decouple proliferation, and other organismal defects; (2) Show similar effects in cell-culture or organoids model. These are not "impossible" but I understand that these experiments will take time to develop and analyze.

The reviewer again brings up the question of suppressed proliferation on the LD accumulation. We have already disputed this by showing that: (1) distal small intestine does not accumulate LDs despite having comparable proliferation defect as proximal small intestine (ED Fig. 5c), (2) fat-free diet sturdily decreased accumulation of LDs in proximal small intestine, despite no effect on the proliferation deficiency (Fig. 4a). Furthermore, in our mice, tamoxifen-inducible ablation of DARS2 occurs in all enterocytes. Since crypt proliferation is suppressed already 3 days after tamoxifen, there are no new enterocytes produced that would need to differentiate or mature (ED Fig. 6d). Therefore, the lipids accumulate in enterocytes that are already differentiated and matured, otherwise they would not have the capacity to absorb fatty acids from the lumen. For this reason, we do not see how using another Cre line such as Alpi-Cre could help in this case.

Nevertheless, as suggested by the reviewer, we have put a major effort in developing a cellular system recapitulating the phenotype induced by mitochondrial dysfunction in enterocytes in vivo. We have now succeeded to establish such a cellular system, using IEC-6 cells, an intestinal, non-transformed, epithelial cell line that is derived from rat small intestine and displays typical characteristics of small intestinal enterocytes (Quaroni *et al*, 1979). We opted for using mitochondrial inhibitors instead of directly depleting DARS2, SDHA or COX10, as our preliminary results showed that even minor amounts of these enzymes (typically present after *siRNA* depletion) could sustain normal respiration in cell culture, while full knockouts of these

genes are not viable. Our broad experience with inhibitors of mitochondrial function allowed us to choose the ones most closely mimicking the effect of the loss of DARS2 or SDHA in cells. Specifically, we used actinonin, which causes a defect in mitochondrial protein synthesis and mimics DARS2 deficiency, and Atpenin A5 (AA5), an inhibitor of succinate dehydrogenase (SDH). As a control, we used treatment with brefeldin A (BFA), which inhibits anterograde transport from ER to Golgi and in particular COPII vesicular trafficking, leading to a strong defect in Golgi integrity. We could show that treatment with actinonin or AA5 caused disorganisation of the Golgi apparatus, evidenced by the extensive lack of a compact, juxtannuclear TGN38 staining (Figure 1a for the Reviewers, Extended Data Figure 11a of the revised manuscript). Importantly, this effect was clearly exaggerated in the presence of oleic acid (OA) in the medium, which also caused the accumulation of large LDs in cells treated with actinonin or AA5. In contrast, the presence of oleic acid did not affect the structure of the Golgi in control cells that were not treated with mitochondrial inhibitors (Figure 1b for the Reviewers, Extended Data Figure 11b of the revised manuscript). Therefore, inhibition of mitochondrial function induced Golgi disorganisation that was exacerbated in the presence of increased lipid load in a cellular system, recapitulating our *in vivo* findings in mice with IEC-specific knockout of DARS2. These results provide additional support that the lipid processing defect observed in our mice is a direct effect caused by mitochondrial dysfunction rather than an unspecific consequence of organ or organismal failure.

Figure 1 for Reviewers. Interference with mitochondrial function causes Golgi disorganisation and lipid accumulation in IEC-6 cells.

a, Representative fluorescence microscopy images depicting IEC-6 cells treated for 48 hours with actinonin (100uM), Atpenin A5 (AA5, 1uM) or 1% dimethyl sulfoxide; DMSO (control). Short treatment (6 hours) with Brefeldin A (BFA, 5ug/ml) was used as a positive control for Golgi dispersal. **b**, Representative fluorescence microscopy images of IEC-6 cells grown under the same conditions as described in **a** were incubated with oleic acid (OA, 600uM) for the last 24 hours prior imaging. In this case, BFA was applied in the last 6 hours of OA treatment to avoid cytotoxicity. Anti-TGN38 (red) antibody was used to visualize Golgi, Anti-COX1 (green) to stain mitochondrial networks (**a**), BODIPY (green) to stain lipid droplets (**b**), and DAPI (blue) for nuclei. TGN38 staining is additionally depicted in white. Scale bars = 50µm. Quantification of the observed Golgi morphology of the IEC-6 cells based on five distinct categories, as illustrated at the right (n=3 independent biological experiments).

Figure 2 for Reviewers. Mitochondrial dysfunction caused by *dars-2* depletion leads to disruption in Golgi apparatus and defect in vitellogenin transport

a, Representative confocal images and graphs depicting quantification of GFP signal in *C. elegans* expressing α -mannosidase II-GFP or SPCS-1 fused to GFP under the control of the gut-specific *vha-6* promoter grown either on a control empty vector (EV) or RNAi against *dars-2* at the first (D1) and fourth day of adulthood (D4). Scale bars = 1 μ m. **b**, Immunofluorescence micrographs of *C. elegans* carrying *vit-2::GFP* reporter on a control empty vector (EV) and RNAi against *sar-1*, *sec-13*, *fum-1* and *dars-2* at D1. Insets show magnification of a selected area of the worm.

To provide additional support that mitochondrial dysfunction directly impacts on the organisation of the Golgi and intestinal lipid trafficking and illustrate evolutionary conservation of this pathway, we turned to our studies in *C. elegans* part of which were included in the first version of the manuscript. We first took advantage of a transgenic *C. elegans* strain expressing the Golgi-specific α -mannosidase II fused to GFP under the control of the gut-specific *vha-6* promoter, to assess whether DARS2 deficiency also affects the Golgi in the intestine of worms. Indeed, *Dars-2* RNAi considerably reduced the number of α -mannosidase II-stained puncta in *C. elegans* gut at adult day 1 (D1) and 4 (D4), showing that DARS2-deficiency caused dispersion of the Golgi also in worms, while not affecting the ER morphology, as evaluated using a strain expressing the ER-specific SPCS-1 fused to GFP under the control of the gut-specific *vha-6* promoter (Xu *et al*, 2012) (Figure 2a for Reviewers and ED Fig. 11c). Next, we set out to check if this deficiency would affect the trafficking of lipids in *C. elegans*. The best known and studied lipid trafficking model in *C. elegans* is transport of vitellogenin, VIT-2 in particular (Balklava *et al*, 2007). Vitellogenins are large lipo-glyco-phosphoproteins that recruit lipids and other nutrients before they are secreted into circulation and are taken up by oocytes via receptor-mediated endocytosis. They are believed to be the ancestor of human ApoB, and the COP II vesicular trafficking is similarly required for their secretion and transport (Baker, 1988). In *C. elegans*, VIT-2 is predominantly found in oocytes, but in the absence of SAR-1 (SAR1B homologue) or other proteins essential for COP II vesicle formation (SEC-13), it accumulates in the intestine (Figure 2b for Reviewers and ED Fig 11d). *Dars-2* depletion

equally strongly prevents VIT-2 transport into oocytes, as visualised by the VIT-2:: GFP reporter. In contrast, strong depletion of fumarate hydratase (FUM-1), a mitochondrial TCA cycle enzyme essential for development and numerous metabolic processes that also acts as a tumour suppressor, did not affect vitellogenin transport (Schmidt *et al.*, 2020). Downregulation of other mitochondrial enzymes (not shown), e.g. *suc1-1* (Succinyl-CoA Ligase, alpha subunit), *icl-1* (isocitrate lyase) similarly had no effect on vitellogenin transport. Further support for our results and the role of mitochondria in the transport of lipoprotein complexes in *C. elegans*, comes from a screen for factors essential for vitellogenin transport using the same reporter (VIT-2:: GFP) (Balklava *et al.*, 2007). Remarkably, this screen identified 40 mitochondrial proteins, out of which 28 directly regulate mitochondrial protein synthesis (15 mitochondrial subunits, five mitochondrial transcription or translation factors and eight mitochondrial tRNA amino-acyltransferases, enzymes that belong to the same group as DARS2) (Balklava *et al.*, 2007). Collectively, these results support that mitochondria have an important, evolutionarily conserved function in regulating Golgi organisation and lipid transport.

As an alternative, the authors examine Dars acute deletion models as suggested (3d-5d after the last tamoxifen dose) to show that LD accumulation is not apparent at this stage, and there is a delay in the kinetics to manifest the phenotype. Unfortunately, this acute deletion result is open to interpretation (I could see the argument go in both ways).

We were surprised to read the reviewer's comment that LD accumulation is not apparent at 5 days after the last tamoxifen injection. In the revised version of our manuscript, we had included experimental evidence showing that 5 days after tamoxifen injection proximal enterocytes showed clear signs of Golgi fragmentation, as well as the appearance of lipid droplets, in some but not all enterocytes (Figure 5d and ED Fig 10a). Consistent with these results, our latest metabolic tracing experiments presented in ED Figure 9 demonstrated that already 5 days after tamoxifen induction the mice showed impaired transport of dietary lipids to the circulation. Furthermore, the immunoblot analysis of plasma lipoproteins presented in Figure 5b revealed a clear reduction of the protein levels of ApoB48 in DARS2^{tamIEC-KO} mice at both 5 and 7 days after tamoxifen, whereas ApoB48 levels in enterocytes were increased in these animals (Figure 5a). ApoB48 is exclusively a component of enterocyte-produced chylomicrons, therefore these experiments provided additional support that already 5 days after tamoxifen DARS2-deficient enterocytes showed impaired chylomicron production and release. Collectively, the results obtained from three different experimental approaches clearly showed that Golgi dispersal, impaired chylomicron production and release and lipid accumulation in lipid droplets occurs already at 5 days after tamoxifen, long before the mice developed severe stress which takes another 2-3 days to manifest. We consider this as strong and unequivocal evidence that lipid accumulation occurs as a direct consequence of mitochondrial dysfunction and could not be caused by unspecific "organismal defects".

My previous points encouraged the authors to delve into mechanistic insights on LD accumulation. Which cell types are responsible? What aspects of mitochondria are needed? I think all reviewers are asking for the same question in their background, providing potential ways to probe this question. My thought was that (specific) mitochondrial function is required for mature enterocyte specification (which is not tested in this manuscript, and this is fine). The manuscript does an excellent job of demonstrating accumulating LDs – but what I don't see from this manuscript is how the authors think about this lipid processing problem at the mechanistic level, what aspects of mitochondria as a whole organelle, and in which cell types are connected to LD biology. Does this stem from defects in mitochondrial energetics? Specific metabolites?

Our work reveals a new and important function of mitochondria in controlling dietary lipid processing in enterocytes. Although we initially considered that an energy crisis might underlie the lipid processing defect, our data showing that even 8 days after tamoxifen injection, despite

the severe OXPHOS deficiency, $DARS2^{tamIEC-KO}$ mice have normal ATP/ADP ratio and increased glycolytic intermediates (ED Fig. 4f), clearly demonstrated that the $DARS2$ -deficient enterocytes do not suffer from energy depletion. This is consistent with data from several studies showing that an energy crisis appears very late even in primary OXPHOS deficiencies, such as those caused by ablation of $DARS2/SDHA/COX10$, as metabolic plasticity of most cells allows them to very efficiently compensate for the ATP production by switching to glycolysis. Over the last years it became clear that mitochondria are much more than just “cellular power houses” and that shutting them down would not simply cause the bioenergetic collapse of the cells/tissues/organisms. Such a view of mitochondria is outdated, as these organelles are now widely recognized to perform many important functions that are not directly linked to energy production. To obtain insights into the specific metabolic consequences of $DARS2$ deficiency, we have performed extensive metabolomic analysis that identified distinct alterations in metabolites in $DARS2$ -deficient enterocytes that could be involved in the phenotype. While these analyses provide interesting leads for future investigation, an exhaustive functional interrogation of all possible pathways will likely take several years and therefore, unfortunately, cannot be included in this manuscript. If we may make a general comment here, this is almost always the case in cutting-edge research, which often elucidates novel biological functions but at the same time opens up many important questions that need to be answered in the future. As we clearly discuss in our manuscript, our data reveal a previously unappreciated and important role of mitochondria in dietary lipid processing and transport in enterocytes. While the precise underlying molecular mechanism remains to be identified, this is different from saying that the phenotype is unspecifically induced by vaguely defined organismal defects and not a specific consequence of mitochondrial dysfunction, which appears to be the major concern of the reviewer. As discussed extensively in our response to the respective specific comment of the reviewer above, our results clearly support that mitochondrial dysfunction causes impaired chylomicron production associated with Golgi disorganization, which explains the impaired transport and accumulation of dietary lipids in large LDs in enterocytes. In our view, this provides a convincing mechanistic explanation of the *in vivo* biological data, which is also supported by a large amount of literature.

Referee #2 (Remarks to the Author):

I feel that the concerns expressed in my review have been fully addressed and I have no further comments. I congratulate the authors to a well executed and interesting study.

We are glad that our revised manuscript fully addressed the concerns of the reviewer. We wish to thank the reviewer for their constructive critique that helped us improve our manuscript and for their positive assessment of our work.

Referee #4 (Remarks to the Author):

Although some of the comments from the original review have been addressed, and the manuscript is more complete in the current form, I continue to have major reservations about the physiological conclusions that can be drawn from the data.

The phenotype is profound, as the authors show dramatic lipid accumulation characterized by big cytosolic lipid droplets. The phenotype has a dramatic effect on multiple physiological processes, including cell proliferation and nutrient absorption. In the rebuttal letters the authors stress the idea that “an indirect and unspecific consequence of general malfunction at the cellular organ or organismal level is not supported by any evidence”. However, they do not provide any convincing mechanistic explanation of why they observe this phenotype. The mechanism proposed is that mitochondrial dysfunction determines Golgi disorganization and that, in turns, this prevents maturation of pre-chylomicrons to mature chylomicrons. These conclusions were made based on electron micrographs, which are very hard to interpret because even single enterocytes in the same

villus can look very different, and these differences might become more dramatic when looking into different villi or different segment of intestine.

With due respect, we do not agree with the reviewer that our conclusions that mitochondrial dysfunction caused Golgi disorganisation and impaired CM production “...were made based on electron micrographs.” Please allow us to summarise the evidence we provide to support our conclusions:

First, in addition to the electron micrographs presented in Figure 5c, we have used immunofluorescence microscopy to assess the integrity of the Golgi in the intestines of DARS2^{tamIEC-KO} mice at different time point after tamoxifen injection. Specifically, we showed in ED Figure 10 that enterocytes from DARS2^{tamIEC-KO} mice fed a normal chow diet presented widespread and profound Golgi disorganization in the proximal SI concomitant with the accumulation of lipids in large LDs. Furthermore, in Figure 5d, we showed that Golgi dispersal is already apparent 5 days after tamoxifen treatment and provided quantification showing this is a consistent and statistically significant finding. The reviewer somehow overlooked this dataset, which we think unequivocally demonstrates that mitochondrial dysfunction has a profound effect on the Golgi organization in enterocytes. In the same figure (Fig. 5d) but also in ED Fig. 6h, we showed that small lipid droplets were already present in several enterocytes 5 days after tamoxifen induction, a time point when the mice are not sick yet. These findings provide experimental evidence that Golgi dispersal precedes the formation of the lipid droplets, both occurring at a time when the mice are not yet under severe stress arguing these are direct effects and not unspecific consequences of ‘organismal defects’.

Second, our conclusion that mitochondrial dysfunction in enterocytes caused impaired maturation and secretion of chylomicrons is not only based on EM analysis. Specifically, in Figure 5b, we provided immunoblot analysis of plasma lipoproteins showing strongly reduced levels of ApoB48, a specific structural protein of enterocyte-secreted chylomicrons, in the plasma of DARS2^{tamIEC-KO} mice 5 and 7 days after tamoxifen induction. These results clearly showed that IEC-specific DARS2 deficiency suppressed chylomicron production and release to the circulation. Moreover, in Fig. 4 and ED Fig. 9, we provided experimental evidence showing that enterocyte-specific DARS2 ablation suppressed the transport of dietary lipids from the intestinal lumen to the circulation, consistent with an impairment of chylomicron secretion.

Collectively, our results provided clear evidence that DARS2 ablation in enterocytes caused Golgi disorganization and suppressed the formation of mature chylomicrons and the transport of dietary lipids resulting in their accumulation in large lipid droplets. These findings were confirmed using different experimental approaches, including EM, immunofluorescence microscopy, immunoblot analysis of plasma lipoproteins and metabolic tracing studies.

This work would be potentially interesting if the authors elucidated why and how mitochondrial dysfunction affects Golgi disorganization and chylomicrons release, but the current data do not establish such clear connections

Our work reveals a new and previously unappreciated function of mitochondria in dietary lipid processing including chylomicron production and Golgi dysfunction. Our newest data in cell culture as well as in *C. elegans* (see Figures 1 and 2 for reviewer above and new ED Figure 11 in the revised manuscript) support that this is primary defect and not a secondary consequence of organ or organismal failure and show strong conservation of this pathway across phyla, highlighting its importance. Furthermore, this exciting new finding is highly relevant for mitochondrial disease patients, or patients on long term antiviral therapies, as both suffer from the OXPHOS impairment and poorly understood, understudied gastrointestinal problems. As already stated in our response to reviewer 1, dissecting the precise underlying mechanisms will be the natural next step which, however, may take years and therefore goes beyond the scope of the current study.

References

- Auclair N, Sane AT, Ahmarani L, Patey N, Beaulieu JF, Peretti N, Spahis S, Levy E (2021) Sar1b mutant mice recapitulate gastrointestinal abnormalities associated with chylomicron retention disease. *J Lipid Res* 62: 100085
- Baker ME (1988) Is vitellogenin an ancestor of apolipoprotein B-100 of human low-density lipoprotein and human lipoprotein lipase? *Biochem J* 255: 1057-1060
- Baklava Z, Pant S, Fares H, Grant BD (2007) Genome-wide analysis identifies a general requirement for polarity proteins in endocytic traffic. *Nat Cell Biol* 9: 1066-1073
- Gordon SM, Neufeld EB, Yang Z, Pryor M, Freeman LA, Fan X, Kullo IJ, Biesecker LG, Remaley AT (2019) DENND5B Regulates Intestinal Triglyceride Absorption and Body Mass. *Sci Rep* 9: 3597
- Iqbal J, Parks JS, Hussain MM (2013) Lipid absorption defects in intestine-specific microsomal triglyceride transfer protein and ATP-binding cassette transporter A1-deficient mice. *J Biol Chem* 288: 30432-30444
- Quaroni A, Wands J, Trelstad RL, Isselbacher KJ (1979) Epithelioid cell cultures from rat small intestine. Characterization by morphologic and immunologic criteria. *J Cell Biol* 80: 248-265
- Schmidt C, Sciacovelli M, Frezza C (2020) Fumarate hydratase in cancer: A multifaceted tumour suppressor. *Semin Cell Dev Biol* 98: 15-25
- van Es JH, Haegebarth A, Kujala P, Itzkovitz S, Koo BK, Boj SF, Korving J, van den Born M, van Oudenaarden A, Robine S *et al* (2012) A critical role for the Wnt effector Tcf4 in adult intestinal homeostatic self-renewal. *Mol Cell Biol* 32: 1918-1927
- Xu N, Zhang SO, Cole RA, McKinney SA, Guo F, Haas JT, Bobba S, Farese RV, Jr., Mak HY (2012) The FATP1-DGAT2 complex facilitates lipid droplet expansion at the ER-lipid droplet interface. *J Cell Biol* 198: 895-911
- Young SG, Cham CM, Pitas RE, Burri BJ, Connolly A, Flynn L, Pappu AS, Wong JS, Hamilton RL, Farese RV, Jr. (1995) A genetic model for absent chylomicron formation: mice producing apolipoprotein B in the liver, but not in the intestine. *J Clin Invest* 96: 2932-2946
- Zhao B, Chen Y, Jiang N, Yang L, Sun S, Zhang Y, Wen Z, Ray L, Liu H, Hou G *et al* (2019) Znhit1 controls intestinal stem cell maintenance by regulating H2A.Z incorporation. *Nature communications* 10: 1071

Reviewer Reports on the Third Revision:

Referees' comments:

Referee #1 (Remarks to the Author):

This revision convincingly argues the following points: (1) mitochondria dysfunction leads to the LDs accumulation; (2) LD accumulation is not likely caused by cellular or organismal death, clarified by additional cell culture data that de-couple organismal death from LD accumulation; (3) There may be conserved biology of mitochondrial regulation of protein/lipid secretion, as supported by *C. elegans* vitellogenin studies that model intestine to oocyte protein deposition. The authors have done a thorough job answering most of my concerns. Although the exact mechanism(s) whereby mitochondria dysfunction leads to LD accumulation will require significant future investigation, I feel the brute force and rigorous nature of the current study (and prominent phenotype) will be of broad interest to the readership of *Nature*. Finally, I encourage authors to include data provided for the reviewers in the extended data section to strengthen the manuscript's message.

Referee #4 (Remarks to the Author):

The revised manuscript does not add substantial new data to clarify the underlying mechanism or physiological relevance of the descriptive mouse phenotype. Importantly major concerns from prior rounds of review have not been addressed as outlined below.

1) The speculation regarding Golgi disorganization is not adequately supported by experimental evidence. I accept the authors' description of a big phenotypical effect in term of intestinal absorption of lipids. That is very well documented. What is not convincing is the molecular explanation of this phenotypical effect. The conclusion of Golgi disorganization is based on EM micrographs and immunostaining, and this is simply not enough to prove the point.

The manuscript reports: "In contrast, an extensive disorganisation of the secretory pathway was observed in enterocytes of DARS2IEC-KO mice, which showed a profound lack of Golgi cisternae containing CM (Fig. 5c)."

This observation can result simply from more of having more cytosolic lipid droplets that occupy more space in the cell body and restrain other organelles, including ER and Golgi. In the EM micrographs for DARS2IEC-KO mice included in fig. 5c there is no Golgi, which makes it difficult to conclude that it is a matter of Golgi disorganization. If the authors believe that Golgi disorganization is the driving force of the observed phenotype, they should do specific experiments to prove it. What would happen if they disorganize Golgi in F/F and *Dars2* deficient mice (by treating with brefeldin A for example)?

<https://journals.physiology.org/doi/epdf/10.1152/ajpgi.1994.266.2.G292>

2) Prior comment: "this work would be potentially interesting if the authors elucidated why and how mitochondrial dysfunction affects Golgi organization and chylomicrons release".

But this remains unanswered. This paper describes a sick mouse with a prominent intestinal phenotype but it is not clear how these findings relate to physiology. What is needed is proof that Golgi disorganization is the key event and to show how this is linked mitochondrial dysfunction. How is disrupting mitochondrial function affecting Golgi organization and integrity?

As it stands, the paper is still largely descriptive and lacks sufficient mechanistic insight or physiological relevance to be of broad interest.

Author Rebuttals to Third Revision:

Referees' comments:

Referee #1 (Remarks to the Author):

This revision convincingly argues the following points: (1) mitochondria dysfunction leads to the LDs accumulation; (2) LD accumulation is not likely caused by cellular or organismal death, clarified by additional cell culture data that de-couple organismal death from LD accumulation; (3) There may be conserved biology of mitochondrial regulation of protein/lipid secretion, as supported by *C. elegans* vitellogenin studies that model intestine to oocyte protein deposition. The authors have done a thorough job answering most of my concerns. Although the exact mechanism(s) whereby mitochondria dysfunction leads to LD accumulation will require significant future investigation, I feel the brute force and rigorous nature of the current study (and prominent phenotype) will be of broad interest to the readership of Nature. Finally, I encourage authors to include data provided for the reviewers in the extended data section to strengthen the manuscript's message.

We wish to thank the reviewer for their very positive evaluation of our manuscript and for recognizing the rigorous nature of our work and the novelty of our findings. All the data provided in the previous response to the reviewers are included in Extended Data Fig. 11.

Referee #4 (Remarks to the Author):

The revised manuscript does not add substantial new data to clarify the underlying mechanism or physiological relevance of the descriptive mouse phenotype. Importantly major concerns from prior rounds of review have not been addressed as outlined below.

1) The speculation regarding Golgi disorganization is not adequately supported by experimental evidence. I accept the authors' description of a big phenotypical effect in term of intestinal absorption of lipids. That is very well documented. What is not convincing is the molecular explanation of this phenotypical effect. The conclusion of Golgi disorganization is based on EM micrographs and immunostaining, and this is simply not enough to prove the point.

The manuscript reports: "In contrast, an extensive disorganisation of the secretory pathway was observed in enterocytes of DARS2IEC-KO mice, which showed a profound lack of Golgi cisternae containing CM (Fig. 5c)."

This observation can result simply from more of having more cytosolic lipid droplets that occupy more space in the cell body and restrain other organelles, including ER and Golgi. In the EM micrographs for DARS2IEC-KO mice included in fig. 5c there is no Golgi, which makes it difficult to conclude that it is a matter of Golgi disorganization. If the authors believe that Golgi disorganization is the driving force of the observed phenotype, they should do specific experiments to prove it. What would happen if they disorganize Golgi in F/F and Dars2 deficient mice (by treating with brefeldin A for example)?

<https://journals.physiology.org/doi/epdf/10.1152/ajpgi.1994.266.2.G292>

We were surprised to read the comments of reviewer 4 that our manuscript does not provide sufficient evidence to support that mitochondrial dysfunction causes disorganisation of the Golgi in enterocytes, which could explain the lipid processing phenotype. First, the reviewer questions the technical soundness of our experiments

assessing the Golgi phenotype. Specifically, the reviewer writes: “*The conclusion of Golgi disorganization is based on EM micrographs and immunostaining, and this is simply not enough to prove the point.*” With due respect to the reviewer’s opinion, we disagree with their assessment that our experiments are not sufficient to support the conclusion that mitochondrial dysfunction causes Golgi disorganisation. The reviewer argues that EM analysis and immunostaining is not enough to prove this point. However, we would like to stress that **the terms Golgi disorganization, fragmentation or integrity are by default morphological terms and can only be described by microscopy techniques, such as the EM and immunofluorescence analysis we performed.** As the reviewer points out, we certainly considered the possibility that in enterocytes with massive accumulation of giant lipid droplets (LDs) all other organelles could be constrained in limited areas of the cytoplasm making it difficult to be recognized in EM micrographs. However, **we could rule out this possibility** by performing confocal microscopy and being able to inspect the entire volume of enterocytes by performing z-scans. Importantly, the fact that the fragmentation/dispersal of the transmembrane Golgi marker is measurable already at 5-days after tamoxifen injections and before the appearance of massive LDs (Figure 5d and ED Figure 10a) argues against Golgi disorganization being a mere artifact of an LD-filled cytoplasm. We have further convincingly shown that treatment with actinonin (inhibitor of mitochondrial translation) and Atpenin A5 (SDH inhibitor) induced Golgi dispersal in cultured IEC-6 enterocytes, and this defect was further exaggerated by the addition of lipids in the culture medium (ED Figure 11a, b). Additional support that mitochondrial dysfunction compromises the Golgi is provided from our results in *C. elegans* showing that DARS-2 depletion caused **profound decrease in Golgi, but not the ER** in the worm gut (ED Figure 11c). Furthermore, using VIT-2 transport as an established system for COPII vesicle formation and secretory pathway function in *C. elegans*, we showed that DARS2 knockdown caused impaired lipid transport in addition to the Golgi defect (ED Figure 11c). Collectively, our experiments in mice, together with our results in cultured enterocytes and in *C. elegans*, provide convincing evidence supporting that mitochondrial dysfunction causes Golgi disorganisation.

Furthermore, a more critical question here is whether a Golgi apparatus that has lost its stacked morphology can still mediate anterograde transport of secretory cargo. The answer is **yes**, cells can still be competent for secretion even in the presence of a vesiculated, rather than a stacked Golgi apparatus. Therefore, it was important to assess the competence of DARS2-deficient enterocytes to secrete dietary lipids in the form of chylomicrons in the blood circulation. We performed exactly this experiment, which provided *in vivo* **functional** (not morphological) evidence exemplifying the impaired secretory capacity of the DARS2-deficient enterocytes. Specifically, our metabolic tracing experiments presented in ED Figure 9 demonstrated that already 5 days after tamoxifen induction the mice showed impaired transport of dietary lipids to the circulation. Furthermore, the immunoblot analysis of plasma lipoproteins presented in Figure 5b revealed a clear reduction of the protein levels of ApoB48 in DARS2^{lamIEC-KO} mice at both 5 and 7 days after tamoxifen, whereas ApoB48 levels in enterocytes were not decreased in these animals (Figure 5a). ApoB48 is exclusively a component of enterocyte-produced chylomicrons, therefore these experiments provided additional support that DARS2-deficient enterocytes showed impaired chylomicron production and release already 5 days after tamoxifen. Taken together, **our experiments**

provided clear evidence that mitochondrial dysfunction causes Golgi fragmentation and impaired chylomicron production, resulting in diminished transport of dietary lipids and their accumulation in large lipid droplets in enterocytes.

Second, in their next comment, the reviewer questions whether the disorganization of the Golgi could explain the lipid transport defect in enterocytes. Specifically, the reviewer writes: *“If the authors believe that Golgi disorganization is the driving force of the observed phenotype, they should do specific experiments to prove it. What would happen if they disorganize Golgi in F/F and Dars2 deficient mice (by treating with brefeldin A for example)*

<https://journals.physiology.org/doi/epdf/10.1152/ajpgi.1994.266.2.G292>”.

This is indeed a very good experiment to assess whether Golgi disorganization could cause an impairment of dietary lipid processing in enterocytes. **However, this evidence is already provided in the paper cited by the reviewer.** In the cited study, Mansbach and Nevin treated rats with brefeldin A (BFA) and assessed how the resulting disruption of the secretory pathway affected dietary lipid processing and release in the form of chylomicrons. Their experiments provided robust experimental evidence demonstrating that indeed, disruption of the Golgi by BFA treatment led to impaired chylomicron production and transport of dietary lipids to the periphery (Figures 1, 2 and 3 of the cited paper), resulting in the accumulation of dietary lipids in large lipid droplets in enterocytes (Figure 5 of the cited paper). **Thus, acute disruption of anterograde transport and Golgi disorganisation by BFA treatment in rats recapitulates the lipid processing phenotype caused by mitochondrial dysfunction in enterocytes,** providing further support to our conclusion that disruption of the Golgi organization is **causative** to the observed chylomicron transport defect and lipid accumulation in LDs in enterocytes of our mice. With due respect, we do not understand the suggestion of the reviewer that we should repeat the same experiment, if this study has already been performed and provided clear evidence supporting our conclusions. As a side comment, the suggestion to treat not only control (F/F) but also the Dars2-deficient mice with BFA does not make sense as these animals already have a severely affected Golgi and there is no scope in exposing them to BFA to make it even worse.

2) Prior comment: “this work would be potentially interesting if the authors elucidated why and how mitochondrial dysfunction affects Golgi organization and chylomicrons release”.

But this remains unanswered. This paper describes a sick mouse with a prominent intestinal phenotype but it is not clear how these findings relate to physiology. What is needed is proof that Golgi disorganization is the key event and to show how this is linked mitochondrial dysfunction. How is disrupting mitochondrial function affecting Golgi organization and integrity?

As it stands, the paper is still largely descriptive and lacks sufficient mechanistic insight or physiological relevance to be of broad interest.

As explained in the previous paragraph, proof that Golgi disorganisation is the key event is provided by the study of Mansbach and Nevin in the paper cited by the

reviewer, therefore this question is already answered. Hence, the only remaining critique of the reviewer is that we do not provide a precise and detailed molecular mechanism of how exactly mitochondrial dysfunction affects Golgi organization. While we understand that the reviewer wants to see more mechanistic details on how impaired mitochondrial function causes Golgi defects, we respectfully suggest that such analysis goes beyond the scope of the current manuscript. Importantly, the other reviewers disagreed with reviewer 4, clearly stating that our manuscript includes sufficient novel, interesting and important findings and that a deeper mechanistic interrogation is not necessary for publication. Indeed, this manuscript not only reveals a previously unappreciated evolutionary conserved role of mitochondria in the transport of dietary lipids, but also establishes a novel causative link between mitochondrial function and the maintenance of Golgi integrity. These results open a whole new area of research and will prompt many scientists in the field to take a fresh look into mitochondrial dysfunction in multiple pathophysiological conditions.

Reviewer Reports on the Fourth Revision:

Referees' comments:

Referee #1 (Remarks to the Author):

As noted in my previous comments, the strength of this paper is that: (1) Multiple in vivo genetic models to impact mitochondrial function have been employed, including DARS2IEC-KO, SDHAIEC-KO, COX10IEC-KO, inducible DARS2IEC-KO; (2) The phenotypic description of these mutants are well done with multi-pronged approaches including microscopy, transcriptomics and proteomics methods – all pointing to the findings that these mutants having altered lipid absorption phenotypes; (3) The authors have included evidence to support that mitochondrial defects can cause chylomicron (CM) or lipid secretion in the cell culture, decoupling organismal death from CM secretion; (4) the *C. elegans* vitellogenin model under the *dars* RNAi regimen further strengthens the argument that mitochondrial defects can cause lipid transport defects and this may be a conserved feature across phyla. With multiple rounds of revisions, the authors tried to delineate organismal death or general enterocyte malfunction from specific defects in lipid transport. With additional data, the manuscript at this current stage is much more complete than previous submissions. Although we do not know the exact mechanism of mitochondrial regulation of CM secretion, the observations put forth by this paper is of general interest to a broad readership, and the findings may be applicable to considering dietary regimen (ex., fat-free diet) to patients who have mitochondrial defects in chronic or acute disease settings. Given that this paper has gone through multiple revisions with improved data to support the authors' claims, I think it is a strong candidate for publication in Nature.

Referee #5 (Remarks to the Author):

I read through the reviews of the paper by Moschandrea et al., and also went through the paper. The authors threw the kitchen sink at analyzing the role of mitochondria dysfunction in enterocytes, which was a bit overwhelming to read. Not surprisingly, they could not explain everything mechanistically. But there were some important findings that were noteworthy-including that mitochondria dysfunction in enterocytes impacts chylomicron production and trafficking, leading to an inhibition of transport of dietary lipids to peripheral organs. Because I think these findings will be of interest to researchers trying to understand why patients with mitochondrial diseases exhibit intestinal and nutritional defects, I think the study is worthy of publication in Nature. That said, I agree with Reviewer 4 that there is much mechanism lacking, but I think their overall data lean strongly to a defect in secretory trafficking in these mitochondria perturbed systems, which is very exciting. Thus, I recommend publication.

Author Rebuttals to Fourth Revision:

Referees' comments:

Referee #1 (Remarks to the Author):

As noted in my previous comments, the strength of this paper is that: (1) Multiple in vivo genetic models to impact mitochondrial function have been employed, including DARS2IEC-KO, SDHAIEC-KO, COX10IEC-KO, inducible DARS2IEC-KO; (2) The phenotypic description of these mutants are well done with multi-pronged approaches including microscopy, transcriptomics and proteomics methods – all pointing to the findings that these mutants having altered lipid absorption phenotypes; (3) The authors have included evidence to support that mitochondrial defects can cause chylomicron (CM) or lipid secretion in the cell culture, decoupling organismal death from CM secretion; (4) the *C. elegans* vitellogenin model under the *dars* RNAi regimen further strengthens the argument that mitochondrial defects can cause lipid transport defects and this may be a conserved feature across phyla. With multiple rounds of revisions, the authors tried to delineate organismal death or general enterocyte malfunction from specific defects in lipid transport. With additional data, the manuscript at this current stage is much more complete than previous submissions. Although we do not know the exact mechanism of mitochondrial regulation of CM secretion, the observations put forth by this paper is of general interest to a broad readership, and the findings may be applicable to considering dietary regimen (ex., fat-free diet) to patients who have mitochondrial defects in chronic or acute disease settings. Given that this paper has gone through multiple revisions with improved data to support the authors' claims, I think it is a strong candidate for publication in Nature.

We wish to thank the reviewer for their constructive critique throughout the multiple rounds of review, which helped us to further improve our manuscript. We are glad that the reviewer finds our revised manuscript suitable for publication in Nature and appreciate their support.

Referee #5 (Remarks to the Author):

I read through the reviews of the paper by Moschandrea et al., and also went through the paper. The authors threw the kitchen sink at analyzing the role of mitochondria dysfunction in enterocytes, which was a bit overwhelming to read. Not surprisingly, they could not explain everything mechanistically. But there were some important findings that were noteworthy- including that mitochondria dysfunction in enterocytes impacts chylomicron production and trafficking, leading to an inhibition of transport of dietary lipids to peripheral organs. Because I think these findings will be of interest to researchers trying to understand why patients with mitochondrial diseases exhibit intestinal and nutritional defects, I think the study is worthy of publication in Nature. That said, I agree with Reviewer 4 that there is much mechanism lacking, but I think their overall data lean strongly to a defect in secretory trafficking in these mitochondria perturbed systems, which is very exciting. Thus, I recommend publication.

We are glad that the reviewer recognized the novelty of our findings and appreciated the amount of work that we have performed in our efforts to address the role of mitochondria in the regulation of intestinal lipid transport. We wish to thank the reviewer for their very positive evaluation of our manuscript and for recommending publication in Nature.